# Single-cell brain organoid screening identifies developmental defects in autism

Chong Li[1,6 ✉], Jonas Simon Fleck[2,6], Catarina Martins-Costa[1], Thomas R. Burkard[1], Jan Themann[1], Marlene Stuempflen[3], Angela Maria Peer[1], Ábel Vertesy[1], Jamie B. Littleboy[1], Christopher Esk[1,4], Ulrich Elling[1], Gregor Kasprian[3], Nina S. Corsini[1], Barbara Treutlein[2 ✉] & Juergen A. Knoblich[1,5 ✉]

The development of the human brain involves unique processes (not observed in many other species) that can contribute to neurodevelopmental disorders[1–4]. Cerebral organoids enable the study of neurodevelopmental disorders in a human context. We have developed the CRISPR–human organoids–single-cell RNA sequencing (CHOOSE) system, which uses verified pairs of guide RNAs, inducible CRISPR–Cas9-based genetic disruption and single-cell transcriptomics for pooled loss-of-function screening in mosaic organoids. Here we show that perturbation of 36 high-risk autism spectrum disorder genes related to transcriptional regulation uncovers their effects on cell fate determination. We find that dorsal intermediate progenitors, ventral progenitors and upper-layer excitatory neurons are among the most vulnerable cell types. We construct a developmental gene regulatory network of cerebral organoids from single-cell transcriptomes and chromatin modalities and identify autism spectrum disorder-associated and perturbation-enriched regulatory modules. Perturbing members of the BRG1/BRM-associated factor (BAF) chromatin remodelling complex leads to enrichment of ventral telencephalon progenitors. Specifically, mutating the BAF subunit *ARID1B* affects the fate transition of progenitors to oligodendrocyte and interneuron precursor cells, a phenotype that we confirmed in patient-specific induced pluripotent stem cell-derived organoids. Our study paves the way for high-throughput phenotypic characterization of disease susceptibility genes in organoid models with cell state, molecular pathway and gene regulatory network readouts.

Human cortical development involves unique and intricate processes. Following neural tube formation, neuroepithelial cells within the telencephalon proliferate, expand and generate radial glial progenitors, intermediate progenitors and outer radial glial progenitors. In the dorsal region, these progenitors give rise to layered excitatory neurons. In the ventral telencephalon, they generate interneurons that migrate into the dorsal cortex to integrate with excitatory neurons. These processes are governed by precise and highly orchestrated genetic and molecular programs, many of which have remained elusive[3]. Research into neurodevelopmental disorders (NDDs) has advanced our understanding of human brain development and helped to reveal how it can go awry. However, many NDDs, such as autism spectrum disorder (ASD), are diagnosed only after birth, when brain development is almost complete. Analysing the developmental and cell type-specific defects associated with ASD in a human context is crucial but is often constrained to neuroimaging and postmortem tissue studies. Moreover, coexpression network analyses of ASD genes indicate that the developmental defects

associated with ASD may arise during fetal stages[5,6], periods that are difficult to investigate.

Studying the genetic aetiology of NDDs enhances our understanding of disease mechanisms[1,7,8], but it usually requires access to the developmental processes of the human brain. Brain organoids recapitulate early brain development and generate diverse cell types found in vivo[9]. Although organoids have been used to investigate disease-associated genes[9–11], they are limited by phenotypic variability and low throughput. Recent studies combining CRISPR screening technology with organoids have revealed the power of such strategies for discovering new gene functions[12,13]. However, such screens are limited by low-content readouts, which often use guide RNA (gRNA) counts to assess proliferation phenotypes when challenged with genetic perturbations. Although CRISPR screening coupled with single-cell transcriptomic readout provides unprecedented resolution for phenotypic characterization[14–16], such approaches have not been fully explored in organoids. The feasibility of single-cell perturbation screening in heterogeneous

[1]Institute of Molecular Biotechnology of the Austrian Academy of Science (IMBA), Vienna, Austria. [2]Department of Biosystems Science and Engineering, ETH Zürich, Basel, Switzerland. [3]Department of Radiodiagnostics, Medical University of Vienna, Vienna, Austria. [4]Institute of Molecular Biology, University of Innsbruck, Innsbruck, Austria. [5]Department of Neurology, Medical University of Vienna, Vienna, Austria. [6]These authors contributed equally: Chong Li, Jonas Simon Fleck. ✉e-mail: chong.li@imba.oeaw.ac.at; barbara.treutlein@bsse.ethz.ch; juergen.knoblich@imba.oeaw.ac.at

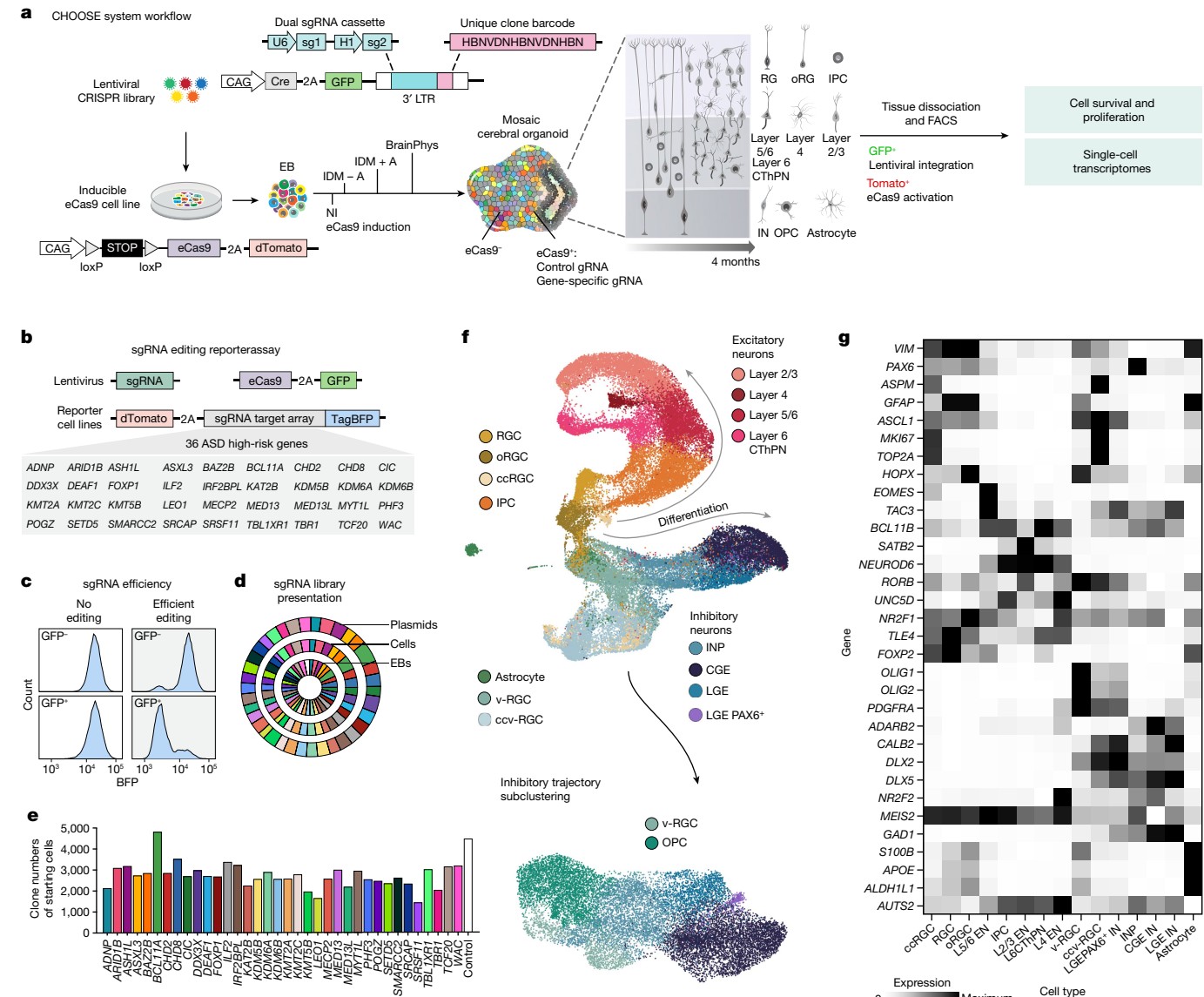

**Fig. 1 | The CHOOSE system for multiplexed screening of ASD risk genes in human cerebral organoids. a**, CHOOSE system overview. Barcoded dual sgRNA cassette located within the 3′ LTR of the lentivirus. **b**, Reporter assay to test gRNA efficiencies for 36 ASD risk genes. **c**, Editing efficiencies of gRNAs determined by flow cytometry. Plots show examples of gRNAs with no or efficient editing. **d**, sgRNA sequence read distributions of gRNAs sequenced from the ASD plasmid library, lentivirus-infected hES cells and embryoid bodies at day 5. **e**, Numbers of clones from the starting hES cells for each perturbation used to generate mosaic cerebral organoids. Control, non-targeting gRNAs. **f**, UMAP embedding of the scRNA-seq dataset containing dorsal and ventral telencephalon trajectories. Subclustering and UMAP embedding of the ventral telencephalon trajectory excluding astrocytes and ccv-RGCs to annotate the OPC cluster. **g**, Heat map showing the expression of marker genes in different cell types. ccRGC, cycling radial glial cell; CThPN, cortical thalamic projection neuron; oRGC, outer radial glial cell; IDM, improved differentiation medium; NI, neural induction.

tissues that undergo long-term differentiation and consist of diverse cell types remains unclear.

Here, we describe the CHOOSE system which combines parallel genetic perturbations with single-cell transcriptomic readout in mosaic cerebral organoids. We deliver barcoded pairs of gRNAs as a pooled lentiviral library to stem cells and generate telencephalic organoids to identify the loss-of-function phenotypes of 36 high-risk ASD genes at the level of cell types and molecular pathways. Using single-cell multiomic data, we construct a developmental gene regulatory network (GRN) of cerebral organoids and identify ASD-enriched regulatory hubs connected to the genes that are dysregulated in response to genetic perturbations. Among the 36 genes, one of the most significant changes in cell type composition was identified in the context of *ARID1B*. Specifically, perturbing *ARID1B* expands ventral radial glia

cells and increases their transition to early oligodendrocyte precursor cells (OPCs), a phenotype we verify in brain organoids generated from *ARID1B* patient-derived induced pluripotent stem cell (iPS cell) lines.

## Single-cell CRISPR screening in barcoded organoids

Single-cell RNA sequencing (scRNA-seq) is a high-throughput method used to analyse cellular heterogeneity in complex tissues. To establish an organoid system that enables CRISPR perturbations with single-cell transcriptomic readout, we used a human embryonic stem cell (hES cell) line expressing an enhanced specificity SpCas9 (eCas9), which has substantially reduced off-target effects and is controlled by an upstream loxp stop element[12] (Fig. 1a). To regulate eCas9 induction, we engineered a lentiviral vector to deliver 4-hydroxytamoxifen-inducible CRE

recombinase and a dual single-guide RNA (sgRNA) cassette (Fig. 1a). The dual sgRNA consists of two sgRNAs targeting the same gene, expressed under the *U6* or *H1* promoter. The dual gRNA is located within the 3′ long terminal repeat (LTR) and is thus transcribed by RNA polymerase II to be captured by scRNA-seq assays[17]. To ensure efficient generation of loss-of-function alleles, we determined the editing efficiency of each sgRNA pair using a flow cytometry-based gRNA reporter assay (Fig. 1b and Extended Data Fig. 1a). In this assay, a pre-assembled array of gRNA-targeting sequences fused with TagBFP is used to generate 3T3 fibroblast reporter cell lines. sgRNA and eCas9 are then delivered into the reporter cell lines by lentiviruses. Successful genome editing causes frameshift mutations, resulting in the loss of blue fluorescent protein (BFP) fluorescence, which enables the quantitative evaluation of gRNA efficiency (Fig. 1c and Extended Data Fig. 1b,c), although it does not allow for the determination of whether a heterozygous or homozygous mutation was introduced. Using our reporter assay, we selected efficient sgRNA pairs for 36 ASD genes (Extended Data Fig. 1d,e and Supplementary Table 1). Immunohistochemistry analysis of several perturbations in organoids further confirmed the loss of protein products for the majority of selected gRNAs (Extended Data Fig. 2).

We individually cloned sgRNA pairs and pooled them equally to construct a lentiviral plasmid library (Extended Data Fig. 3a,b). To ensure that lentiviral integration frequency was limited to one per cell, we used a low infection rate of 2.5% (ref. 18) (Extended Data Fig. 3c–e). Our analysis indicated homogeneous distribution of the gRNAs in both the plasmid library and hES cells, which was maintained after the formation of embryoid bodies (Fig. 1d). It is important to note that the development of human brain tissues exhibits the generation of clones with highly variable sizes both in vivo and in vitro[12,19]. To monitor the clonal complexity of the founder cells, we introduced a unique clone barcode (UCB; $1.4 \times 10^7$ combinations) for each dual sgRNA cassette to label individual lentiviral integration events (Fig. 1a and Extended Data Fig. 4a,b). Using this strategy, we obtained an average of 2,770 unique clones for each perturbation, which we used to generate mosaic embryoid bodies (Fig. 1e). Altogether, we established a highly efficient and controlled pooled screening system with high clonal complexity in the organoid.

## CHOOSE organoids generate diverse cell types

Cortical abnormalities are a prominent feature of ASD[8,20]. Many ASD risk genes associated with transcriptional regulation and chromatin remodelling are crucial for cortical development[21,22]. Therefore, we aimed to leverage our methodology to explore loss-of-function phenotypes for 36 transcriptional control genes with high ASD causal confidence (Simons Foundation Autism Research Initiative (SFARI) gene score 1)[7].

We used previously established protocols that reproducibly generate human telencephalon organoids[23,24] (Extended Data Fig. 5a–c). eCas9 was induced in 5-day-old embryoid bodies, followed by neural induction. Fluorescence-activated cell sorting (FACS)-based analyses suggest that mutant cells (GFP+/dTomato+) remained at low percentages throughout development, with an average of 21.8% on day 120 (Extended Data Fig. 5d–f). It is likely that this could limit the mutant cell–cell interactions within the mosaic tissues. Single-cell transcriptome profiling of cerebral organoids at 4 months revealed a large diversity of dorsal and ventral telencephalon cell populations (Fig. 1f,g and Extended Data Fig. 6a–j) (14 independent pools of organoids, 3–7 organoids each pool, 65 organoids in total, three independent batches). We first annotated cell clusters based on control cells (non-targeting gRNA control) and eCas9-uninduced cells (35,203 cells) (Extended Data Fig. 6d–f). Cell-type labels for the full CHOOSE dataset (49,754 cells) were then derived through a label transfer. Broadly, we found that perturbed cells adopted cell states resembling those found in the unperturbed controls (Fig. 1f). We identified progenitor cells with dorsal (*PAX6*) or ventral (*ASCL1*, *OLIG2*) origins. Among the cells with

dorsal identity, we identified radial glial cells (RGCs; *VIM*), cycling RGCs (*ASPM*), outer RGCs (oRGCs; *HOPX*) and intermediate progenitor cells (IPCs; *EOMES*). These progenitors differentiated into excitatory neurons with specific layer identities, including layer 5/6 neurons (L5/6; *BCL11B*), L6 cortical thalamic projection neurons (*FOXP2*, *TLE4*)[25], L4 neurons (*RORB*, *UNC5D*, *NR2F1*)[26,27] and L2/3 neurons (*SATB2*). Ventral radial glial cells (v-RGCs; cell cycling ventral RGCs, ccv-RGCs) differentiated into interneuron precursor cells (INPs; *DLX2*), which generated interneurons with lateral ganglionic eminence (LGE) origin (LGE-IN; *MEIS2*) or caudal ganglionic eminence (CGE) origin (CGE-IN; *NR2F2*)[28]. Notably, we found a cluster of interneurons expressing *MEIS2* and high levels of *PAX6* (LGE PAX6+ interneurons), a signature that resembles mouse olfactory bulb precursors that were recently reported to generate neurons redirected to white matter in primates[29]. In addition to neuronal populations, we identified glial cell populations including astrocytes (*S100B*, *APOE*, *ALDH1L1*) and OPCs (*OLIG2*, *PDGFRA*) (Fig. 1f,g). RNA velocity analysis[30,31] revealed developmental trajectories from neural progenitor cells to neuronal populations in both the dorsal and ventral telencephalons (Extended Data Fig. 6k). Further analysis of our organoid dataset and a primary developing human brain dataset[32] revealed cell type-specific expression patterns for several ASD genes that we targeted in our screen (Extended Data Fig. 6l,m). In summary, our scRNA-seq dataset of 4-month-old cerebral organoids recapitulates diverse telencephalic cell types that are present in the developing human brain.

## Cell proliferation and depletion phenotypes

Aberrant cell proliferation during brain development has been suggested to contribute to ASD pathology[33]. To test whether ASD genetic perturbations could affect cell proliferation, we recovered gRNA information from scRNA-seq complementary DNA libraries as well as bulk-extracted genomic DNA from organoid pools of four different batches (Extended Data Fig. 7a). We observed a heterogenous gRNA representation in eCas9-induced cells at 4 months. A time course analysis revealed a deviation from the initial gRNA distribution as early as day 20 (Extended Data Fig. 7b). When comparing eCas9-induced with uninduced cells from scRNA-seq libraries, we found significantly enriched and depleted gRNAs (Fig. 2a,b) (induced cells, $n = 14$ independent pools of organoids, three batches; uninduced cells, $n = 8$ independent pools of organoids, two batches). Using a FACS-based approach, we further confirmed the enrichment and depletion phenotypes in individually perturbed organoids for the four genes (*KMT2C*, *LEO1*, *ADNP* and *WAC*) that showed the largest effect sizes (Extended Data Fig. 8).

To ensure that the observed cell proliferation phenotypes were not driven by clonal effects, we determined the complexity of clones recovered from scRNA-seq libraries. On average, we recovered 125 clones for each perturbation and found that the clones were distributed across all libraries (Extended Data Fig. 4c,d). Analysing the size of each clone, we found a mean average cell number per clone of 4.4 (Extended Data Fig. 4e,f). These data suggest that cells captured in the CHOOSE screen came from diverse and relatively small clones, which are crucial for mitigating dominant clonal effects. Bulk analysis of the genomic DNA with a higher cell number input (50,000–150,000) revealed high clonal complexity in both eCas9-uninduced and eCas9-induced cells, with a homogenous distribution only in uninduced cells (Extended Data Fig. 4g,h). In conclusion, using a pooled, high-complexity barcoding screening system, we successfully identified ASD risk genes that play essential roles in cell proliferation and survival.

## Cell-type-specific effects of perturbing ASD genes

Cell type-specific alterations have been observed in patients with ASD and brain organoid disease models[11,34]. The large cellular diversity detected in our system enables us to systematically assess, compare

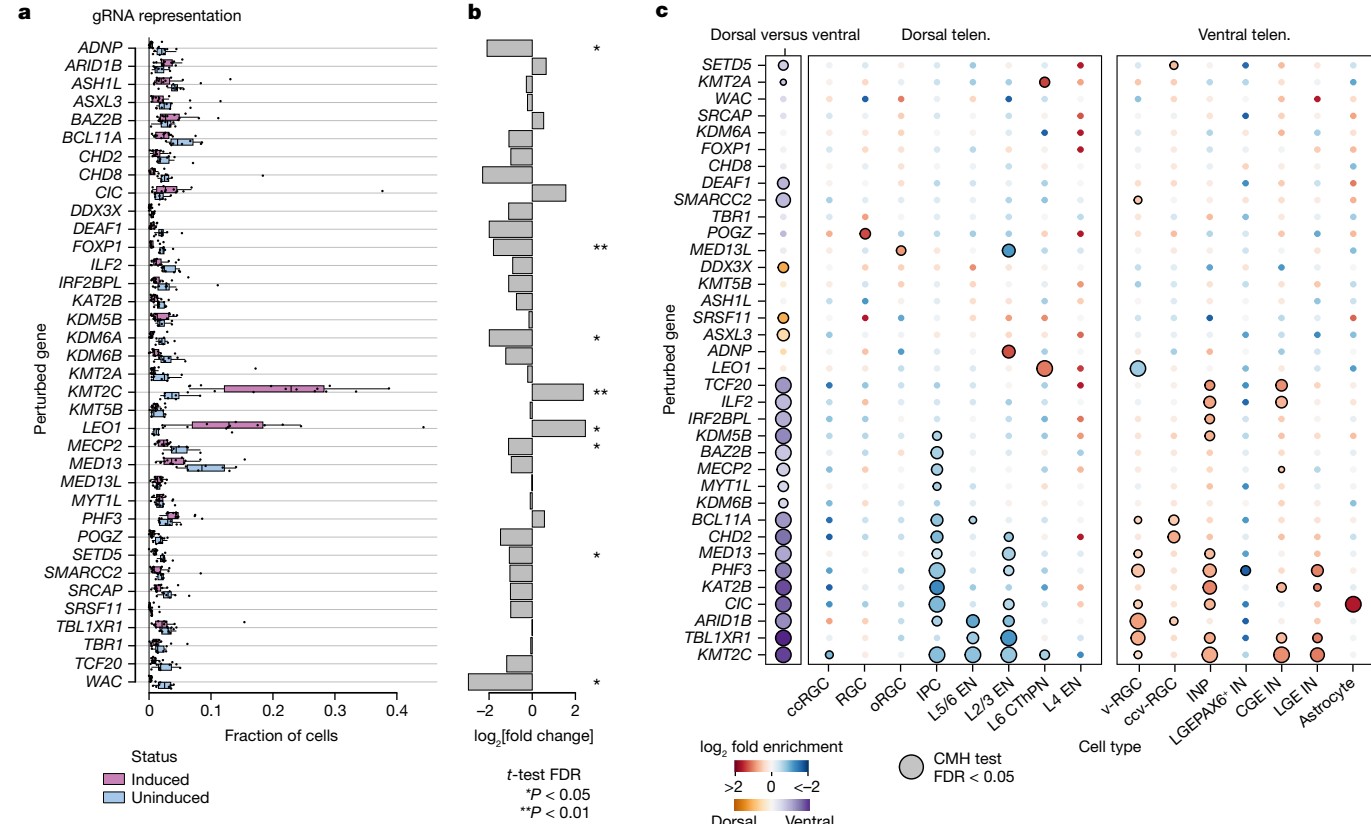

**Fig. 2 | Cell type-specific effects of ASD risk gene perturbations. a**, Box plot showing the representation of gRNAs in eCas9-induced and uninduced organoid pools (induced cells, $n = 14$ independent pools of organoids, three batches; uninduced cells, $n = 8$ independent pools of organoids, two batches). The centre lines represent the medians, the box limits show the 25–75% interquartile ranges and the whiskers indicate 1.5× the interquartile ranges.

**b**, Bar plot showing fold change of gRNA representation in eCas9-induced versus uninduced cell pools. **c**, Heat map showing enrichment of gRNAs versus control in dorsal versus ventral telencephalon (telen.) cells (left panel, orange to purple) and individual cell types (centre and right panels, red to blue). Colours indicate the log odds ratio, and sizes indicate $-\log_{10}$ FDR-corrected $P$ values of a two-sided CMH test stratified by organoid pool.

and categorize the effects of ASD gene perturbations on cell states. Using a Cochran–Mantel–Haenszel (CMH) test stratified by library replicates, we first assessed the differential abundance of dorsal versus ventral telencephalic cells, as well as the abundance of each individual cell type in each perturbation versus a non-targeting gRNA control (11 independent pools of organoids, two independent batches) (Fig. 2c and Extended Data Fig. 9).

For 24 perturbations, we detected significant changes in the ratio of dorsal versus ventral cells (Fig. 2c). Notably, most perturbations (21 of 24) lowered the dorsal-to-ventral ratio. Perturbation of *KMT2C*, for example, led to a strong enrichment of ventral cells. For 23 perturbations, we detected changes in the abundance of at least one cell type (CMH test, false discovery rate (FDR) < 0.05) (Fig. 2c). On the other hand, six perturbations specifically targeted one cell type without affecting others, including *ADNP* (L2/3 enrichment), *POGZ* (RGC enrichment) and *SETD5* (ccv-RGC enrichment).

Among the progenitors, we identified IPC depletion as a strong, convergent effect in 12 perturbations (Fig. 2c) (for example, *CHD2*, *KAT2B* and *KMT2C*). Additionally, we observed an enrichment of v-RGCs and/or ccv-RGCs in 10 perturbations (for example, *ARID1B*, *BCL11A* and *DEAF1*) and an enrichment of INPs in 10 perturbations (for example, *ILF2*, *MED13* and *TCF20*). To validate these phenotypes, we generated individually perturbed organoids for several genes. We performed immunohistochemical analyses of IPCs and INPs at day 60, a stage when organoids present a radially organized structure including ventricular zone (VZ), subventricular zone (SVZ) and cortical plate, enabling robust examination of the progenitors[35]. Consistent with the screen data, we detected significantly decreased EOMES+ IPCs in *KMT2C* and *PHF3* perturbations

(Extended Data Fig. 11) and increased DLX2+ INPs in *KMT2C*, *MED13*, *PHF3* and *TBL1XR1* perturbations (Extended Data Fig. 12). Furthermore, we analysed specific neuronal subpopulations and found that L2/3 excitatory neurons were more impacted and depleted in the majority of perturbations (Fig. 2c). Analysis of interneurons revealed enrichment of both CGE-INs and LGE-INs in three perturbations and enrichment of only CGE-INs or LGE-INs in four perturbations. In addition, LGE PAX6+ interneurons were depleted by *PHF3* perturbation. These data indicate an interneuron subtype-specific response to ASD genetic perturbations. Beyond neuronal cell clusters, we found that astrocytes were significantly enriched in *CIC* perturbation.

To assess the consistency of the effects across replicates from different batches, we performed a *t*-test on individual enrichment and depletion effects (Extended Data Fig. 10). We observed that most effects detected at the single-cell level were also largely consistent across organoids grown from different batches, further supporting the robustness and reproducibility of our system.

Collectively, the CHOOSE system allowed us to simultaneously investigate the effects of multiple ASD genes on cell fate determination. We found that progenitors, including IPCs and INPs, as well as L2/3 excitatory neuronss, were among the most affected cell types. Furthermore, our data indicate that ASD pathology could emerge as early as the neural progenitor stage.

## Altered gene expression upon ASD gene perturbation

To further assess the molecular changes caused by each perturbation, we performed a differential gene expression analysis, comparing each

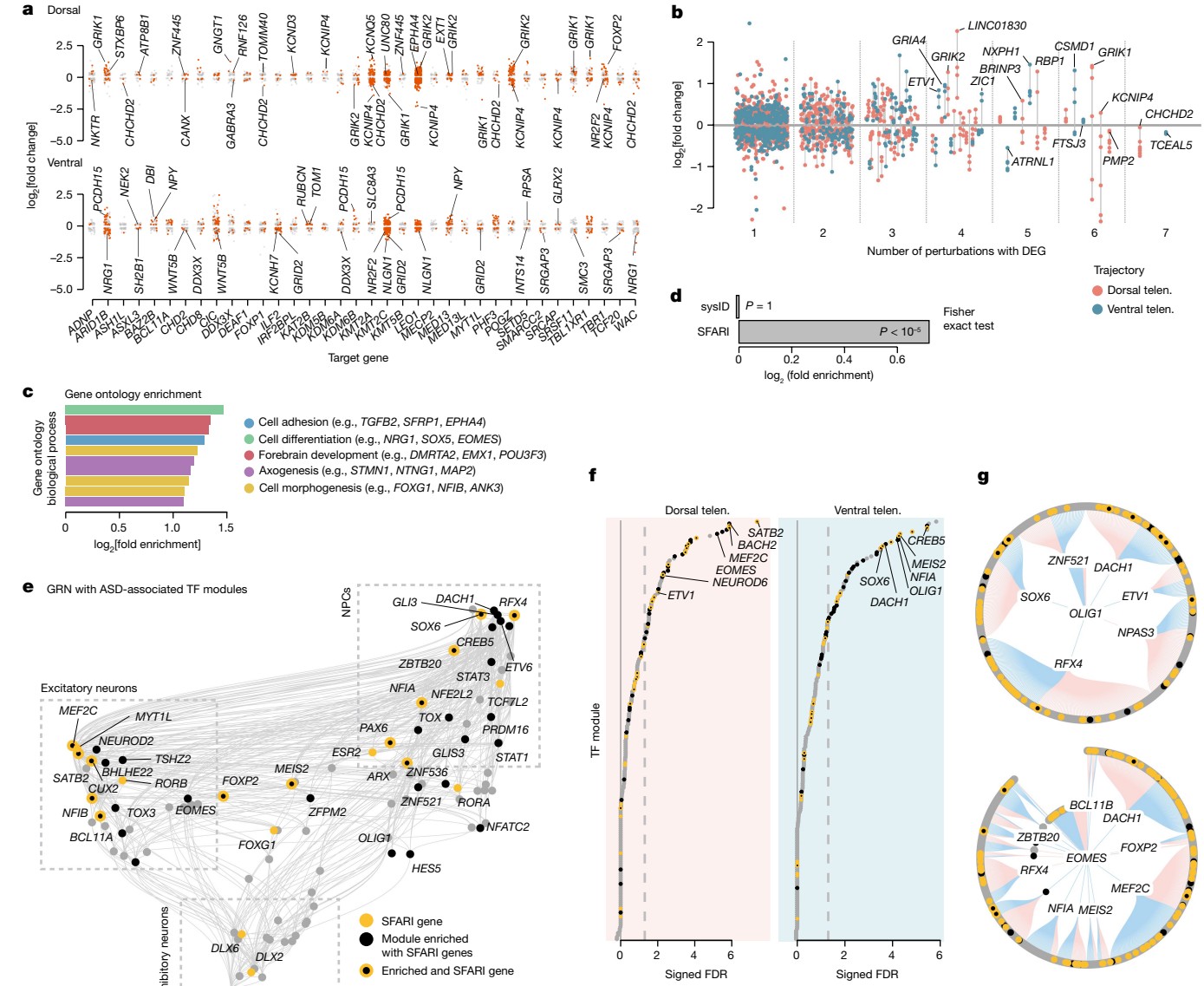

**Fig. 3 | Dysregulated gene expression and regulatory networks caused by perturbations of ASD risk genes. a**, Jitter plots showing DEGs detected in dorsal and ventral trajectories from each genetic perturbation. **b**, Jitter plot shows the frequency of DEGs detected from all perturbations separated by dorsal (orange) and ventral (blue) trajectories. Points belonging to the same gene are connected with a grey line. **c**, Top biological pathways enriched for TOP-DEGs (top 30 DEGs per perturbation) identified from all perturbations. **d**, Enrichment test of TOP-DEGs per perturbation in ID (sysID database) or ASD (SFARI database) genes. Two-sided Fisher exact test, $P = 3.8 \times 10^{-7}$. **e**, GRN of 4-month-old cerebral organoids inferred by Pando showing developmental

TF modules constructed on the basis of their coexpression and interaction strengths. ASD-associated TF modules are highlighted in yellow (SFARI genes) and/or black (regulator of SFARI genes). **f**, Lolliplots show CHOOSE DEG-enriched TF modules in dorsal and ventral trajectories. The *x* axis represents the sign of the log odds ratio multiplied by the $-\log_{10}$ FDR-corrected *P* value of a two-sided Fisher exact test (signed $-\log_{10}$ FDR). The dashed lines indicate an FDR of 0.05. **g**, Circular gene regulatory subnetwork plots show primary and secondary targets of *OLIG1* and *EOMES*. ASD-specific TF modules are highlighted in black and yellow as described above. Blue edges indicate repressive connections, and red edges indicate activating connections. NPC, neural progenitor cell.

perturbation with controls within dorsal and ventral trajectories. We detected 2,071 differentially expressed genes (DEGs) across all perturbations (Fig. 3a and Supplementary Data 2). Additionally, we could identify genes dysregulated in both dorsal and ventral populations, as well as those specifically dysregulated in one population (*KMT2C*, *LEO1* perturbation) (Extended Data Fig. 13a). We ranked DEGs by detection frequency and discovered that many genes were differentially expressed in multiple perturbations (Fig. 3b). Notably, in the dorsal populations, seven perturbations caused *CHCHD2* downregulation (Fig. 3a). *CHCHD2* encodes a mitochondrial protein, and its downregulation has been observed in neurons from the postmortem brains of patients with ASD[34]. In the ventral cell populations, the most frequently

detected DEG is the adhesion molecule gene *CSMD1* (Fig. 3a,b), which is upregulated by *ARID1B*, *CIC*, *MED13* and *PHF3* perturbations and downregulated by *LEO1* and *KMT2C* perturbations. To ensure a balanced impact of differential DEG numbers across all perturbations on downstream analyses, we selected the top 30 DEGs for each perturbation (TOP-DEGs) for gene ontology term enrichment analysis. We found that cell adhesion, cell differentiation, forebrain development and axogenesis were among the most associated biological processes (Fig. 3c and Supplementary Data 3). We also observed many perturbation-specific gene ontology terms, many of which confirm previous studies, further supporting the power of detecting complex biological phenotypes with the CHOOSE system (Extended Data Fig. 13b and Supplementary

Data 4). Some notable examples include the gene ontology terms ribosome assembly (*SETD5* perturbation)[36], mitochondrion organization (*FOXP1* perturbation)[37], lipid homoeostasis (*IRF2BPL* perturbation)[38], autophagosome maturation (*KAT2B* perturbation)[39] and cilium development (*MECP2* perturbation)[40].

## ASD-associated regulatory modules

When combining the TOP-DEGs from all perturbations, we found that they were significantly enriched in risk genes associated with ASD (SFARI database, 1,031 genes; approximately twofold enrichment, Fisher exact test $P < 10^{-5}$ against genes expressed in 5% of cells as background) (Fig. 3d and Supplementary Data 2). Notably, we did not observe enrichment in risk genes for other NDDs, such as intellectual disability (ID; sysID database, 936 primary ID genes) (Fig. 3d), suggesting that certain biological processes and regulatory programs might be specifically relevant to ASD-associated gene perturbations. To explore these potential gene regulatory 'hubs', we first generated single-cell multiome data, including single-cell transcriptome and chromatin accessibility modalities, from 4-month-old cerebral organoids (Extended Data Fig. 14a–g). Using Pando[41], we harnessed these multimodal measurements to infer a GRN of the developing telencephalon and extract sets of genes regulated by each transcription factor (TF) module, as well as positive and negative regulatory interactions between the TFs (Extended Data Fig. 14h,i and Supplementary Data 5). We visualized this GRN on the level of TFs using a uniform manifold approximation and projection (UMAP) embedding[42], which revealed distinct TF groups active in neural progenitor cells (*PAX6*, *GLI3*), inhibitory neurons (*DLX2*, *DLX6*) and excitatory neurons (*NEUROD2*, *NFIB*, *SATB2*) as well as regulatory interactions between the TFs (Fig. 3e).

To test whether regulatory subnetworks indeed exist at which ASD risk genes accumulate, we tested all TF modules for enrichment with SFARI genes. We found significant enrichment for a set of 40 TFs (adjusted Fisher test $P < 0.01$, more than twofold enrichment; for example, *EOMES*, *OLIG1*, *DLX2*) (Extended Data Fig. 14j), among which 14 TFs were encoded by ASD risk genes (for example, *NFIA*, *BCL11A*, *MEF2C*) (Fig. 3e). All TF regulatory modules enriched in SFARI genes together form an ASD-associated sub-GRN (Supplementary Data 6).

Next, we assessed the transcriptomic effect of ASD genetic perturbations in the context of the inferred GRN. We performed enrichment tests (Fisher exact test) on perturbation-induced TOP-DEGs (CHOOSE DEGs) from dorsal and ventral telencephalic cells separately. We found that, similar to ASD risk genes, CHOOSE DEGs were enriched in specific TF modules (Fig. 3f and Supplementary Data 7). In the ventral telencephalic cells, *CREB5*, *MEIS2*, *NFIA* and *OLIG1* were most strongly affected, whereas dorsal telencephalon-specific DEGs were strongly enriched in *SATB2*, *BACH2*, *MEF2C* and *EOMES* modules. Notably, some of the ASD-associated TF modules were among the most strongly enriched in CHOOSE DEGs, supporting their role in ASD-associated gene dysregulation (Fig. 3f). We finally present gene regulatory subnetworks of *OLIG1* and *EOMES*, which are both enriched in ASD risk genes and strongly affected by ASD genetic perturbations (Fig. 3g). Oligodendrocyte transcription factor 1 (OLIG1) is preferentially expressed in the ventral telencephalon and is a key regulator for interneuron and oligodendrocyte lineages. Eomesodermin (EOMES) is a key TF for the fate specification of IPCs in the dorsal telencephalon. The enrichment of *OLIG1* and *EOMES* regulomes suggests potentially vulnerable cell fate specification-related regulatory networks upon ASD genetic perturbations.

Thus, we have characterized gene expression changes for each genetic perturbation in both dorsal and ventral telencephalon and uncovered molecular programs shared between different perturbations. Leveraging GRN inference from multiomic data, we further identified ASD-associated TF modules during cortical development and critical regulatory hubs underlying the detected gene expression changes.

## Effects of *ARID1B* perturbation on v-RGCs

Among the 36 genes, we found that *ARID1B* perturbation caused one of the most significant enrichments of v-RGCs (Fig. 2c). Notably, the *OLIG1* regulatory module was also enriched in DEGs caused by *ARID1B* perturbation (Extended Data Fig. 15a). These data motivated us to further investigate how v-RGCs are affected by *ARID1B* perturbation in the screen. We used Cellrank[43] to delineate the developmental trajectories leading to different interneuron subtypes and OPC populations (Fig. 4a). We visualized the terminal fate probabilities for each cell as a circular projection, which revealed a distinct differentiation trajectory from ventral progenitors towards early OPCs (*OLIG2*, *PDGFRA*) and a branching of INPs (*DLX2*) into different inhibitory neuronal fates (*DLX5*) (Fig. 4b). We found that *ARID1B*-perturbed cells were strongly enriched in the OPC trajectory and had a higher percentage of *OLIG2*+ v-RGCs (Fig. 4c,d). This is an interesting finding given that *OLIG2* is known to regulate progenitor self-renewal at earlier developmental stages and is a master regulator for oligodendrocyte lineage specification in the ventral telencephalon[44,45]. We then analysed the fate transition probabilities of ventral progenitors and found that *ARID1B*-perturbed v-RGCs have significantly higher transition probabilities towards early OPCs than neuronal fates (Fig. 4e).

Loss-of-function mutations in *ARID1B* have been shown to cause ID and ASD[7,46]. To confirm whether our findings are relevant to human disorders, we recruited two patients with heterozygous *ARID1B* mutations. Patient 1 harbours a nucleotide duplication (c.2201dupG), resulting in an early STOP codon. Patient 2 carries a microdeletion (6q25.3del) that includes exon 8 and the downstream region of the *ARID1B* locus (Extended Data Fig. 15b,c). We established iPS cell lines from both patients and a mutation-corrected cell line for patient 1 as an isogenic control. To investigate the behaviour of v-RGCs, we used a previously published protocol that uses smoothened agonist (SAG) and inhibitor of Wnt production-2 (IWP2) to specifically guide organoids to develop ventral telencephalic tissue[47,48]. In line with our previous findings, we observed considerably increased *OLIG2*+, *DLX2*+ and *DLX2*+*OLIG2*+ cells in 40-day-old organoids from both patients compared with control organoids (Fig. 4f,g). To further explore the potential consequences of such defects in patients, we analysed the prenatal brain structure of patient 2 at two gestation stages (gestational weeks (GW) 22 and GW31) by intrauterine super-resolution magnetic resonance imaging (MRI). Three-dimensional reconstruction of the ganglionic eminence (GE), the source of ventral telencephalon progenitors, and normalization to cortical and total brain volume revealed an enlarged GE compared with multiple age-matched controls at both examined developmental stages (Fig. 4h and Extended Data Fig. 15d–h), which could be partially due to an increase in ventral progenitors. Taken together, the enrichment of v-RGCs and ccv-RGCs (Fig. 2a), the higher transition probability of v-RGCs to early OPCs, and the increased proportion of *OLIG2*+ cells in our screen and in organoids generated from two patient iPS cell lines all suggest that *ARID1B* perturbation leads to abnormal ventral progenitor expansion and aberrant cell fate specification (Fig. 4i). The enlarged volume of GE in the patient with an *ARID1B* mutation is consistent with these observations.

## Discussion

We have developed the CHOOSE system to characterize loss-of-function phenotypes of high-risk ASD genes across dozens of cell types spanning early brain developmental stages in cerebral organoids. By employing a pooled CRISPR screening system in conjunction with validation, our study provides a developmental and cell type-specific phenotypic database for ASD gene loss-of-function research. IPCs, transit-amplifying dorsal progenitors that generate neurons for all cortical layers and contribute to the evolutionary expansion of the human cortex[49,50], appeared to be particularly susceptible to ASD genetic perturbations. Among the

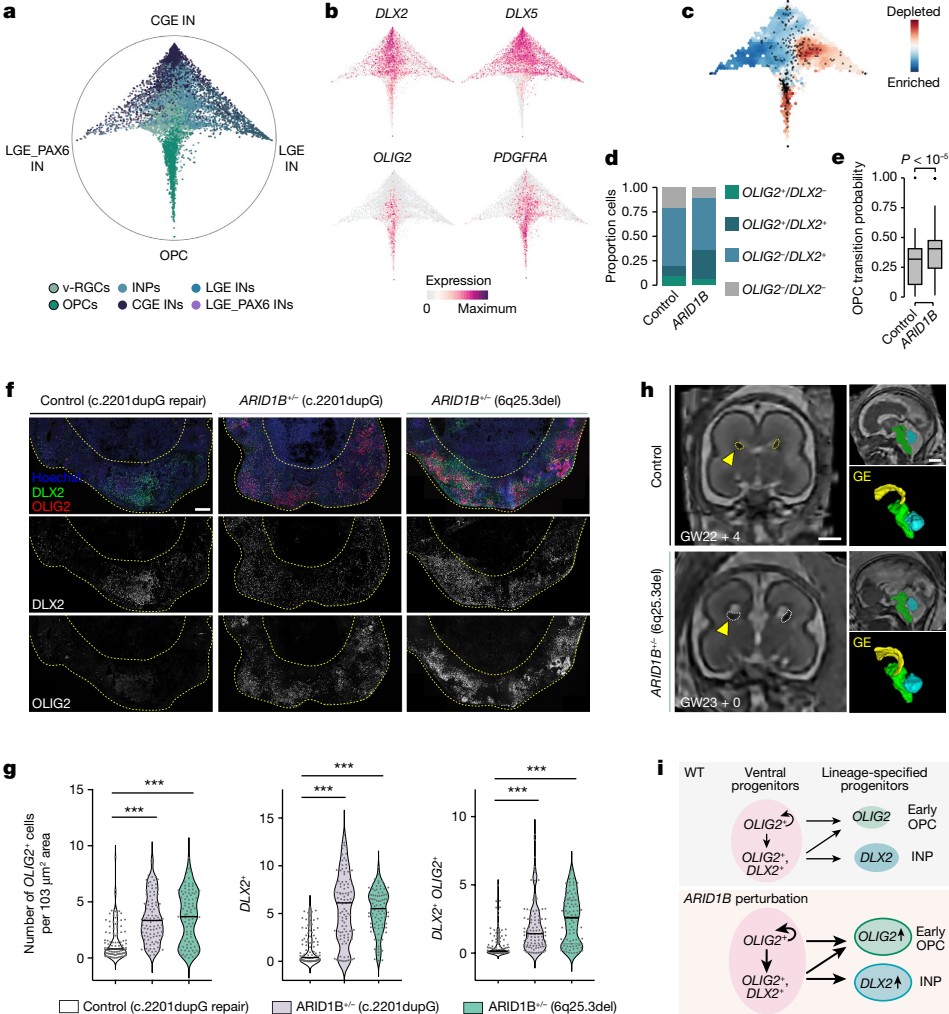

**Fig. 4 | Perturbation of *ARID1B* increases the transition of v-RGCs to early OPCs. a**, Circular projection of terminal fate probabilities shows ventral telencephalon differentiation trajectories. **b**, Trajectory branches defined by gene expressions of *DLX2* (INP), *DLX5* (IN), *OLIG2* (early OPC) and *PDGFRA* (late OPC). **c**, Differential density of cells with *ARID1B* perturbation versus control. **d**, Bar graph shows cells within the v-RGCs that are positive for *DLX2* (68.6 versus 81.5%), *OLIG2* (19.6 versus 36.2%) and both (9.8 versus 30.6%) in control versus *ARID1B* perturbation. **e**, Box plots showing transition probabilities of control (*n* = 51) and *ARID1B*-perturbed (*n* = 46) ventral progenitor cells towards OPCs. The centre lines represent the medians, the box limits show the 25–75% interquartile ranges and the whiskers indicate 1.5× the interquartile ranges. Two-sided Wilcoxon test *P* value = 7.7 × 10$^{-8}$. **f**, Immunohistochemistry for early OPCs (OLIG2) and INPs (DLX2) of day 40 ventralized brain organoids derived from control (c.220dupG repair) and two ARID1B patient iPS cells. Scale bar, 200 μm. **g**, Violin plots (all data points and median values) show numbers of cells positive for OLIG2 and/or DLX2. Control, *n* = 108 areas from 13 organoids, four batches; *ARID1B*$^{+/-}$ (c.2201dupG), *n* = 104 areas from 15 organoids, four batches; *ARID1B*$^{+/-}$ (6q25.3del), *n* = 94 areas from 15 organoids, three batches. One-way analysis of variance (ANOVA) post hoc Tukey test; ***P* < 0.001. **h**, Prenatal magnetic resonance imaging scan and 3D reconstruction of LGE and CGE (marked as GE) from age-matched controls and a patient with an *ARID1B* mutation showing enlarged GE in the patient (quantified in Extended Data Fig. 10). Scale bar, 1 cm. **i**, Diagram showing *ARID1B* perturbation-induced cellular responses of ventral progenitors. WT, wild type.

ventral telencephalon cells, we have identified strong enrichment of v-RGCs, ccv-RGCs and INPs, which are progenitors that differentiate into interneurons and oligodendrocytes[44,51]. Furthermore, our findings indicate that L2/3 excitatory neurons are more vulnerable than other neuronal populations to ASD perturbations. This aligns with the observation that ASD risk gene coexpression networks are enriched in upper-layer neurons during development and that these neurons are preferentially affected in ASD patients[6,34]. In our screen, we assessed 19 ASD genes known to be involved in epigenetic regulation. Despite their broad role in cell differentiation, perturbations of these genes impacted specific cell types and biological processes during brain development. DEG and gene ontology enrichment analyses revealed both common and distinct molecular processes impaired in different perturbations, suggesting that both convergent and divergent mechanisms contribute to ASD pathophysiology. Furthermore, we

constructed a telencephalon developmental trajectories-based GRN and identified ASD-associated regulatory modules in dorsal and ventral cell populations. The *OLIG1* module is particularly interesting as many of its downstream targets are ASD risk genes. This module was previously identified as crucial for oligodendrocyte differentiation in the developing human cortex[22], highlighting the involvement of the oligodendrocytes in ASD pathophysiology.

The combination of high-content perturbation screening and validation in a patient-specific context exemplifies the effectiveness of employing organoid systems to study NDDs. We discovered that loss of *ARID1B* leads to increased transition of ventral progenitors to early OPCs. Importantly, perturbations of three BAF complex members (*ARID1B*, *BCL11A* and *SMARRC2*) all lead to enrichment of v-RGCs, indicating the critical role of the BAF complex in regulating ventral telencephalon cell fate specification. Given the cell type-specific

expression of each BAF subunit and their involvement in NDDs[52], it would be interesting to investigate how ARID1B or other subunits regulate oligodendrocyte and interneuron specification, as well as their contribution to NDDs.

Our study has limitations. First, our system lacks certain brain cell types, such as microglia, and does not include interneurons derived from the medial ganglionic eminence. Future studies should explore the impact of ASD gene alterations on these cell types or use a system that better resembles in vivo cell-type complexity. Second, we do not know whether perturbed cells are heterozygous or homozygous for each mutant. It would be interesting to generate precisely edited cells to compare mutation-specific phenotypes. Additionally, the effects of perturbations of ASD risk genes on cell-type abundances can sometimes be transient[11,53], and therefore certain abnormalities during development may not be captured.

The ability to determine cell type-specific contributions to genetic disorders in a systematic, scalable and efficient manner will greatly enhance our understanding of disease mechanisms. As the CHOOSE system provides a robust, precisely controlled screening strategy, we anticipate that it will be widely applied beyond brain organoids to study disease-associated genes.

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

# Methods

## Stem cell and cerebral organoid culture conditions

Feeder-free hES cells or iPS cells were cultured on hES cell-qualified Matrigel (Corning, catalogue no. 354277)-coated plates with Essential8 stem cell medium supplemented with bovine serum albumin (BSA). H9 embryonic stem cells were obtained from WiCell. Cells were maintained in a 5% $CO_2$ incubator at 37 °C. All cell lines were authenticated using a short tandem repeat assay, tested for genomic integrity using single-nucleotide polymorphism (SNP) array genotyping and routinely tested negative for mycoplasma.

Cerebral organoids were generated using a previously published protocol with modifications[23]. In brief, cells were cultured to 70–80% confluent, and 16,000 live cells in 150 µl Essential8 medium supplemented with Revitacell (ThermoFisher, catalogue no. A2644501) were added to each well of a U-bottom ultralow attachment 96-well plate (Corning, catalogue no. CLS3473) to form embryoid bodies. For eCas9 induction, 4-hydroxytamoxifen (Sigma-Aldrich, catalogue no. H7904) was added on day 5 at a concentration of 0.3 µg ml$^{-1}$. Neural induction was started on day 6. Embryoid bodies were embedded in Matrigel (Corning, catalogue no. 3524234) at day 11 or 12 based on morphology check. CHIR99021 (Merck, catalogue no. 361571) at 3 µM was added from day 13 to day 16, and medium was switched to improved differentiation medium supplemented with B27 minus vitamin A (IDM-A) at day 14. On day 25, medium was switched to improved differentiation medium supplemented with B27 plus vitamin A (IDM+A); 1% dissolved Matrigel was added to the medium from day 40 to day 90. From day 60 to day 70, medium was gradually switched to Brainphys neuronal medium (Stemcell Technologies, catalogue no. 05790) and supplemented with brain-derived neurotrophic factor (BDNF) (20 ng ml$^{-1}$; Stemcell Technologies, catalogue no. 78005.3), glial cell line-derived neurotrophic factor (GDNF) (20 mg ml$^{-1}$; Stemcell Technologies, catalogue no. 78058.3) and bucladesine sodium (1 mM; MedChemExpress, catalogue no. HY-B0764)[24]. For ventralized organoids, we followed a previously published protocol[47]. Embryoid bodies were not embedded, and patterning factors, including 100 nm SAG (Merck-Millipore, catalogue no. US1566660) and 2.5 µM IWP2 (Sigma-Aldrich, catalogue no. IO536), were added from day 5 to day 11.

## CHOOSE screen

**sgRNA selection and cloning.** The top four sgRNAs were first selected on the basis of predictions using multilayered Vienna Bioactivity CRISPR (VBC) score[54] and then subjected to the reporter assay (below) to test editing efficiency. sgRNAs were cloned into the gRNA reporter assay lentivirus construct (containing the dual sgRNA cassette: U6-sgRNA1-H1-sgRNA2) using the GeCKO cloning protocol[55]. The two sgRNAs were cloned using type IIS class restriction enzymes FastDigest BpiI (ThermoFisher, catalogue no. FD1014) and Esp3I (ThermoFisher, catalogue no. FD0454) separately and verified using Sanger sequencing. All gRNAs used for this study can be found in Supplementary Table 1.

**sgRNA reporter assay.** A construct containing dTomato-2A-gRNA target array-TagBFP under the RSV promoter was assembled using Gibson assembly. The construct was packaged into retrovirus using the Platinum-E retroviral packaging cell line via the calcium phosphate-based transfection method. Virus-containing supernatant (Dulbecco's modified Eagle's medium (DMEM), 10% fetal bovine serum (FBS), 2 mM L-glutamine, 100 U ml$^{-1}$ penicillin and 0.1 mg ml$^{-1}$ streptomycin) was collected for up to 72 h, filtered through a 0.45-µm filter and then stored on ice. Retroviruses were then used to infect NIH-3T3 cells, and dTomato-positive cells were sorted using flow cytometry into single cells to establish reporter cell lines. To deliver sgRNAs, the lentiviral construct containing the dual gRNA cassette and the spleen focus-forming virus (SFFV) promoter driving eCas9 were packaged using HEK293 cells to produce lentivirus. The reporter 3T3 cell lines generated above were cultured in six-well plates and infected with lentivirus containing dual sgRNA cassette targeting each gene individually. BFP fluorescence was measured at 7, 14 and 20 days postinfection. Fluorescent changes at 20 days postinfection were used to evaluate the gRNA editing efficiency. In total, 98 dual sgRNA cassettes were tested for 36 genes.

**Generation of barcoded CHOOSE lentivirus, hES cell infection and embryoid body generation.** The CHOOSE lentiviral vector was constructed based on a previously published lentiviral vector that carries a CAG driving ERT2-Cre-ERT2-P2A-EGFP-P2A-puro cassette[12]. A multicloning site including NheI and SgsI recognition sequences was introduced to the 3′ LTR of the lentivirus backbone according to the CROP-seq vector design[17]. Then, the original U6 sgRNA expression cassette was removed; instead, the dual sgRNA (U6-sgRNA1-H1-sgRNA2) cassette was introduced to the 3′ LTR cloning site. To generate a barcoded library, the following primers were used to individually amplify (8–10 cycles, monitored using a quantitative polymerase chain reaction (qPCR) machine, stopped when reaching to logarithmic phase) each dual sgRNA cassette from the lentiviral construct used in the reporter assay while introducing a 15 base pair barcode.

FW primer: 5′-tcgaccgctagcagggcctatttcccatga-3′.

RV primer: 5′-cagtagggcgcgcc<u>NVDNHBNVDNHBNVD</u>ccggcg aaccatgatcaaa-3′.

Equal molar amounts of amplicons for the ASD library (36 paired sgRNAs targeting ASD genes) or control library (a paired non-targeting control gRNA) were pooled. Amplicons and lentiviral backbone were then digested with FastDigest NheI (ThermoFisher, catalogue no. FD0973) and FastDigest SgsI (ThermoFisher, catalogue no. FD1894) and gel purified. Ligation was performed using T4 DNA ligase (ThermoFisher, catalogue no. EL0011) and cleaned up by phenol-chloroform extraction. In total, 90 ng of ASD library plasmids and 30 ng of control library plasmids were used for electroporation of MegaX DH10B T1R Electrocomp Cells (ThermoFisher, catalogue no. C640003) following the manufacturer's guide. Bacteria were plated on lithium borate medium plates containing ampicillin. Dilutions were performed to calculate the complexity; $2.6 \times 10^7$ colonies were obtained for the ASD library, and $0.5 \times 10^7$ colonies were obtained for the control library. Lentiviruses were packaged using HEK293T cells, and infection of hES cells was performed as before[12]. Infection rate was controlled to be lower than 5% to prevent multiple infections[18]; $6.6 \times 10^5$ ASD library cells and $2.3 \times 10^5$ control library cells positive for GFP were sorted by flow cytometry. Cells were recovered and passaged two times in 10 cm dishes to maintain maximum complexity. Cells were mixed with a ratio of 96:4 (ASD:control) and then used to make embryoid bodies. For individual gene validations, lentivirus carrying a dual gRNA cassette only targeting one gene was packaged and used to infect the eCas9-inducible cell line. Cells were then collected by FACS and used to make embryoid bodies. Organoids were cultured using the conditions described above.

**Cerebral organoid tissue dissociation, FACS and scRNA-seq.** For each library, three to seven organoids at 4 months were pooled, washed twice in Dulbecco's phosphate-buffered saline (DPBS)$^{-/-}$ and dissociated using the gentleMACS dissociator in trypsin–accutase (1×) solution with TURBO DNase (2 µl ml$^{-1}$; ThermoFisher, catalogue no. AM2238). After dissociation, DPBS$^{-/-}$ supplemented with 10% FBS (DPBS−10% FBS) was gradually added to stop the reaction. Samples were then centrifuged at 400$g$ for 5 min at 4 °C, and the supernatant was aspirated without touching the pellet. The pellet was then resuspended in an additional 1–2 ml of DPBS−10% FBS and then, filtered through a 70 µm strainer and FACS tubes. Cells were then stained with viability dye DRAQ7 (Biostatus; DR70250, 0.3 mM). Target live cells were sorted with a BD FACSAria III

on Alexa 700 filter with low pressure (100 μm nozzle) and collected in DPBS–10% FBS at 4 °C. Cells were then centrifuged and resuspended in DPBS–10% FBS to achieve a target concentration of 450–1,000 cells per microliter. Samples with more than 85% viability were processed. For each library, 16,000 cells were loaded onto a 10× chromium controller to target a recovery of 10,000 cells. Libraries using the Chromium Single Cell 3′ Reagent Kits (v.3.1) were prepared following the 10× user guide. Libraries were sequenced on a Novaseq S2 or S4 flow cell with a target of 25,000 paired-end reads per cell.

**Custom genomic reference.** Each cell expresses eCas9 from a genomic locus (AAVS1) and a polyadenylated dual sgRNA cassette, which is delivered by lentivirus and integrated into the genome. To cover these extrinsic elements, we built a custom genomic reference for mapping 10× single-cell data by amending the GRCh38 human reference. As the individual gRNA sequences differed, we masked them by Ns so as not to interfere with mapping (individual gRNA information is distinguished in a separate counting pipeline). The sequences added covered the genomic loci of *AAVS1* with eCas9-dTomato-WPRE-SV40 and the masked lentiviral construct.

## Emulsion PCR and target amplification

Emulsion polymerase chain reaction (PCR) was used to recover gRNA and UCB sequences from plasmid libraries, genomic DNA extracted from lentivirus-infected hES cells and cells sorted from CHOOSE mosaic organoids as well as 10× single-cell complementary DNA libraries to reduce PCR bias and to prevent the generation of chimeric PCR products[56,57]. AmpliTaq Gold 360 master mix (ThermoFisher, catalogue no. 4398876) was used for all PCR reactions. Emulsion PCR was performed using the Micellula DNA Emulsion & Purification Kit (EURX, catalogue no. E3600) according to the manufacturer's guide. For target amplification from 10× single-cell libraries, heminested emulsion PCRs were performed using the following primers:

First PCR: forward primer (FW): 5′-gcagacaaatggctgaacgctgacg-3′, reverse primer (RV): 5′-ccctacacgacgctcttccgatct-3′; second PCR: FW: 5′-ggagttcagacgtgtgctcttccgatcttgggaatcttataagttctgtatgagacc actctttcc-3′, RV: 5′-ccctacacgacgctcttccgatct-3′.

Amplicons were then indexed with unique NEB dual indexing primers, and amplifications were monitored in a qPCR machine and stopped when reaching the logarithmic phase. Amplicons were sequenced using the Illumina Nextseq2000 or Novaseq6000 system. All primers used can also be found in Supplementary Table 1.

## gRNA and UCB recovery and analyses

gRNA sequences were extracted by cutting 5′- and 3′-flanking regions with cutadapt (10% error rate, 1–3 nucleotide (nt) overlap, no indels)[58]. Sequences were filtered to be between 15 and 21 nt long. The corrected cell barcode (CBC) and the unique molecular identifier (UMI) of each read were derived via the 10× Genomics Cell Ranger 6.0.1 alignment[59]. Only reads with a corresponding gene expression (GEX) cell were accepted. Reads and target sequences were joined by allowing partial overlaps and hamming distances of two. Reads are counted towards unique CBC–UMI–gRNA combinations. A read count cutoff of 1% of the median read count of the UMI with the highest reads count per cell was applied. Cells with only one gRNA and more than one read were kept. In addition, within unique CBC–UMI combinations, only gRNA with more than 20% of the maximal read count of that group was kept. After read filtering, UMIs were counted for each CBC–gRNA combination. If more than one gRNA was found within a cell, only the gRNAs with equal UMI count compared with the maximum UMI count were kept. Only one-to-one combinations were considered further. Analogous to gRNA extraction, UCB was extracted with at least 6 nt overlap to the flanks. Sequences with 12 nt length were selected and had to follow the synthesis pattern. Further processing was done analogous to gRNA.

## Preprocessing of single-cell transcriptomics data

We first aligned reads to the above defined custom genomic reference with Cell Ranger 6.0 (10x Genomics) using pre-mRNA gene models and default parameters to produce the cell by gene UMI count matrix. UMI counts were then analysed in R using the Seurat v.4 (ref. 60). We first filtered features detected in a minimum of three cells. Next, we filtered high-quality cells based on the number of genes detected (minimum 1,000, maximum 8,000), removing cells with high mitochondrial (less than 15%) or ribosomal (less than 20%) messenger RNA content. Thereafter, expression matrices of high-quality cells were normalized (LogNormalize) and scaled to a total expression of 10,000 UMIs for each cell. Principal component analysis (PCA) was performed based on the z-scaled expression of the 2,000 most variable features (FindVariableFeatures()).

## Integration and annotation of single-cell transcriptomics data

To annotate the dataset, we first extracted cells with control gRNAs and merged them with cells from uninduced organoids (35,203 cells). We integrated these unperturbed cells across libraries using Harmony[61] with default parameters. Using the integrated space, we clustered the dataset at a resolution of one using the Louvain algorithm[62] and annotated the clusters as dorsal and ventral telencephalons based on marker gene expression. We then split both trajectories and clustered again with a resolution of two to annotate the cell types more finely. This annotation of unperturbed cells was used to perform a label transfer onto the full dataset with perturbed cells using Seurat. The full CHOOSE dataset was further filtered for cells for which gRNAs were detected and integrated across libraries using the Seurat anchoring method. The integrated count matrix was log-normalized and scaled before computing a PCA. To visualize the dataset, the first 20 principal components were used to compute a UMAP embedding.

## Assessment target gene expression in organoid and primary cell types

To assess how the target genes in our screen were expressed in organoid and primary tissue, we obtained gene expression data from cell clusters in the developing human brain[32] from https://storage.googleapis.com/ linnarsson-lab-human/HumanFetalBrainPool.h5. For both the primary data and our organoid dataset, we summarized log-normalized expression for each cell type ('CellClass' in the primary dataset) by computing the arithmetic mean. We visualized the expression of CHOOSE target genes with a heat map as displayed in Extended Data Fig. 6l,m.

## RNA velocity

To obtain count matrices for spliced and unspliced transcripts, we used kallisto (v.0.46.2)[63] through the command line tool loompy from fastq from the python package loompy (v.3.0.7; https://linnarsson-lab.org/loompy/). Using scVelo (v.0.2.4)[31], moments were computed based on the first 20 principal components using the function scvelo.pp.moments() with n_neighbors = 30. RNA velocity was subsequently calculated using the function scvelo.tl.velocity() (mode = 'stochastic'), and a velocity graph was constructed using scvelo.tl.velocity_graph(). To obtain a pseudotemporal ordering describing the two differentiation trajectories, we first removed clusters annotated as cycling cells (MKI67+) and astrocytes (S100B+) from the dataset. We then calculated a pseudotime based on the velocity graph using the function scv.tl.velocity_pseudotime() for both trajectories separately.

## Differential gRNA representation analysis

To test whether perturbations affected fitness or proliferation capacity of cells, we compared gRNA representation in eCas9-induced (n = 14 pools of organoids from three batches) versus uninduced (n = 8 pools of organoids from two batches) samples. For each pool of organoids, we computed the fractions of cells with each gRNA. We then computed

the average fold change of detection percentage between induced and uninduced samples and performed a two-sided $t$-test comparing both distributions. Multiple-testing correction on the resulting $P$ values was performed using the Benjamini–Hochberg method.

### Differential abundance testing

To assess how the perturbation of ASD risk genes changes abundances of different organoid cell populations, we tested for enrichment of each gRNA in each annotated cell state versus the control. To control for confounding effects through differential gRNA sampling in different libraries, we used a CMH test stratified by library. Multiple-testing correction was performed using the Benjamini–Hochberg method, and a significance threshold of 0.05 was applied to the resulting FDR. Enrichment effects were plotted using the signed $-\log_{10}$ FDR: that is, the sign of the log odds ratio (effect size) multiplied by the $-\log_{10}$ FDR-corrected $P$ value. To further assess the variability of the differential abundance effects across independent pools of organoids, we computed cell-type fold enrichment for each organoids pool and gRNA. For this, we used 14 scRNA-seq libraries obtained from independent pools of organoids as replicates from three batches. Two batches (11 replicates) used non-targeting gRNA as a control, and a third batch (three replicates) used eCas9-uninduced cells as an alternative control. We additionally computed a background distribution of enrichment effects from randomly permuted gRNA labels. We then performed a $t$-test for each perturbation and cell type against this background distribution.

### Local cell compositional enrichment test

To visualize the compositional changes induced by the genetic perturbations at a finer resolution, we used a method outlined in Nikolova et al.[64] In brief, a $k$-nearest neighbour (kNN) graph ($k = 200$) of cells was constructed on the basis of Euclidean distance on the PCA-reduced CCA (canonical-correlation analysis) space. Next, a CMH test stratified by library was performed on the neighbourhood of each cell, comparing frequencies of the gRNA or gRNA pool and the pool of control gRNAs within and outside of the neighbourhood. The resulting neighbourhood enrichment score of each cell was defined as signed $-\log(P)$, where the sign was determined by the sign of the log-transformed odds ratio. A random walk with restart procedure was then applied to smooth the neighbourhood enrichment score of each cell. The smoothened enrichment scores were visualized on the UMAP embedding using the ggplot2 (ref. 65) function stat_summary_hex() (bins = 50).

### Differential expression analysis

To investigate the transcriptomic changes caused by each perturbation, we performed differential expression analysis based on logistic regression. We used the Seurat function FindMarkers() (test.use = 'LR') to find DEGs for each gRNA label versus control. Tests were performed on log-normalized transcript counts $Y$ while treating library, cell_type and n_UMI as covariates in the model:

$$Y_i \approx \text{n\_UMI} + \text{library} + \text{cell\_type} + \text{condition}.$$

Testing within each developmental trajectory was performed by omitting the cell_type covariate. Multiple-testing correction was performed using the Benjamini–Hochberg method, and a significance threshold of 0.05 was applied to the resulting FDR to obtain a set of DEGs (CHOOSE DEGs). We further selected top 30 DEGs on the basis of absolute fold change for each gRNA (TOP-DEGs).

### DEG enrichment analysis

To assess the biological processes in which the detected DEGs were involved, we performed gene ontology enrichment across all TOP-DEGs globally as well as using all detected DEGs for each target gene in excitatory and inhibitory neuron trajectories separately. As a background

gene set, we used all genes expressed in more than 5% of cells in our dataset. To perform gene ontology analysis, we used the function 'enrichGO' from the R package clusterProfiler[66] with 'pAdjustMethod = 'fdr''. We filtered the results using a significance threshold of FDR < 0.01. To test whether the set of TOP-DEGs was enriched for ASD-associated genes, we first obtained a list of risk genes from SFARI (https://gene.sfari.org/database/gene-scoring/, 11 April 2021). We then tested the enrichment using a Fisher exact test with all genes expressed in more than 5% of cells in our dataset as the background. To assess the specificity of this enrichment, we obtained a list of ID risk genes from sysID (936 primary ID genes, https://sysndd.dbmr.unibe.ch, 17 March 2022) and tested for enrichment among TOP-DEGs in the same way.

### Processing of single-cell multiome data and GRN inference

Initial transcript count and peak accessibility matrices for the multiome data were obtained from sequencing reads with Cell Ranger Arc and further processed using the Seurat (v.4.0.1) and Signac (v.1.4.0)[67] R packages. Peaks were called from the fragment file using MACS2 (v.2.2.6)[68] and combined in a common peak set before merging. Transcript counts were log-normalized, and peak counts were normalized using term frequency–inverse document frequency normalization. To assess the cell composition of the multiome data, integration with the CHOOSE scRNA-seq data was performed using Seurat (FindIntegrationAnchors() -> IntegrateData()) with default parameters. As a preprocessing step to GRN inference with Pando[41], chromatin accessibility data were first coarse grained to a high-resolution cluster level. For this, control cells from the CHOOSE dataset were combined with the multiome dataset, and Louvain clustering was performed at a resolution of 20 based on the first 20 principal components calculated from the 2,000 most variable features (RNA). For each cluster, peak accessibility was summarized by computing the arithmetic mean from binarized peak counts so that each cell in the cluster was represented by the detection probability vector of each peak. To constrain the set of peaks considered by Pando, we used the union of PhastCons conserved elements[69] from an alignment of 30 mammals (obtained from https://genome.ucsc.edu/) and candidate $cis$-regulatory elements derived from the ENCODE project[70] (initiate_grn()). In these regions, we scanned for TF motifs (find_motifs()) based on the motif database shipped with Pando, which was compiled from motifs derived from JASPAR and CIS-BP. Based on motif matches, cell-level log-normalized transcript counts and cluster-level peak accessibilities, we then inferred the GRN using the Pando function infer_grn() (peak_to_gene_method = 'GREAT', upstream = 100,000, downstream = 100,000) for the 5,000 most variable features. Here, genes were associated with candidate regulatory regions in a 100,000 radius around the gene body using the method proposed by GREAT[71]. From the model coefficients returned by Pando, TF modules were constructed using the function find_modules() (P_thresh = 0.05, rsq_thresh = 0.1, nvar_thresh = 10, min_genes_per_module = 5). To visualize subnetworks centred around one TF, we computed the shortest path from the TF to every gene in the GRN graph. If there were multiple shortest paths, we retained the one with the lowest average $P$ value. The resulting graph was visualized with the R package ggraph (https://github.com/thomasp85/ggraph) using the circular tree layout.

### Enrichment testing for TF modules

To find subnetworks of the GRN at which ASD-associated genes accumulate, we first obtained a list of ASD risk genes from SFARI (https://gene.sfari.org/database/gene-scoring/). For all genes included in SFARI (1,031 genes), we tested for enrichment in TF modules using a Fisher exact test. All genes expressed in more than 5% of cells in our dataset (12,079 genes) were treated as the background. Fisher test $P$ values were corrected for multiple testing using the Benjamini–Hochberg method, and significant enrichment was defined as FDR < 0.01 and more than twofold enrichment (odds ratio). To assess which TF modules were most affected by genetic perturbations of ASD-associated

genes, we similarly used a Fisher exact test. For the set of TOP-DEGs, we tested for enrichment in any of the inferred TF modules. Here, all genes included in the GRN (5,000 most variable features) were treated as the background.

## Cell rank analysis

To better understand the differentiation trajectories leading up to inhibitory neuron populations, we used CellRank[43] to compute transition probabilities into each terminal fate based on the previously computed velocity pseudotime. First, the clusters with the highest pseudotime for each terminal cell state were annotated as terminal states. We then constructed a Palantir kernel[72] (PalantirKernel()) based on velocity pseudotime and used Generalized Perron Cluster Cluster Analysis[73] (GPCCA()) to compute a terminal fate probability matrix (compute_absorption_probabilities()). All cell rank functions were run with default parameters. Fate probabilities for each cell were visualized using a circular projection[74]. In brief, we evenly spaced terminal states around a circle and assigned each state an angle $t$. We then computed two-dimensional coordinates ($x_i$, $y_i$) from the $F \in R^{N \times n_t}$ transition probability matrix for $N$ cells and $n_t$ terminal states as

$$x_i = \sum_t f_{it} \cos \alpha_t$$

$$y_i = \sum_t f_{it} \sin \alpha_t.$$

To visualize enrichment of perturbed cells in this space, we used the method outlined in Nikolova et al.[64]. Here, the kNN graph ($k = 100$) was computed using euclidean distances in fate probability space, and enrichment scores were visualized on the circular projection. Otherwise, the method was performed as described above.

## Immunofluorescence

Organoid tissues were fixed in paraformaldehyde at 4 °C overnight followed by washing in PBS three times for 10 min. Tissues were then allowed to sink in 30% sucrose overnight, followed by embedding in O.C.T. compound (Sakura, catalogue no. 4583). Tissues were frozen on dry ice and cryosectioned at 20 μm. For staining, sections were first blocked and permeabilized in 0.1% Triton X-100 in PBS (0.1% PBTx) with 4% normal donkey serum. Sections were then stained with primary and secondary antibodies diluted in 0.1% PBTx with 4% normal donkey serum. Sections were washed in PBS three times for 10 min after each antibody staining and mounted in DAKO fluorescent mounting medium (Agilent Technologies, catalogue no. S3023). The following antibodies were used in this study: DLX2 (Santa Cruz, catalogue no. SC393879, 1:100); OLIG2 (Abcam, catalogue no. ab109186, 1:100); SOX2 (R&D, catalogue no. MAB2018, 1:500); FOXG1 (Abcam, catalogue no. ab18259, 1:200); EOMES (R&D, catalogue no. AF6166, 1:200); ARID1B (Cell Signaling, catalogue no. 92964, 1:100); ADNP (ThermoFisher, catalogue no. 702911, 1:250); BCL11A (Abcam, catalogue no. 191401, 1:250); PHF3 (Sigma, catalogue no. HPA024678, 1:250); SMARCC2 (ThermoFisher, catalogue no. PA5-54351, 1:250); KMT2C (Sigma, catalogue no. HPA074736, 1:250); Alexa 488, 568 and 647 conjugated secondary anti-bodies (ThermoFisher, 1:250); and Hoechst (ThermoFisher, catalogue no. H3569, 1:10,000).

## Microscopy, image processing and quantification

Tissue sections were imaged using an Olympus IX3 Series inverted microscope equipped with a dual-camera Yokogawa W1 spinning disk. Images were acquired with 10× 0.75 (air) working distance (WD) 0.6 mm or 40× 0.75 (air) WD 0.5 mm objectives and produced by the Cellsense software.

For DLX2 and OLIG2 quantification in Fig. 4, images were processed and quantified using Fiji. Based on the size of the tissue, 5–12 regions from each organoid were selected using the Hoechst channel. In total, $n = 108$ areas (13 organoids from four batches) from the *ARID1B* control group (c.2201dupG repair), $n = 104$ areas (15 organoids from four batches) from the *ARID1B*[+/−] (c.2201dupG) group and $n = 94$ areas (15 organoids from three batches) from the *ARID1B*[+/−] (6q25.3del) group are collected and subjected to an automatic segmentation using a Fiji macro. Both DLX2 and OLIG2 channels are used to define the cell body area, followed by the intensity measurement. Area mean intensity was used for setting up the threshold. For protein expression quantification in Extended Data Fig. 2, organoids with individual gene perturbations costained for each gene were processed and quantified using Fiji. Five to fourteen cortical plate regions were analysed per gene. Areas containing both uninduced (dTomato−) as well as induced (dTomato+) cells were selected and subjected to an automated segmentation using a Fiji macro. The Hoechst channel is used to define the cell body area, followed by intensity measurement. Detected cells were separated into wild-type and perturbed cells by setting up a threshold of mean intensity in the dTomato channel. Additionally, KMT2C protein expression was compared between wild-type (dTomato−) and mutant (dTomato+) VZ area. VZs were individually outlined, and mean dTomato as well as KMT2C intensities were measured. For IPC abundance analysis, organoids with individual gene perturbations were costained for EOMES. Mutant columns expressing dTomato were individually segmented, and EOMES+ cells were identified by setting a threshold for EOMES intensity. The number of EOMES+ cells was normalized to the total number of cells. Percentages of EOMES+ cells were compared between individual gene perturbations and non-targeting gRNA control groups. For INP abundance analysis, organoids were costained with DLX2. A Fiji macro for automated segmentation was used to identify DLX2+ cells throughout the entire tissue. Areas containing multiple rosettes from each organoid were collected for quantification. The number of DLX2+ cells was normalized to the tissue area and compared between individual gene perturbations and non-targeting gRNA control groups.

## Patient sample collection

The study was approved by the local ethics committee of the Medical University of Vienna. Study inclusion criteria were as follows: (1) mutation in the *ARID1B* gene proven by whole-exome sequencing, (2) age between 0 and 18 years old, (3) continuous follow-up at the Vienna General Hospital and (4) availability of fetal brain MRI data. After informed consent, 10 ml of blood was collected from two selected patients for iPS cell reprogramming.

## Reprogramming of PBMCs into iPS cells

iPS cells were generated from peripheral blood mononuclear cells (PBMCs) isolated from patient blood samples as previously described[75]. In brief, 10 ml blood was collected in sodium citrate collection tubes. PBMCs were isolated via a Ficoll–Paque density gradient, and erythroblasts were expanded for 9 days. Erythroblast-enriched populations were infected with Sendai Vectors expressing human OCT3/4, SOX2, KLF4 and cMYC (CytoTune; Life Technologies, A1377801). Three days after infection, cells were switched to mouse embryonic fibroblast feeder layers. Five days after infection, the medium was changed to iPS cell medium (KoSR + FGF2). Ten to 21 days after infection, the transduced cells began to form colonies that exhibited iPS cell morphology. iPS cell colonies were picked and passaged every 5–7 days after transfer to the mTeSR culture system (Stemcell Technologies).

## Generation of isogenic control cell line for patient 1

Isogenic control cell lines for patient 1 were generated using CRISPR–Cas9. *Streptococcus pyogenes* Cas9 protein with two nuclear localization signals was purified as previously described[76]. gRNA transcription was performed with the HiScribe T7 High Yield RNA Synthesis Kit (NEB) according to the manufacturer's protocol, and gRNAs were purified

via phenol:chloroform:isoamyl alcohol (25:24:1; Applichem) extraction followed by ethanol precipitation. The homology-directed repair (HDR) template (custom single-stranded oligodeoxynucleotides; Integrated DNA Technologies) was designed to span 100 base pairs up- and downstream of the mutation site. iPS cells had been grown in mTeSR for 14 passages before the procedure. For generation of isogenic control cell lines, cells were washed with DPBS$^{-/-}$ and incubated for 5 min at 37 °C with 1 ml of accutase solution (Sigma-Aldrich, A6964-500ML). The plate was gently tapped to detach cells, and cells were gently pipetted to generate a single-cell suspension, pelleted by spinning at 200$g$ for 3 min and counted using Trypan Blue solution (ThermoFisher Scientific). For nucleofection, $1.0 \times 10^6$ cells were spun down and resuspended in Buffer R of the Neon Transfection System (ThermoFisher Scientific) at a concentration of $2 \times 10^7$ cells per millilitre. Twelve nanograms of sgRNA and 5 ng of Cas9 protein were combined in resuspension buffer to form the Cas9–sgRNA ribonucleoprotein complex. The reaction was mixed and incubated at 37 °C for 5 min. Five microliters of the HDR template (100 µM) were added to the Cas9–sgRNA ribonucleoprotein complex and combined with the cell suspension. Electroporations were performed using a Neon Transfection System (ThermoFisher Scientific) with 100 µl Neon Pipette Tips using the embryonic stem cells electroporation protocol (1,400 V, 10 ms, three pulses). Cells were seeded in one matrigel-coated well of a six-well plate in mTeSR. After a recovery period of 3 days, a single-cell suspension was generated, and cells were split into another well of a six-well plate for banking and sparsely into two 10-cm dishes for colony formation from single cells. After colony growth for 1 week, individual colonies were picked and seeded each into one well of a 96-well plate. After colony expansion, gDNA was extracted using DNA QuickExtract Solution (Lucigen), followed by PCR and Sanger sequencing to determine efficient repair of the mutation.

## Fetal MRI and 3D reconstruction

Women with singleton pregnancies undergoing fetal MRI at a tertiary care centre from January 2016 to December 2021 were retrospectively reviewed. This study was approved by the institutional ethics board, and all examinations were clinically indicated. A retrospective review of patient records was performed, and a patient with a positive genetic testing report for *ARID1B* mutation was selected. The participant was included in further analysis, and the gestational age (given in gestational weeks and days postmenstruation) was determined by first-trimester ultrasound. High-quality super-resolution reconstruction was obtained[13]. Age-matched control subjects were identified and included if they presented an absence of confounding comorbidities, including structural cerebral or cardiac anomalies or fetal growth restriction.

Fetal MRI scans were conducted using 1.5-T (Philips Ingenia/Intera) and 3-T magnets (Philips Achieva). The mother was examined in a supine position or if necessary, left recumbent to achieve sufficient imaging quality. The examinations were performed within 45 min, neither sedation nor MRI contrast medium was applied, and both the fetal head and body were imaged. Fetal brain imaging included T2-weighted sequences in three orthogonal planes (slice thickness = 3–4 mm, echo time = 140 ms, field of view = 230 mm) of the fetal head. Postprocessing was conducted as previously described[77]. Superresolution imaging was generated using a volumetric superresolution algorithm[77]. The resulting superresolution data were quality assessed, and only cases that met high-quality standards (score of less than or equal to two of five) were included in the analysis. Atlas-based segmentation was performed for the fetal cortex and total brain volume by nonrigid mapping of a publicly available spatiotemporal, anatomical fetal brain atlas for each investigated case[77,78]. Segmentation of the GE was performed manually using the open-source application ITK-SNAP[79]. To delineate the T2-weighted hypointense GE, histological fetal atlases by Bayer and Altman[80,81] were used as a reference guide. Volumetric

data were generated and calculations for the GE were made based on the investigated gestational ages.

## Statistics

Information on the statistical analyses used is described in each method section. No statistical methods were used to predetermine sample size unless specified. No blinding and randomization were used unless specified.

## Reporting summary

Further information on research design is available in the Nature Portfolio Reporting Summary linked to this article.

## Data availability

Raw sequencing datasets were deposited into ArrayExpress with the following accession codes: single-cell RNA sequencing and associated amplicon (E-MTAB-13148), bulk genomic DNA-derived amplicon (E-MTAB-13140) and single-cell mutiomics (E-MTAB-13144). *ARID1B* cell line genotype data were deposited into the European Genome-Phenome Archive (EGAS00001007381). Processed Seurat objects were deposited into Zenodo (https://zenodo.org/record/7083558).

## Code availability

The Pando R package is available on GitHub (https://github.com/quadbiolab/Pando). Other custom code used in the analyses has been deposited on GitHub (https://github.com/quadbiolab/ASD_CHOOSE).

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

**Acknowledgements** We thank the patients and their families for participating in this study. We thank all members of the laboratory of J.A.K. for support and discussions; F. Bonnay, E. Chatzidaki, O. L. Eichmüller, R. Najm and J. Sidhaye for comments on the manuscript; M. Nezhyba and T. Lendl from the IMP/IMBA Biooptics Facility for technical support; A. Vogt, F. Drochter and T. Grentzinger from the VBCF NGS facility for single-cell RNA sequencing library preparation; the IMBA Stem Cell Core facility for generation of induced pluripotent stem cell lines; and M. Balmana Esteban and the Christoph Bock group (CeMM) for the help with single-cell multiomics sequencing. Work in the laboratory of J.A.K. is supported by SFARI (pilot award no. 724430); the Austrian Federal Ministry of Education, Science and Research; the Austrian Academy of Sciences; the City of Vienna; the Austrian Science Fund (FWF) Special Research Programme (grant no. F 7804-B) and two Stand-Alone grants (grant no. P 35680 and no. P 35369); and a European Research Council (ERC) Advanced Grant under the European Union's Horizon 2020 programs (grant no. 695642 and no. 874769). Work in the laboratory of B.T. is supported by the European Research Council (organomics and braintime (to B.T.)); the Chan Zuckerberg Initiative DAF, an advised fund of the Silicon Valley Community Foundation (grant no. CZF2019-002440); the Swiss National Science Foundation (grant no. 310030_192604); and the National Centre of Competence in Research Molecular Systems Engineering. A.V. is supported by an EMBO Fellowship (grant no. ALTF-1112-2019). J.S.F. was supported by a Boehringer Ingelheim Fonds PhD fellowship. C.M.-C. was supported by the SCORPION Austrian Science Fund (FWF) DOC 72-B27.

**Author contributions** C.L. and J.A.K. conceived the project and experimental design and secured the funding. C.L., J.S.F. and J.A.K. prepared the manuscript with input from all authors. C.L. performed all the experiments and analysed the data with the help of T.R.B., J.T., A.M.P., A.V., J.B.L. and C.E. J.S.F. performed the analysis of all single-cell RNA sequencing and multiome data under the supervision of B.T. M.S. and G.K. performed patient diagnosis and analysed magnetic resonance imaging. C.M.-C. generated induced pluripotent stem cell lines under the supervision of N.S.C. U.E. provided sgRNA predication.

**Competing interests** J.A.K. is on the supervisory and scientific advisory board of a:head bio AG (https://aheadbio.com) and is an inventor on several patents relating to cerebral organoids. The remaining authors declare no competing interests.

## Additional information
**Correspondence and requests for materials** should be addressed to Chong Li, Barbara Treutlein or Juergen A. Knoblich.

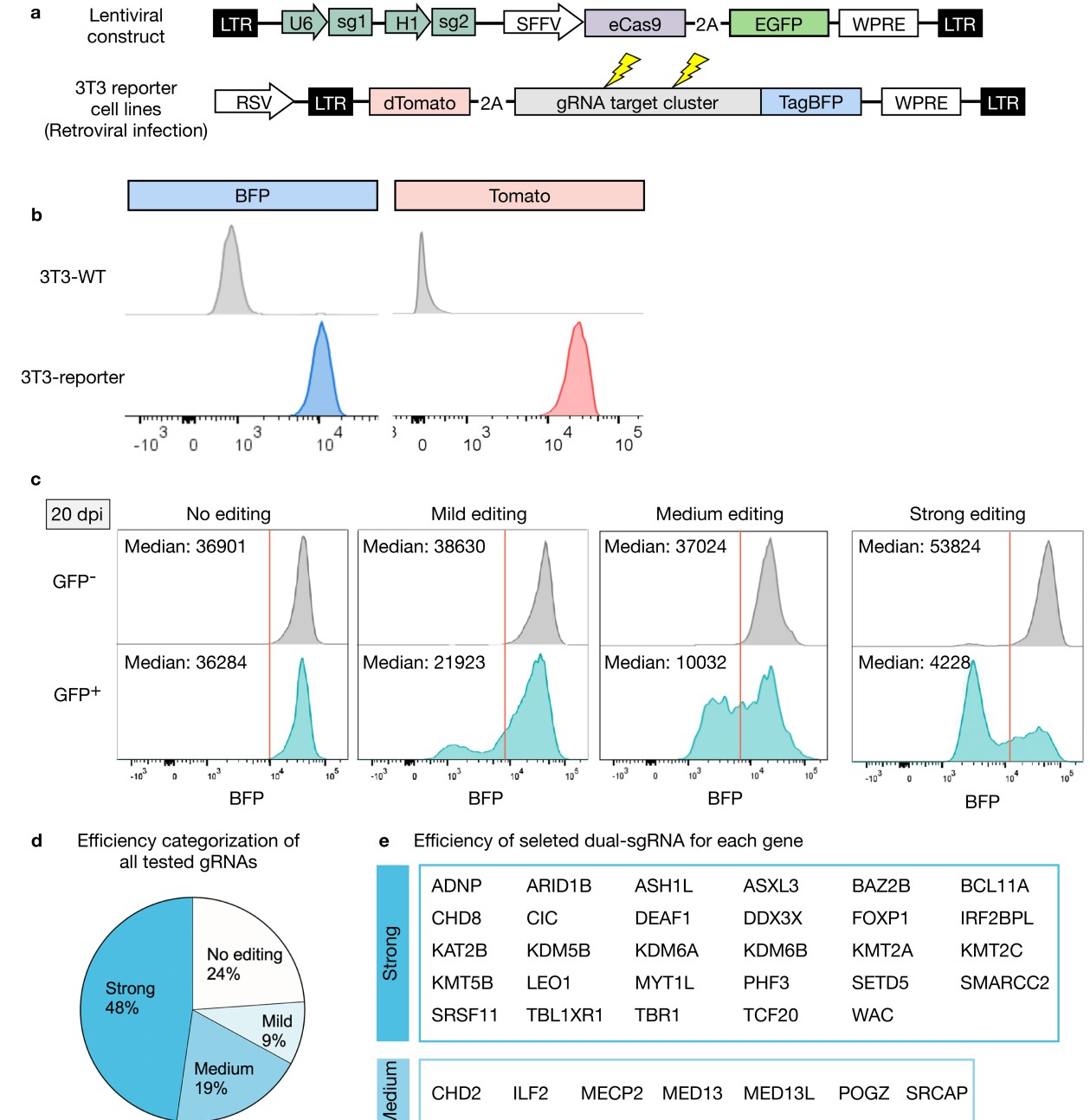

**Extended Data Fig. 1 | A gRNA reporter assay to determine gRNA efficiency.**
**a**, Diagram of the lentiviral construct delivering dual-sgRNA cassette and
eCas9 under the spleen focus forming virus (SFFV) promotor. Lentivirus
infected cells are labelled with GFP. Retroviral transduction is used to generate
3T3 cell lines. A pre-assembled array of gRNA-targeting sequences is fused with
TagBFP. **b**, Flow cytometry graph shows a 3T3 reporter cell line is positive for
both BFP and Tomato. After lentiviral infection, transduced cells (GFP+) and
internal control cells (GFP-) are subjected to FACS-based analysis at 20 dpi.
Efficient editing causing frameshift mutations lead to the loss of BFP
fluorescence. **c**, Four dual-sgRNA examples show no (0–10% reduction), mild
(10–45% reduction), medium (45–75% reduction) and strong editing efficiency
(> 75% reduction) at 20 dpi, respectively, based on median fluorescence
intensity. **d**, 98 pairs of gRNAs were tested and categorized into four groups.
**e**, Editing efficiency of selected paired gRNAs from 36 ASD genes.

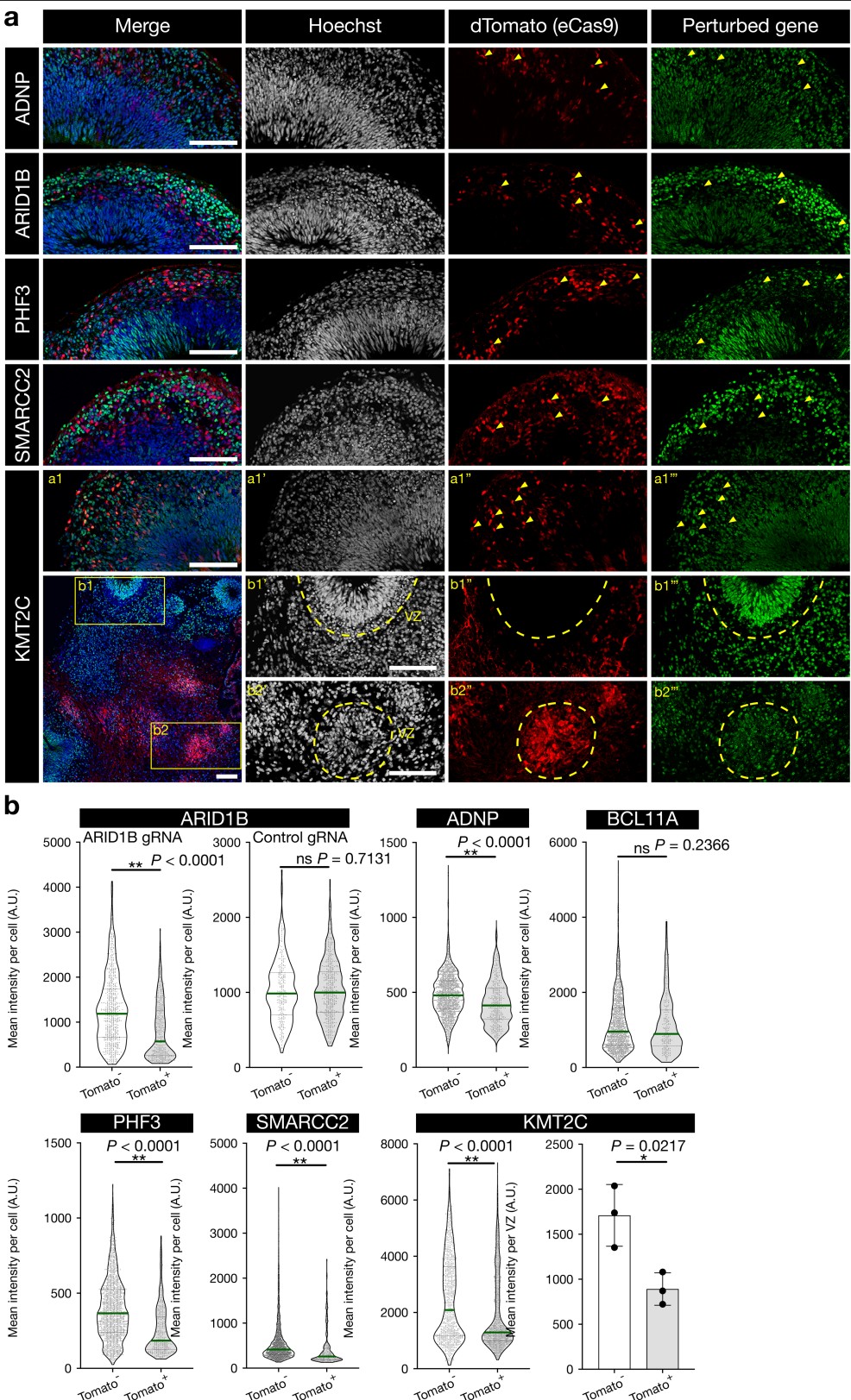

**Extended Data Fig. 2 | Perturbations of ASD genes efficiently decrease protein expression levels. a,** Confocal images showing IHC analyzes of protein expression levels in organoids with individual genes perturbed (*ADNP, ARID1B, PHF3, SMARCC2, KMT2C*). Note that both wild-type (dTomato-) and perturbed cells (dTomato+) present in each organoid. Arrowheads and dotted lines indicate perturbed cells or ventricular zones, respectively. **b,** Violin plots (all data points and median values; each data point represents a single cell; cells were measured from 4 organoids) showing quantification of protein expression levels, as measured by fluorescent intensity. Bar plot (mean ± S.D., n = 3 areas from 3 individual organoid) showing quantification of protein expression levels from eCas9-uninduced and induced VZ area. Two-sided student's t-test. *P < 0.05, **P < 0.01. Scale bar, 100 μm.

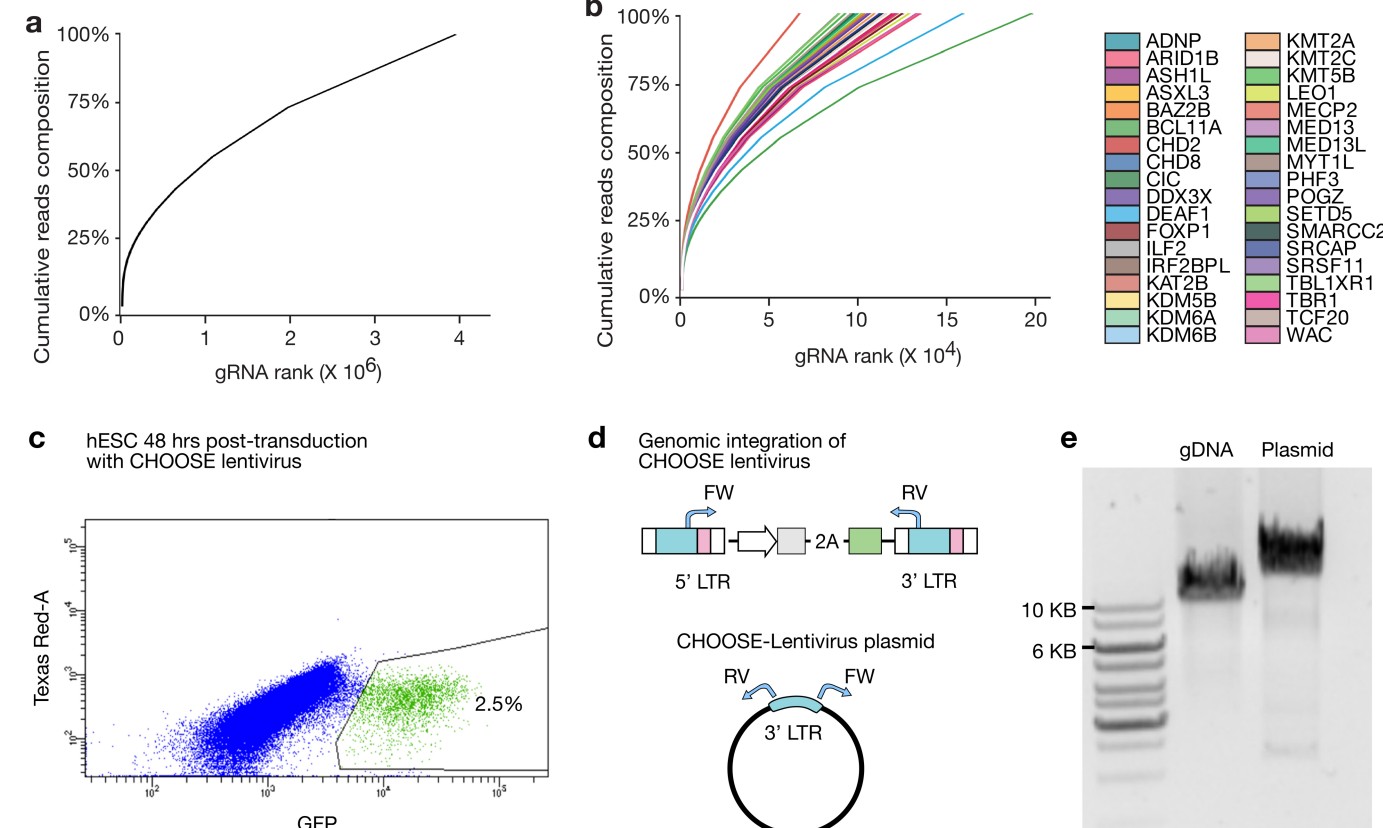

**Extended Data Fig. 3 | gRNA library, infection and integration of CHOOSE lentivirus in hES cells. a**, Plot showing cumulative fraction of uniquely barcoded gRNA cassettes from the CHOOSE ASD plasmid library. **b**, Plot showing cumulative fraction of uniquely barcoded gRNA cassettes separated by each target gene from the plasmid library. **c**, Infection rate of CHOOSE lentivirus on hES cells determined by flow-cytometry. Infected cells are positive for GFP. **d**, Top diagram showing the duplication of the dual-sgRNA cassette after the lentiviral integration into the host genome. FW and RV are paired primers used to demonstrate the successful integration of the lentivirus. The primers amplify specific regions of the genomic DNA extracted from lentivirus-infected hES cells, and the lentiviral plasmid. **e**, Representative gel electrophoresis analysis from 2 independent experiments showing 12 kb and 13 kb bands detected from gDNA and plasmid respectively.

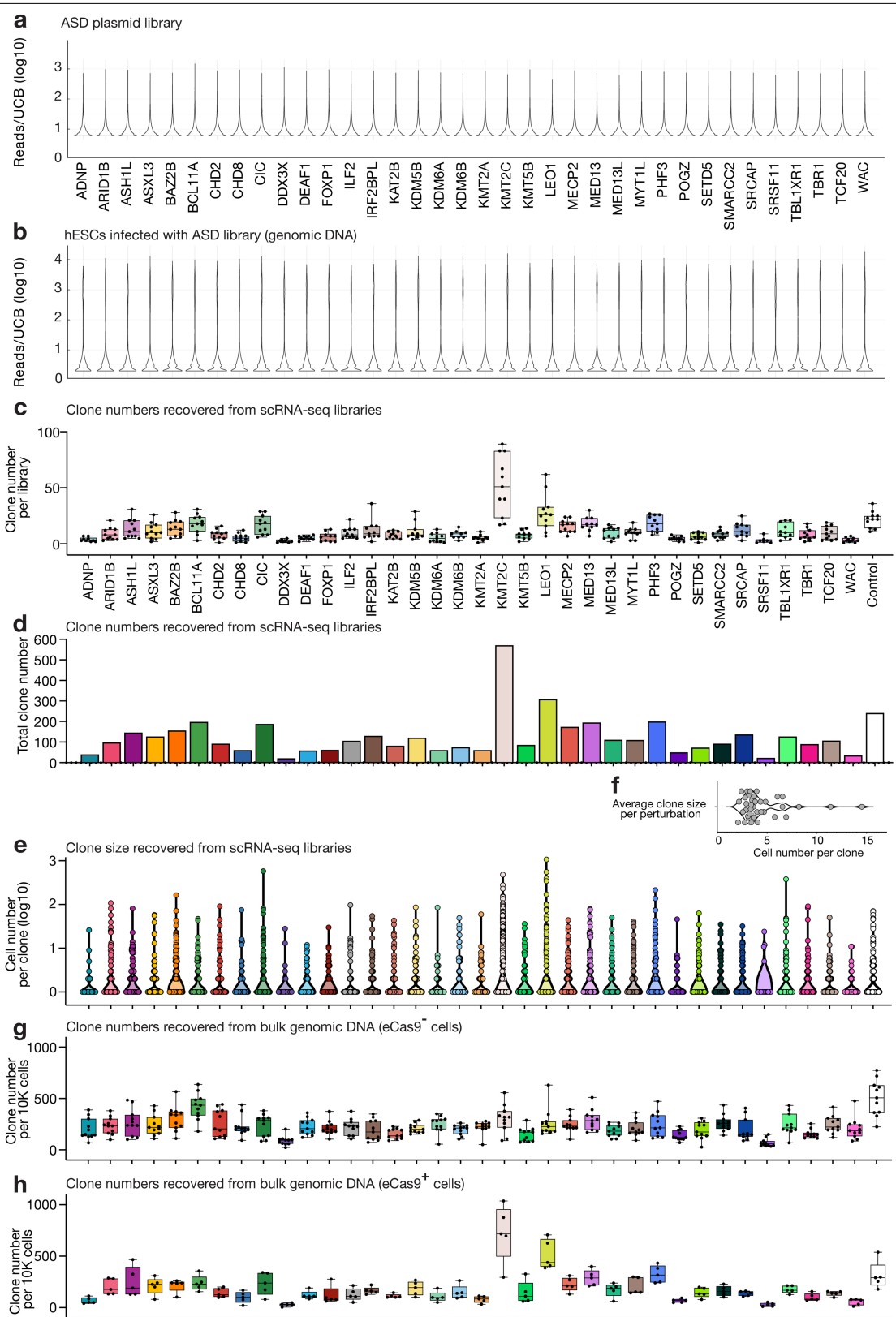

**Extended Data Fig. 4** | See next page for caption.

**Extended Data Fig. 4 | Clonal diversity of the CHOOSE organoids. a,b,** Reads distribution for each barcoded gRNA cassette from the plasmid library (**a**) and from genomic DNA (**b**) extracted from the hES cells infected by lentivirus. **c,** Box plot (minimum, maximum and median values; all data points) showing clone numbers recovered from scRNAseq libraries. Each data point represents a scRNAseq library. n = 11 independent pools (libraries) of organoids as replicates, 2 batches. **d,** Bar plot showing total clone numbers for each perturbation recovered from 11 scRNAseq libraries. **e,** Violin plot showing clone size distribution for each perturbation recovered from 11 scRNAseq libraries. Each data point represents one clone. **f,** Violin plot showing the average size of total clones recovered for each perturbation. **g, h,** Box plots (minimum, maximum and median values; all data points) showing the number of clones recovered from bulk genomic DNA extracted from eCas9-induced and uninduced cells. Each data point represents a bulk genomic DNA sample (50,000–150,000 cells sorted from pooled organoids). n = 5 pools of organoids for eCas9-induced cells; n = 11 pools of organoids for eCas9-uninduced cells. UCB, unique clone barcode.

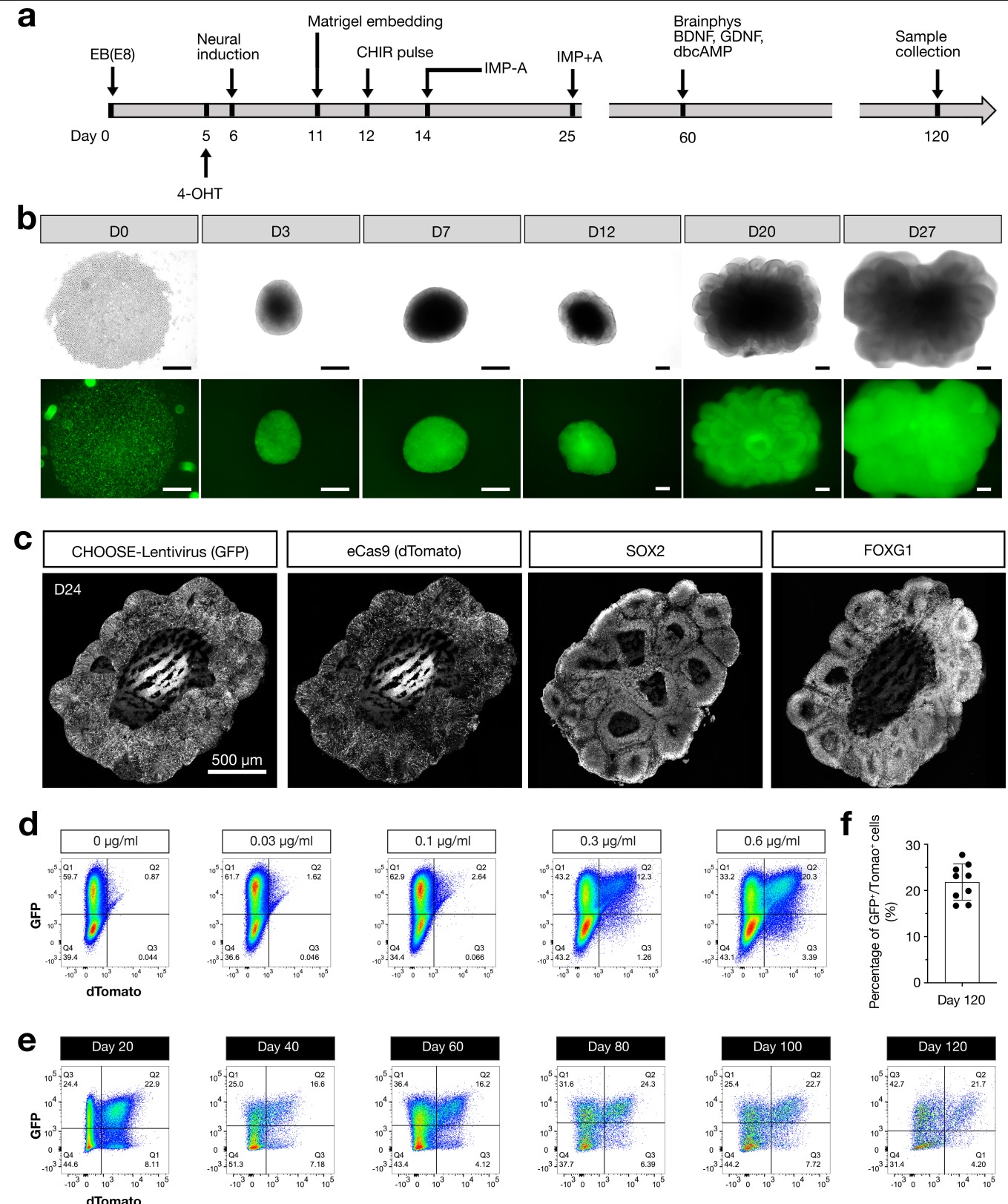

**Extended Data Fig. 5** | See next page for caption.

**Extended Data Fig. 5 | Generation of cerebral organoids using the CHOOSE system. a**, Schematic showing the protocol used for generating cerebral organoids. 4-OHT was added to induce eCas9 expression at day 5. A short 3-day CHIR treatment was applied from day 12–14. Culture medium was switched to Brainphys after day 60 for advanced neuronal maturation. Tissues are collected at day 120 and subjected to scRNA-seq assays. **b**, ES cells were infected by lentivirus and GFP positive cells were collected to make EBs. Microscopic images show the overall morphology and GFP expression of cerebral organoids overtime. Top, bright-field; Bottom, GFP fluorescence. Scale bar, 100 μm. **c**. Immunohistochemical staining of cerebral organoids on day 24. GFP labels cells infected with lentivirus and Tomato is a reporter for eCas9 expression.

SOX2 is a marker for progenitor cells. FOXG1 is a marker for tissues with telencephalon identity. **d**, FACS plots showing percentages of eCas9-induced cell population (GFP+/dTomato+) at day 30. Organoids are treated with different concentrations of 4-OHT at day 5. **e**, FACS plots showing percentages of eCas9-induced cell population at different developmental stages when organoids are treated with 0.3 ug/ml 4-OHT. **f**, Bar plot (mean ± S.D.) showing average eCas9 induction rate at day 120 when organoids are treated with 0.3 ug/ml 4-OHT. Each dot indicates a pool of 3–7 organoids. n = 9 biologically independent pools of organoids, examined over 2 independent experimental batches. Data are presented as mean values ± SD.

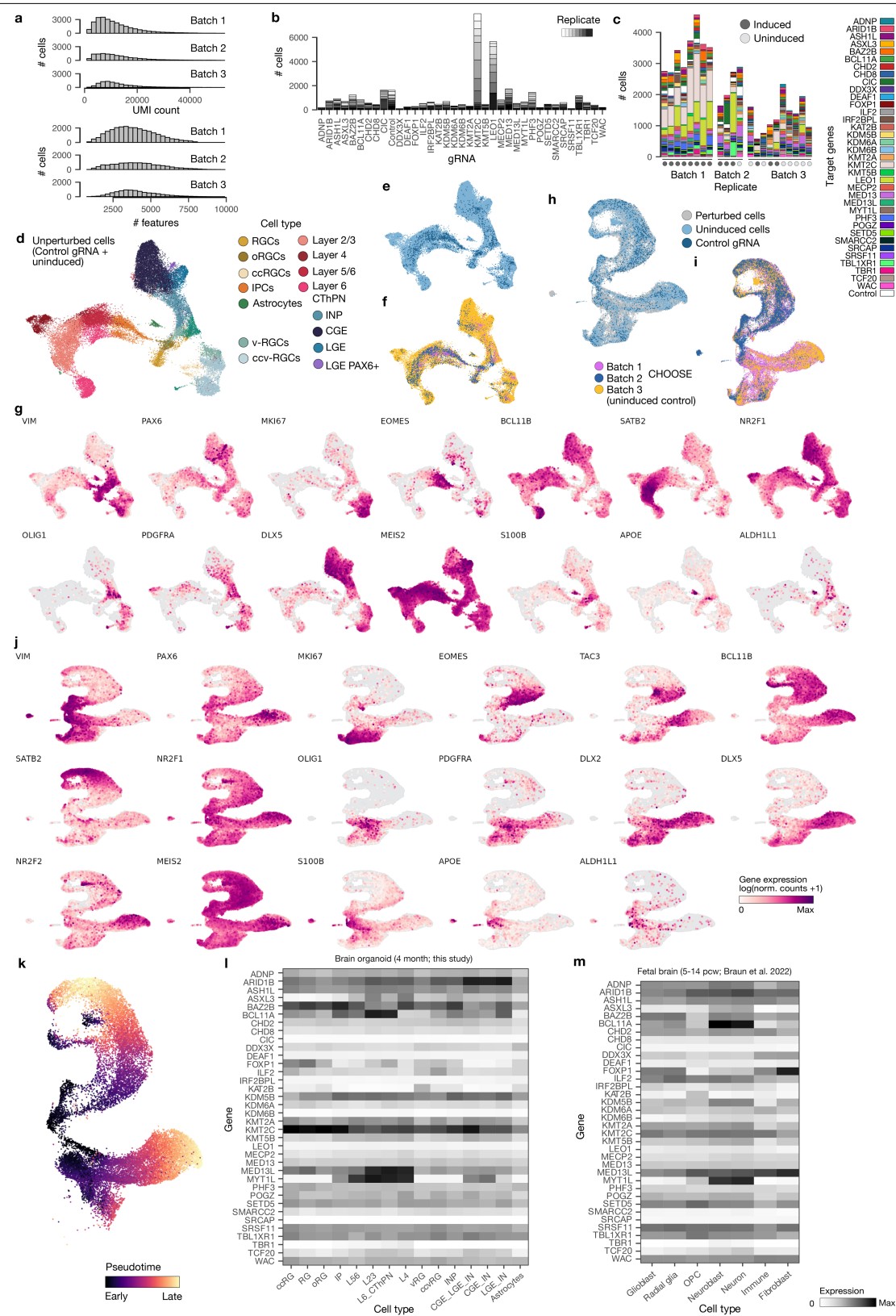

**Extended Data Fig. 6 | Cell type and gRNA composition of scRNA-seq libraries. a**, Histograms showing UMI count and number of detected features in the QC-controlled scRNA-seq dataset for each batch of organoids. **b**, Bar plot showing numbers of recovered cells with assigned gRNAs for each library. **c**, Bar plots showing cell number and gRNA composition for each replicate split by batch. **d**, UMAP embedding of unperturbed cells (control gRNA and eCas9-uninduced) coloured by annotated cell type. **e,f,h,i**, UMAP embedding of the unperturbed cells and full CHOOSE dataset including induced and uninduced samples filtered for gRNA detection. Cells are coloured by perturbation status (**e,h**) and batch (**f,i**). **g,j**, Feature plots showing expression of marker genes on a UMAP embedding for unperturbed cells (**g**) and full CHOOSE dataset (**j**). **k**, UMAP embedding coloured by velocity pseudotime to show developmental directions. **l,m**, Heatmaps showing expression of perturbed target genes in cerebral organoid cell types (**l**) and cell types in the developing human brain (Braun et al.[32]) (**m**).

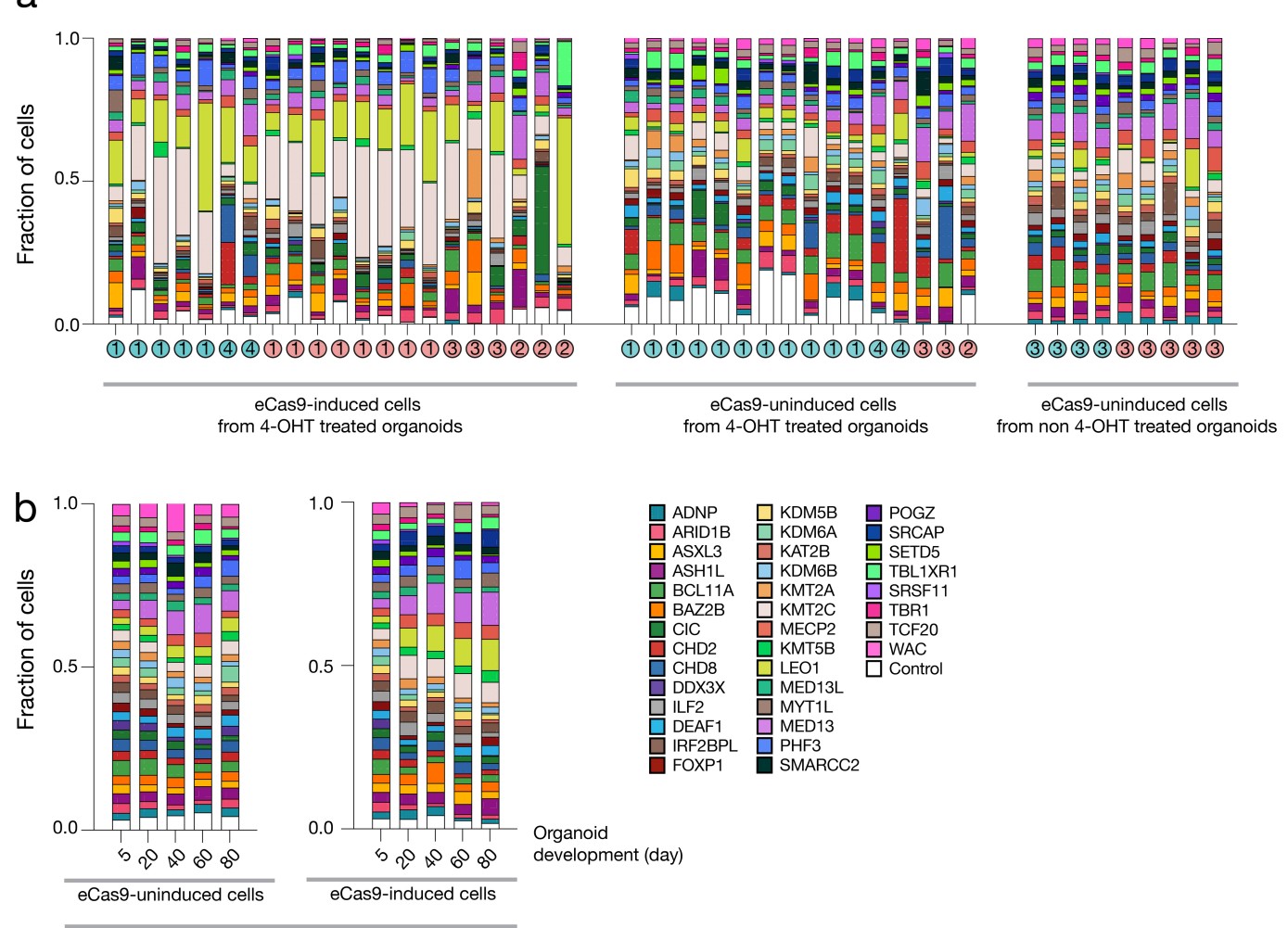

**Extended Data Fig. 7 | gRNA representation in eCas9-induced and uninduced cell populations of the CHOOSE organoids. a**, Bar plots showing gRNA representation in scRNAseq libraries (pink circle; each library is an independent pool of organoids) and in bulk genomic DNA extracted from eCas9-induced and uninduced cell populations (blue circle; each sample is sorted from an independent pool of organoids) at 4 months. The numbers within each circle indicate batch numbers. Note batch 3 does not include non-targeting gRNA control and uses internal eCas9-uninduced cells as an alternative control. **b**, Bar plot showing gRNA representation in bulk genomic DNA extracted from eCas9-induced and uninduced cell populations sorted from organoid pools at different development stages.

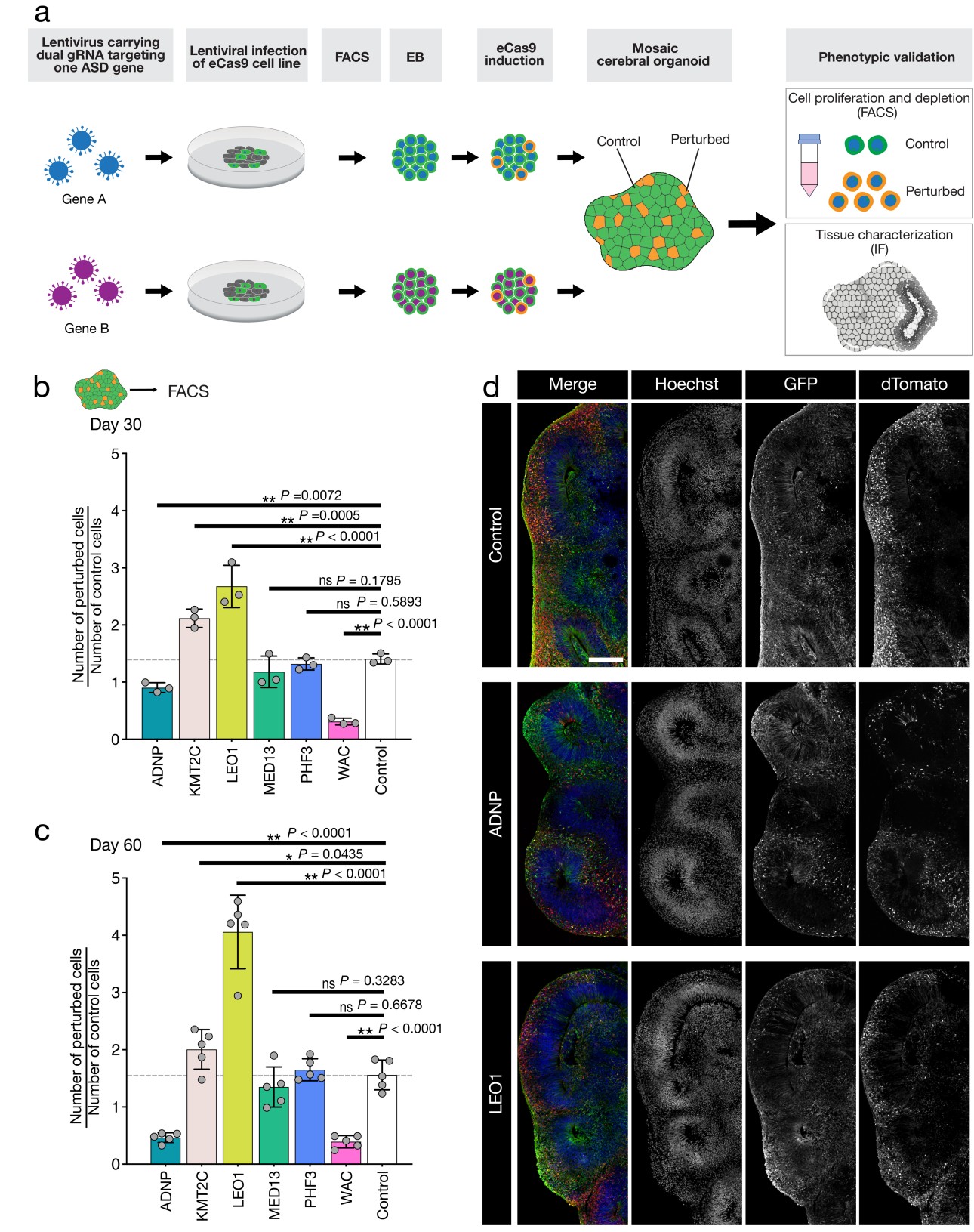

**Extended Data Fig. 8 | Perturbations of ASD risk genes lead to cell proliferation or depletion. a**, Schematic showing organoids perturbed with individual genes are subjected to phenotypic characterizations through FACS and IHC analyzes. Note that wild-type (eCas9-) and perturbed (eCas9+) cells present in each organoid. **b**, **c**, Bar plots (mean ± S.D., n = 3 (**b**) or 5 (**c**) biologically independent pools, each pool contains 3 organoids) showing perturbation-induced over-proliferation (*KMT2C* and *LEO1*) and depletion (*ADNP* and *WAC*). Over-proliferation and depletion were calculated using the

number of induced cells divided by the number of un-induced cells sorted from the same pool of organoids, and compared to non-targeting gRNA control. One-way *ANOVA* post hoc Fisher's LSD test. *$P < 0.05$, **$P < 0.01$. **d**, Confocal images showing representative images of organoids with individual genes perturbed. IHC analyzes were performed on 60-day-old organoids from at least 4 organoids over two independent batches. GFP labels cells expressing gRNA construct and dTomato indicates the induction of eCas9. Scale bar, 200 μm.

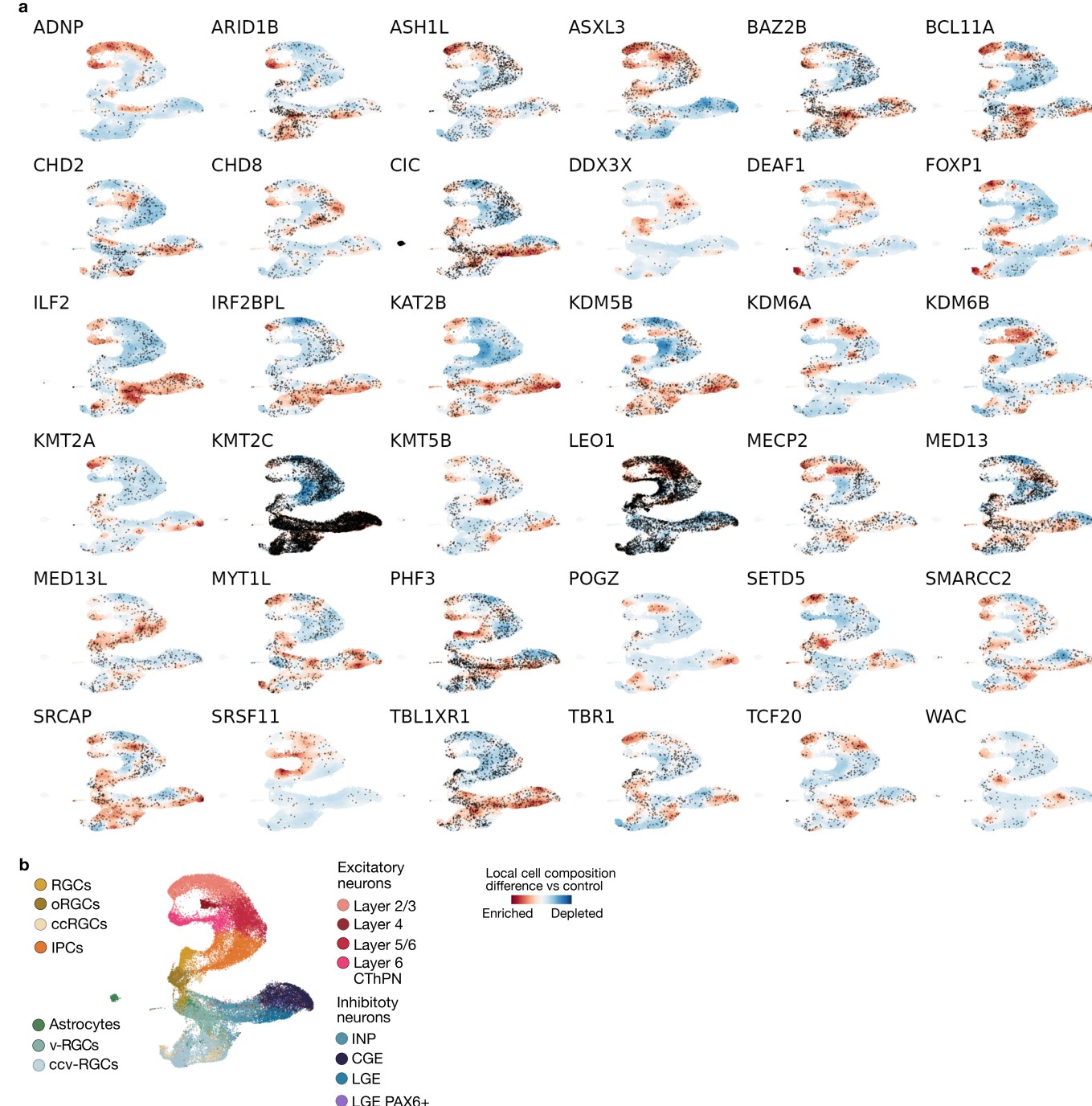

**Extended Data Fig. 9 | Cell type compositional changes caused by ASD gene perturbations. a**, UMAP embeddings show single cell distribution for each perturbed gene. Local cell composition differences of perturbation versus control indicated by red (enrichment) or blue (depletion). **b**, UMAP embedding of the scRNA-seq dataset containing dorsal and ventral telencephalon trajectories.

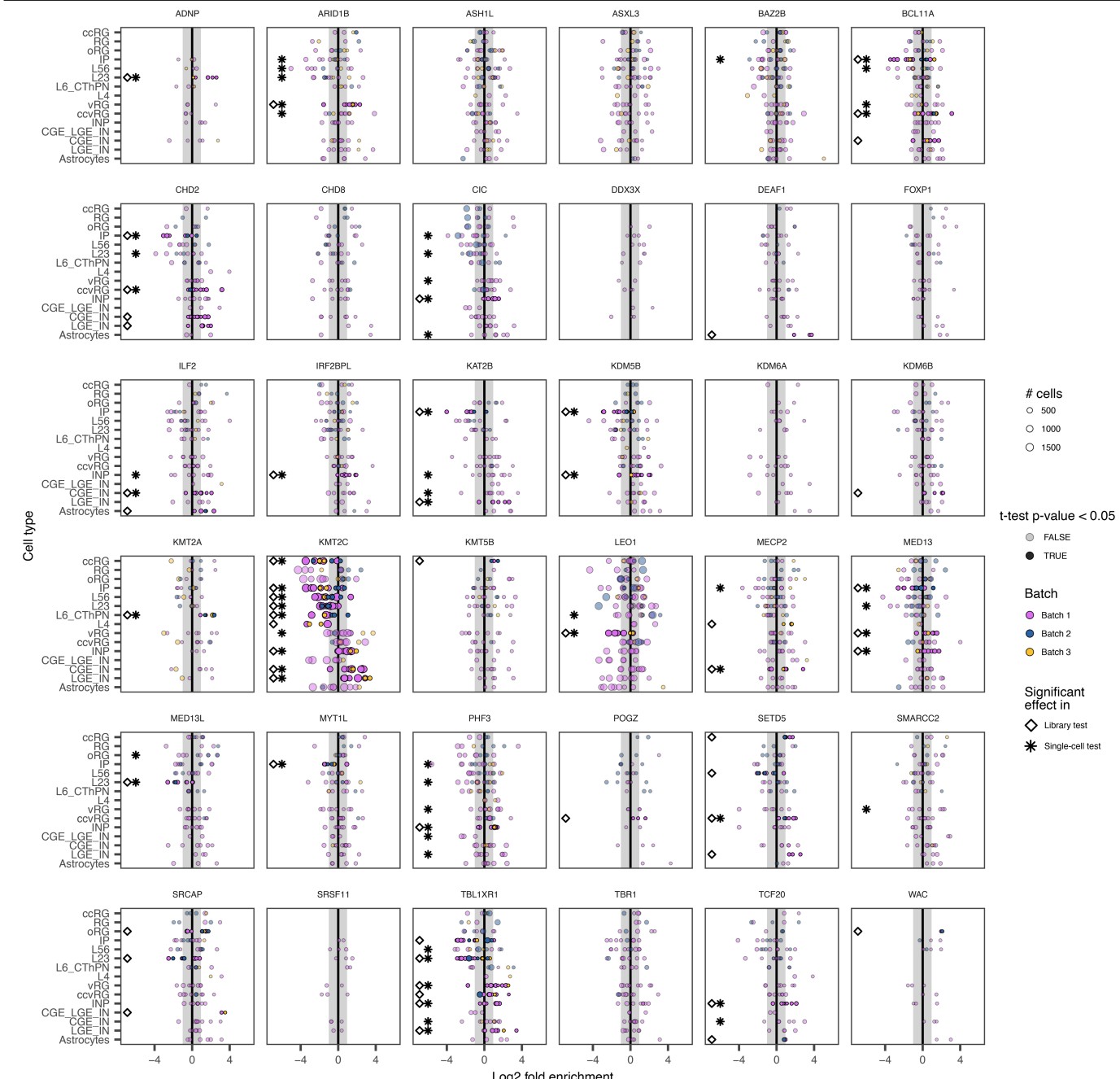

**Extended Data Fig. 10 | Differential abundance effects of perturbations across independent pools of organoids.** Lolliplots showing effect size of differential abundance (log odds ratio) per cell type and target gene. Dot colour indicates batch, size indicates number of cells. A t-test was performed against a permuted background distribution to highlight highly consistent effects across replicates. Stars indicate statistical significance in the single cell-level test, diamonds indicate significance of a two-sided t-test across pools of organoids (*P* < 0.05).

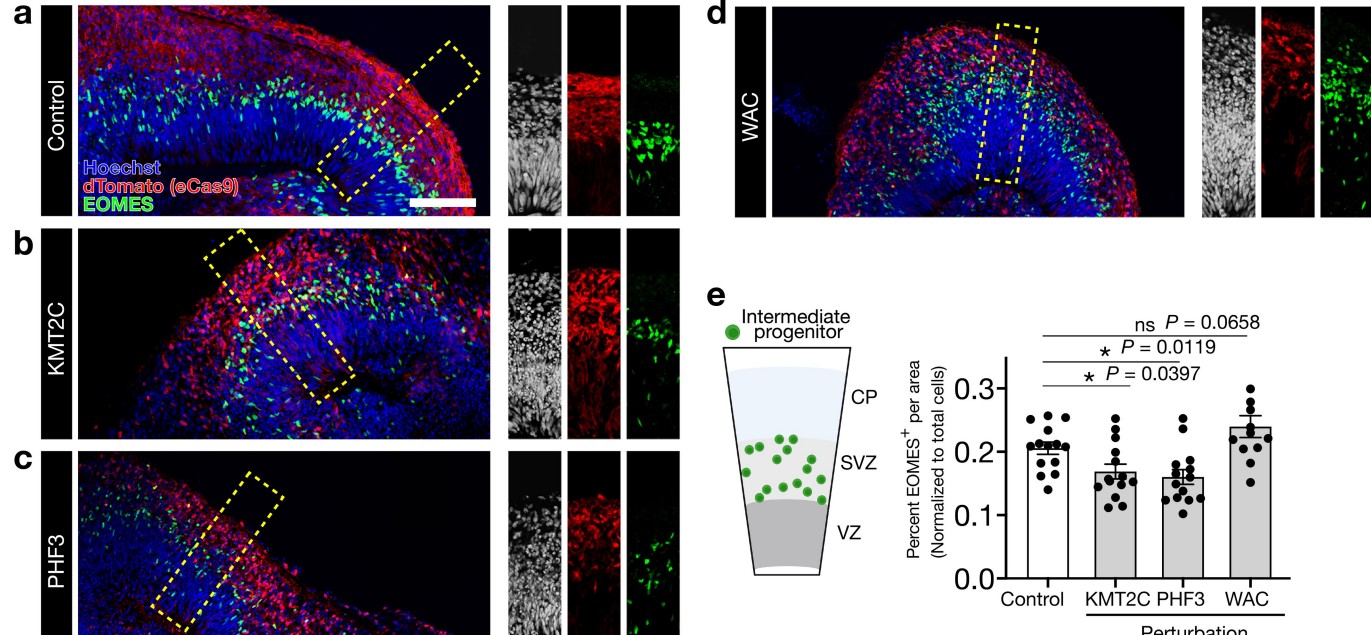

**Extended Data Fig. 11 | Perturbations of ASD risk genes lead to depletion of IPs. a-d,** Confocal images showing immunohistochemical staining of EOMES on 60-day-old organoids perturbed with individual genes. dTomato labels eCas9-induced cells. Yellow box labels radially organized structure with ventricular zone (VZ), subventricular zone (SVZ) and cortical plate (CP).

Note the presence of EOMES+ cells in the SVZ. **e,** Bar plot (mean ± S.E.M.; all data points; from left to right, n = 14, 14, 14, 12 areas collected from 4 organoids for each perturbation) showing quantification of percent EOMES+ cells per eCas9 positive column area. Individual perturbations are compared to non-targeting gRNA control. Two-sided student's t-test. *$P < 0.05$, **$P < 0.01$. Scale bar, 100 µm.

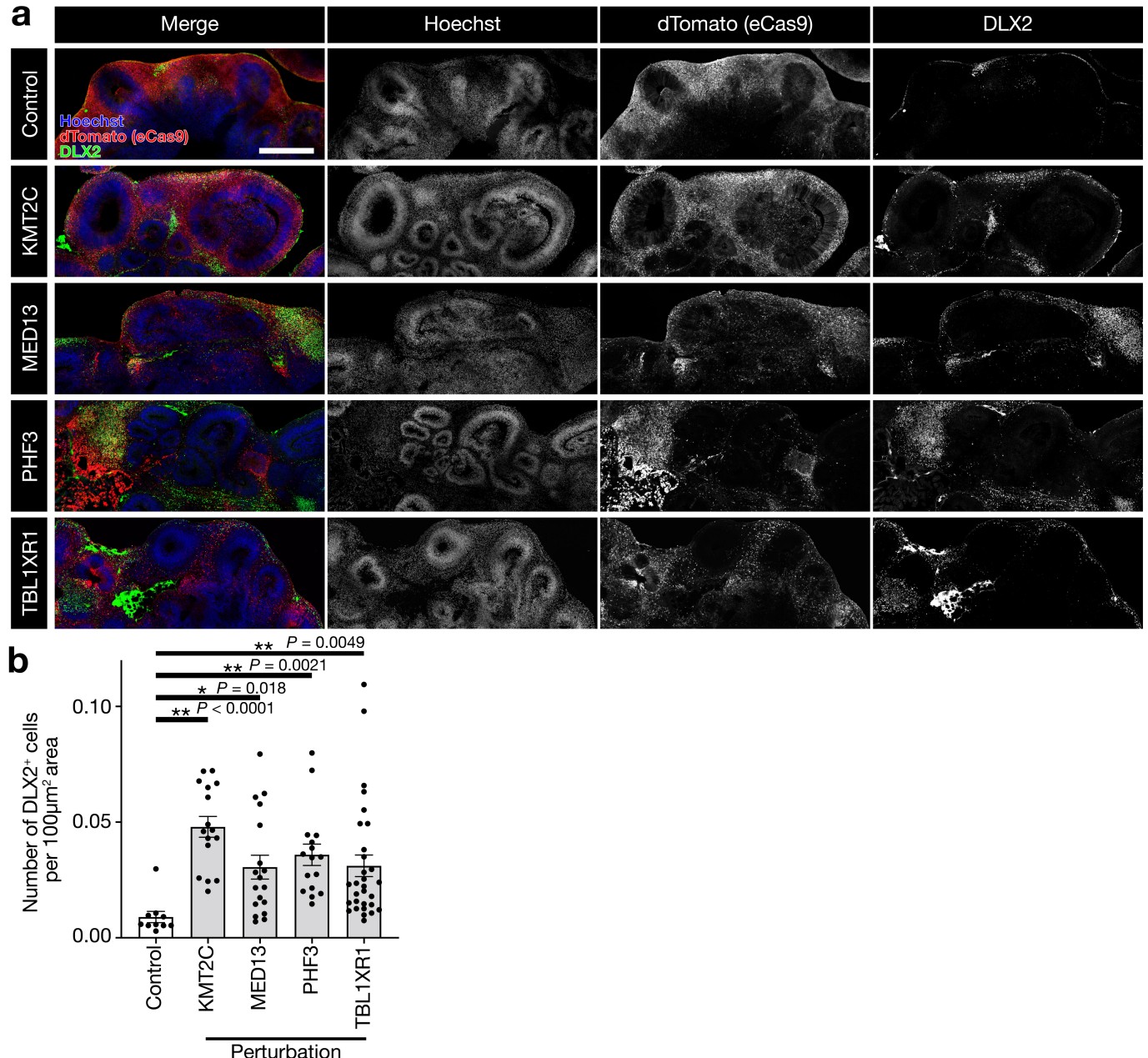

**Extended Data Fig. 12 | Perturbations of ASD risk genes lead to enrichment of INPs. a**, Confocal images showing immunohistochemical staining of DLX2 on 60-day-old organoids perturbed with individual genes. dTomato labels eCas9-induced cells. **b**, Bar plot (mean ± S.E.M.; all data points; from left to right, n = 16, 18, 16, 30 areas collected from 4 organoids for each perturbation) showing quantification of the number of DLX2+ cells per 100 μm² area. Areas containing multiple rosettes from each organoid were collected for quantification. Individual perturbations are compared to non-targeting gRNA control. Two-sided student's t-test. *$P < 0.05$, **$P < 0.01$. Scale bar, 500 μm.

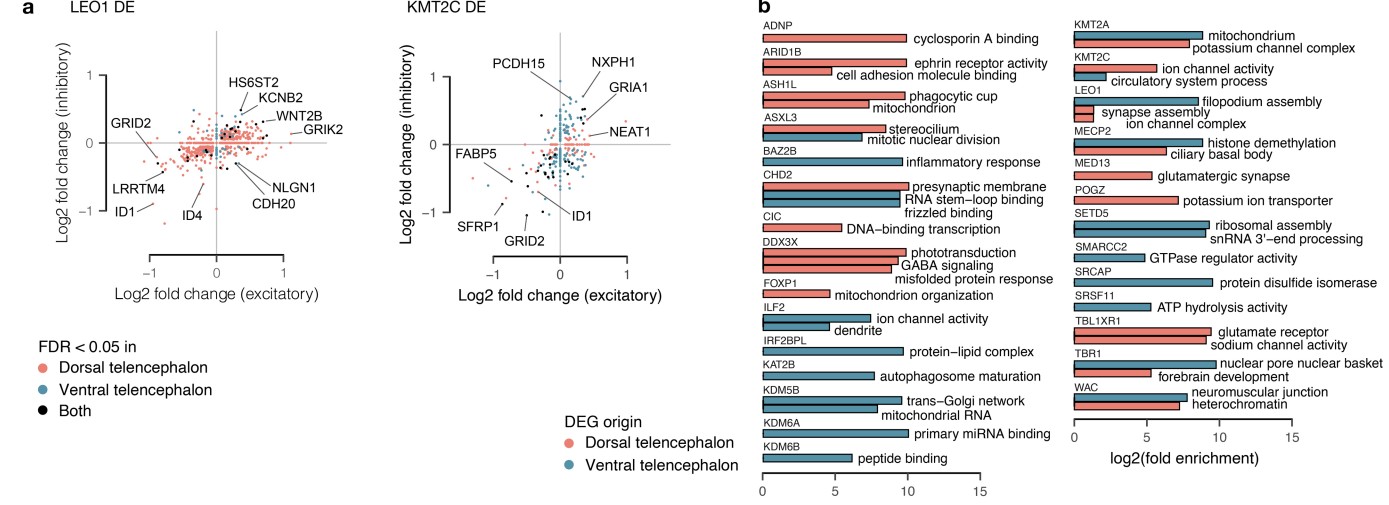

**Extended Data Fig. 13 | Differential expression and GO enrichment analysis of ASD genetic perturbations. a**, Scatter plot showing differential expression log fold change in dorsal and ventral telencephalon cells for LEO1 and KMT2C. Dot colour indicates the trajectory where the gene was differentially expressed.

**b**, Bar plot showing perturbation-specific GO enrichment of DEGs in dorsal and ventral telencephalon cells. As a background gene set we used all genes expressed in >5% of cells in the dataset.

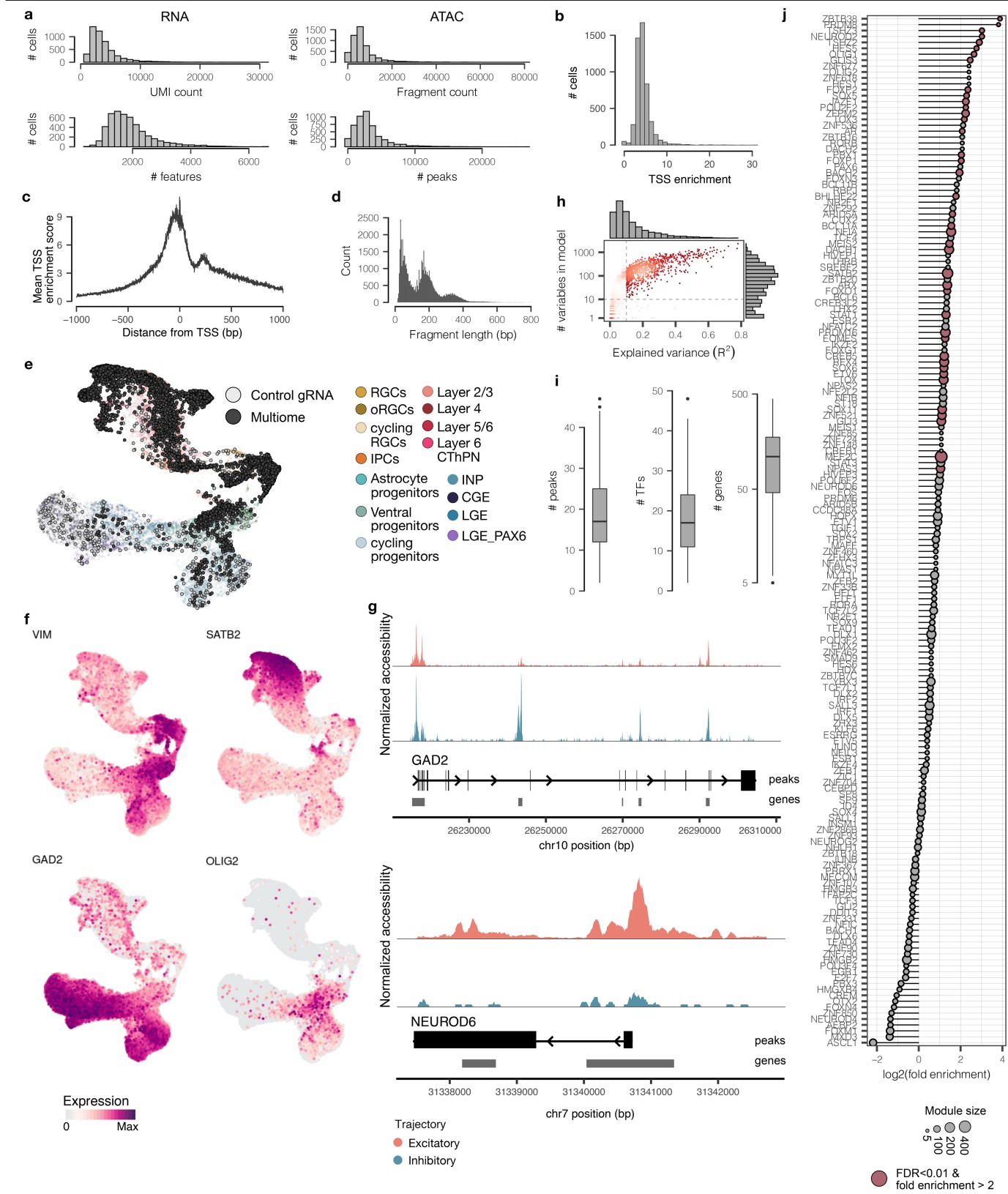

**Extended Data Fig. 14** | See next page for caption.

**Extended Data Fig. 14 | Quality control and GRN inference from the single-cell multiome dataset. a**, Histograms showing UMI count and number of detected features for RNA as well as fragment count and number of detected peaks for ATAC. **b**, Histogram showing distribution of transcription start site (TSS) enrichment scores across cells. **c**, Line plot showing TSS enrichment score at each position relative to the TSS. **d**, Histogram showing distribution of fragment length. **e**, UMAP embedding showing multiome data (black) integrated with the CHOOSE dataset. Multiome data were used in conjunction with the control cells from the CHOOSE dataset (grey) to infer the GRN with Pando. **f**, Feature plots showing expression of marker genes on the UMAP embedding. **g**, Genomic tracks showing accessible peaks in the proximity of GAD2 (ventral marker) and NEUROD6 (dorsal marker). **h**, Density scatter plot histograms showing the distributions of explained variance (x) and number of variables (y) in the fitted models for GRN construction. Dashed lines indicate the thresholds used for model selection. **i**, Boxplots showing the distribution of peaks (left) and TFs assigned per gene (middle, n = 864), and number of genes assigned per TF (right, n = 241) in the inferred GRN. The centre line represents the median, the box limits show the 25–75% interquartile range and the whiskers indicate 1.5× the interquartile range. **j**, Lolliplot showing the enrichment of ASD-associated genes from SFARI in inferred TF modules. Red colour indicates an FDR-corrected two-sided Fisher-test p-value of <0.01. Dot size indicates the total number of genes in the module.

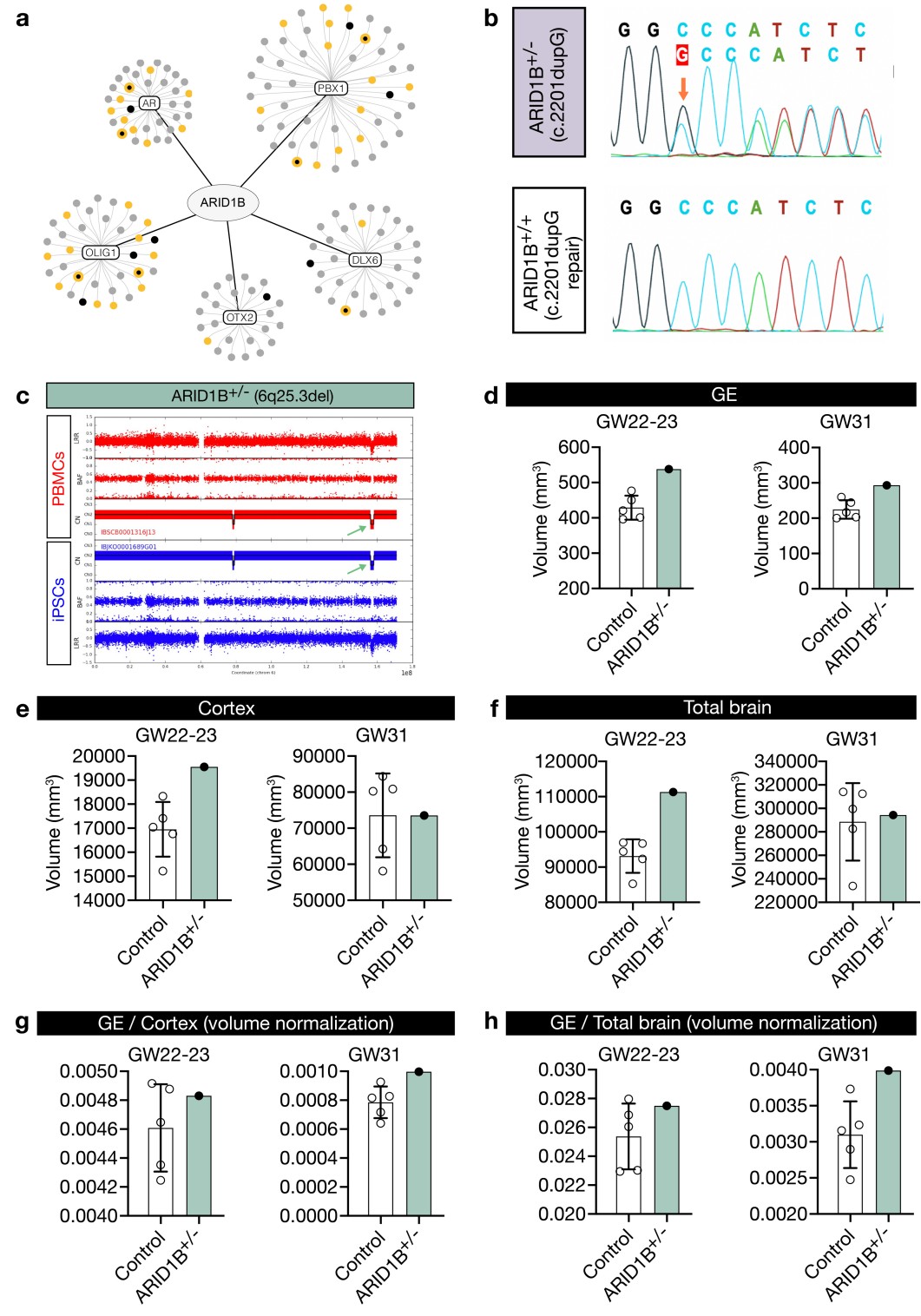

**Extended Data Fig. 15 | An ARID1B patient had enlarged ganglionic eminence. a**, Graph representation of *AR*, *PBX1*, *DLX6*, *OTX2*, and *OLIG1* TF modules, which are most strongly enriched in *ARID1B* DEG (Two-sided Fisher exact test p-value < 0.01, top 4 odds ratio). The targets highlighted in yellow are SFARI genes, and the targets highlighted in black are TF modules enriched in SFARI genes. **b**, Top, sanger sequencing showing a duplication mutation (red arrow) identified in one of the alleles of patient 1. Bottom, sanger sequencing showing the mutation was repaired and this cell line is used as an isogenic control. Sequencing was done on fragments amplified from genomic DNA extracted from the two iPS cell lines. **c**, SNP array genotyping of PBMCs and iPS cells from patient 2 shows a microdeletion (dark green arrow) identified on chromosome 6. **d-f**, Bar plot showing the absolute GE (**d**), cortex (**e**) and total brain (**f**) volumes of control (mean ± S.D.; n = 5, age macheted) and patient 2 at two gestational stages. **g-h**, Bar plots showing normalized GE to cortex volume (**g**) and total brain volume (**h**) of control (mean ± S.D.; n = 5, age macheted) and patient 2 at two gestational stages.

Barbara Treutlein
Juergen A. Knoblich

# Reporting Summary

## Statistics

For all statistical analyses, confirm that the following items are present in the figure legend, table legend, main text, or Methods section.

| n/a | Confirmed | |
|---|---|---|
| ☐ | ☒ | The exact sample size (*n*) for each experimental group/condition, given as a discrete number and unit of measurement |
| ☐ | ☒ | A statement on whether measurements were taken from distinct samples or whether the same sample was measured repeatedly |
| ☐ | ☒ | The statistical test(s) used AND whether they are one- or two-sided<br>*Only common tests should be described solely by name; describe more complex techniques in the Methods section.* |
| ☐ | ☒ | A description of all covariates tested |
| ☐ | ☒ | A description of any assumptions or corrections, such as tests of normality and adjustment for multiple comparisons |
| ☐ | ☒ | A full description of the statistical parameters including central tendency (e.g. means) or other basic estimates (e.g. regression coefficient) AND variation (e.g. standard deviation) or associated estimates of uncertainty (e.g. confidence intervals) |
| ☐ | ☒ | For null hypothesis testing, the test statistic (e.g. $F$, $t$, $r$) with confidence intervals, effect sizes, degrees of freedom and $P$ value noted<br>*Give P values as exact values whenever suitable.* |
| ☒ | ☐ | For Bayesian analysis, information on the choice of priors and Markov chain Monte Carlo settings |
| ☒ | ☐ | For hierarchical and complex designs, identification of the appropriate level for tests and full reporting of outcomes |
| ☐ | ☒ | Estimates of effect sizes (e.g. Cohen's *d*, Pearson's *r*), indicating how they were calculated |

*Our web collection on statistics for biologists contains articles on many of the points above.*

## Software and code

Policy information about availability of computer code

| Data collection | CellSense software was used for microscopic images acquisition.<br>FACSDiva software was used for FACS data acquisition. |
|---|---|
| Data analysis | FlowJO was used for FACS data analysis.<br>FIJI was used for microscopic image processing. |

For manuscripts utilizing custom algorithms or software that are central to the research but not yet described in published literature, software must be made available to editors and reviewers. We strongly encourage code deposition in a community repository (e.g. GitHub). See the Nature Portfolio guidelines for submitting code & software for further information.

## Data

Policy information about availability of data

All manuscripts must include a data availability statement. This statement should provide the following information, where applicable:
- Accession codes, unique identifiers, or web links for publicly available datasets
- A description of any restrictions on data availability
- For clinical datasets or third party data, please ensure that the statement adheres to our policy

Raw sequencing data will be deposited on ArrayExpress. Processed Seurat objects were deposited on Zenodo (DOI: 10.5281/zenodo.7083558). The Pando R package is available on GitHub (https://github.com/quadbiolab/Pando). Other custom code used in the analyses is deposited on GitHub (https://github.com/quadbiolab/ASD_CHOOSE).

# Field-specific reporting

Please select the one below that is the best fit for your research. If you are not sure, read the appropriate sections before making your selection.

☒ Life sciences          ☐ Behavioural & social sciences          ☐ Ecological, evolutionary & environmental sciences

For a reference copy of the document with all sections, see nature.com/documents/nr-reporting-summary-flat.pdf

# Life sciences study design

All studies must disclose on these points even when the disclosure is negative.

| | |
|---|---|
| Sample size | For scRNAseq screening of the 4-month-old CHOOSE organoids, we analyzed 14 10X genomics 3' GEX libraries. Each library consists of a pool of 3-7 organoids, and in total 65 organoids were sampled. For bulk analysis of gRNA representation and clone information, 24 samples (50-150K cells input) were used. For individual validation of over-proliferation and depletion, 3 or 5 pools of organoids (3 organoids each pool) were analyzed. For individual validation of intermediate progenitors and interneuron precursor cells, at least 4 organoids were analyzed for each gene. For phenotypic characterization of ARID1B patient iPSCs-derived organoids, a minimum of 3 independent batches and 13 organoids for each cell line were subjected to analyses. No statistical methods were used to predetermine sample sizes. |
| Data exclusions | No data were excluded. |
| Replication | For the CHOOSE screen, 65 cerebral organoids were sampled from 3 batches. 14 scRNA-seq libraries (biological replicates) were prepared and analyzed. 3-7 organoids were included within each library. For each ARID1B iPSC line, a minimum of 13 organoids from 3 batches were repeated. For individual validation of over-proliferation and depletion, intermediate progenitors and interneuron precursors, replicates were done from 2 batches of organoids. The individual "n" values were indicated either by displaying all data points or in the figure legend. |
| Randomization | Samples were not randomized to each experimental groups. |
| Blinding | Investigators were not blinded in this study. |

# Reporting for specific materials, systems and methods

We require information from authors about some types of materials, experimental systems and methods used in many studies. Here, indicate whether each material, system or method listed is relevant to your study. If you are not sure if a list item applies to your research, read the appropriate section before selecting a response.

## Materials & experimental systems

| n/a | Involved in the study |
|---|---|
| ☐ | ☒ Antibodies |
| ☐ | ☒ Eukaryotic cell lines |
| ☒ | ☐ Palaeontology and archaeology |
| ☒ | ☐ Animals and other organisms |
| ☐ | ☒ Human research participants |
| ☒ | ☐ Clinical data |
| ☒ | ☐ Dual use research of concern |

## Methods

| n/a | Involved in the study |
|---|---|
| ☒ | ☐ ChIP-seq |
| ☐ | ☒ Flow cytometry |
| ☐ | ☒ MRI-based neuroimaging |

## Antibodies

| | |
|---|---|
| Antibodies used | DLX2 (Santa Cruz, cat. no. SC393879, 1:100), OLIG2 (Abcam, cat. no. ab109186, 1:100), SOX2 (R&D, cat. no. MAB2018, 1:500), FOXG1 (Abcam, cat. no. ab18259, 1:200), EOMES (R&D, cat. no. AF6166, 1: 200), ARID1B (cell signaling, cat. no. 92964, 1:100), ADNP (ThermoFisher, cat. no. 702911, 1:250), BCL11A (abcam, cat. no. 191401, 1:250), PHF3 (Sigma, cat. no. HPA024678, 1:250), SMARCC2 (ThermoFisher, cat. no. PA5-54351, 1:250), KMT2C (Sigma, cat. no. HPA074736, 1:250), Alexa 488, 568, 647 conjugated secondary antibodies raised in donkey (ThermoFisher, 1:250) |
| Validation | DLX2 (Santa Cruz, cat. no. SC393879, mouse, 1:100) has been validated by the company and used in 3 scientific literatures. OLIG2 (Abcam, cat. no. ab109186, rabbit, 1:100) has been validated by the company and used in 94 scientific literatures. SOX2 (R&D, cat. no. MAB2018, 1:500) has been validated by the company and used in 145 scientific literatures. FOXG1 (Abcam, cat. no. ab18259, rabbit, 1:200) has been validated by the company and used in 88 scientific literatures. ARID1B (cell signaling, cat. no. 92964, mouse, 1:100), has been validated by the company and used in 9 scientific literatures. ADNP (aThermoFisher, cat. no. 702911, 1:250) has been validated by the company. BCL11A (abcam, cat. no. 191401, rabbit, 1:250), has been validated by the company and used in 2 scientific literatures. PHF3 (Sigma, cat. no. HPA024676, rabbit, 1:250), developed and validated by the Human Protein Atlas (HPA) project. SMARCC2 (ThermoFisher, cat. no. PA5-54351, rabbit, 1:250), has been validated by the company. |

KMT2C (Sigma, cat. no. HPA074736, rabbit, 1:250), developed and validated by the Human Protein Atlas (HPA) project.

# Eukaryotic cell lines

Policy information about cell lines

| | |
|---|---|
| Cell line source(s) | Patient induced pluripotent stem cells were reprogrammed in IMBA SCCF from blood (PBMCs).<br>Inducible eCas9 cell line was previously generated in the lab using H9 cell line (obtained from WiCell).<br>HEK293T and NIH3T3 cell lines were purchased from ATCC.<br>Plat-E cell line was purchased from CellBiolabs. |
| Authentication | All stem cell lines used were authenticated using a short tandem repeat (STR) assay, tested for genomic integrity using SNP array genotyping.<br>HEK293T, NIH3T3 and Plat-E cell lines were not authenticated. |
| Mycoplasma contamination | All stem cell lines were tested negative for mycoplasma.<br>HEK293T, NIH3T3 and Plat-E cell lines were not tested for mycoplasma. |
| Commonly misidentified lines<br>(See ICLAC register) | No ICLAC lines were used. |

# Human research participants

Policy information about studies involving human research participants

| | |
|---|---|
| Population characteristics | Women with singleton pregnancies undergoing fetal MRI at a tertiary care center from January 2016 and December 2021 were retrospectively reviewed. |
| Recruitment | A retrospective review of patient records was performed and a patient with a positive genetic testing report for ARID1B mutation was selected. |
| Ethics oversight | The study was approved by the institutional ethics board (Medical University of Vienna). |

Note that full information on the approval of the study protocol must also be provided in the manuscript.

# Flow Cytometry

## Plots

Confirm that:

☒ The axis labels state the marker and fluorochrome used (e.g. CD4-FITC).

☒ The axis scales are clearly visible. Include numbers along axes only for bottom left plot of group (a 'group' is an analysis of identical markers).

☒ All plots are contour plots with outliers or pseudocolor plots.

☒ A numerical value for number of cells or percentage (with statistics) is provided.

## Methodology

| | |
|---|---|
| Sample preparation | Dissociation of organoids samples are described in the methods. |
| Instrument | BD FACSAria™ III |
| Software | FACSDivam FlowJo |
| Cell population abundance | Cell population abundance was indicated in Extended Data Fig. 2a. 2.5% of cells are positive for GFP. |
| Gating strategy | FSC/SSC were used for identify cell population. GFP, Tomato and Alexa 700 were used to select cell populations for scRNA seq. |

☒ Tick this box to confirm that a figure exemplifying the gating strategy is provided in the Supplementary Information.

# Magnetic resonance imaging

## Experimental design

| | |
|---|---|
| Design type | Cross-sectional fetal MRI |

| Design specifications | Patients undergoing fetal MRI were subjected to two fetal MRI scans during the second and third trimester. MRI was performed without sedation and without contrast media. |
|---|---|
| Behavioral performance measures | No behavioral parameters were evaluated. MRI data was used to calculate regional sub-volumes within the fetal brain throughout gestation. |

## Acquisition

| Imaging type(s) | Structural |
|---|---|
| Field strength | 1.5 and 3.0 T |
| Sequence & imaging parameters | Fetal MRI was conducted in accordance with the ISUOG guidelines of 2017. For fetal brain volumetry, T2-weigthed TSE sequences were used with a slice thickness of 3-4 mm, an echo time of 140 ms, and a field of view of 230 mm. |
| Area of acquisition | The fetus was investigated from head to toe, however in post-processing, only scans of the fetal head were analysed and quantified |

Diffusion MRI ☐ Used  ☒ Not used

## Preprocessing

| Preprocessing software | Horos, ITK-SNAP |
|---|---|
| Normalization | Segmentation of cerebral regions of interest was performed by nonrigid mapping of a publicly-available, spatiotemporal, anatomical fetal brain atlas for each individual case (1). To account for inaccuracies in ultrasound-based estimation of the exact date of conception as well as individual variability in neuronal development, we also included atlases that covered the prior and consecutive weeks of estimated gestational age for each case and merged them using a label fusion technique(2).<br><br>1. Gholipour A, Rollins CK, Velasco-Annis C, Ouaalam A, Akhondi-Asl A, Afacan O, et al. A normative spatiotemporal MRI atlas of the fetal brain for automatic segmentation and analysis of early brain growth. Sci Rep. 2017 Mar 28;7(1):476.<br>2. Wang H, Suh JW, Das SR, Pluta JB, Craige C, Yushkevich PA. Multi-Atlas Segmentation with Joint Label Fusion. IEEE Trans Pattern Anal Mach Intell. 2013 Mar;35(3):611–23. |
| Normalization template | See above. |
| Noise and artifact removal | The investigated woman was instructed with respiratory commands during the MRI scan. Additionally, during post-processing, data was denoised using the methodology described by Coupe P et al.<br><br>Coupe P, Yger P, Prima S, Hellier P, Kervrann C, Barillot C. An optimized blockwise nonlocal means denoising filter for 3-D magnetic resonance images. IEEE Trans Med Imaging. 2008 Apr;27(4):425–41. |
| Volume censoring | Automated brain masking was conducted as previously described by Ebner M et al.<br><br>Ebner, M. et al. An automated framework for localization, segmentation and super-resolution reconstruction of fetal brain MRI. Neuroimage 206, 116324 (2020). |

## Statistical modeling & inference

| Model type and settings | Cross-sectional brain volumes of one patient were extracted and analyzed. No additional statistical analysis was performed. |
|---|---|
| Effect(s) tested | NA. |

Specify type of analysis: ☐ Whole brain  ☒ ROI-based  ☐ Both

Anatomical location(s)  *Describe how anatomical locations were determined (e.g. specify whether automated labeling algorithms or probabilistic atlases were used).*

| Statistic type for inference<br>(See Eklund et al. 2016) | Voxel-wise analysis and calculation of respective volumes in mm^3. |
|---|---|
| Correction | NA. |

## Models & analysis

| n/a | Involved in the study |
|---|---|
| ☒ ☐ | Functional and/or effective connectivity |
| ☒ ☐ | Graph analysis |
| ☒ ☐ | Multivariate modeling or predictive analysis |

