## [Peer Review File · Nature]

Manuscript Title: Single-cell brain organoid screening identifies developmental defects in autism

Reviewer Comments & Author Rebuttals

Reviewer Reports on the Initial Version:

Referees' comments:

Referee #1 (Remarks to the Author):

Here, Li and Fleck et al. employ brain organoids and functional genomics to tackle a critical gap in knowledge of our understanding of Autism Spectrum Disorder (ASD) - how do genetic risk factors driver pathology during human neurodevelopment? First, they develop and optimize a CRISPR/Cas9 screening platform for efficient knockout of risk genes in mosaic organoids modeling ASD relevant brain regions, of which they dub the CRISPR-human organoids-scrRNA-seq method (CHOOSE). Utilizing this method, they target 36 known ASD-risk genes in a loss-of-function screen and subsequently employ single cell multi-omics to characterize: 1) the effects of these perturbations on cell fate determination; and 2) differential developmental progression across pseudo-time trajectories. Following this, with Gene Regulatory Network (GRN) analysis via use of their tool Pando, they find that many risk genes cause dysregulation of shared disease-relevant pathways, recapitulating much known biology in the field. Their main findings include the casual depletion of IPCS, enrichment of Radial Glial Cells, and more prominently the significant role that ARID1B loss-of-function plays in these pathological changes. In my opinion, the major strength of the paper comes from the experiments shown in Figure 4 where the authors demonstrate a recapitulation of their ARID1B hypothesis in patient-derived iPSC lines and a reversal of the phenotype with mutation correction. I find this result to be individually compelling and I commend the authors for performing this experiment and including the corrected mutation as a negative control. However, while I feel that the authors have identified an important problem and employed interesting and cutting-edge methods, I have significant reservations with respect to the logic of the study and the relevance of the system to human disease. The novelty of both their CHOOSE technique and the biological insight to ASD causal drivers appears questionable. Compounding this, I find the organization and presentation of data to be unintuitive and feel that the authors have overstated the impact of the work in certain places.

I have outlined my suggestions for improvement below, highlighting both major and minor suggestions.

Major suggestions:

- 1) The title, claiming to identify the "origins of autism" is very overstated.
- 2) The authors focus on 36 high-risk ASD genes but do not include any real controls in this set. Many of these genes are epigenetic regulators with broad activities across nearly every biological function. Because no controls are included that target similar genes that have not been implicated in ASD, its challenging to understand whether the results obtained are related to ASD or just the consequence of knocking out important genes. A cynical reader might expect that knockout of these genes in any culture system would produce marked effects. At least in other systems (for ex. the hematopoietic system) it is well understood that perturbation of epigenetic regulators leads to defects in differentiation.

3) My understanding is that the vast majority if not all of the genes being studied are mutated in ASD causing haploinsufficiency. To me, there is a big difference between the consequences of loss of one copy of a gene versus complete knockout. Since I do not believe the authors are able to determine the consequences of the sgRNAs on each individual cell, it seems impossible to know what percent of cells are null and what percent have only lost one copy. My intuition is that most cells with successful editing would be null which complicates the interpretability and disease relevance of the results. This causes major issues for the interpretation of these results and the extrapolation of these results to the human condition.

4) I'm not convinced that the CHOOSE system is novel. It is my understanding that this system was nearly fully developed previously (by the authors – PMID 33122427) and that the primary novelty is the addition of a second guide. If the authors disagree with this interpretation, then I would encourage them to dedicate time in the manuscript to describing how the CHOOSE system is important and novel in the context of previously established CRISPR mosaic tools in organoids.

5) In the introduction, the authors highlight the “phenotypic variability” of brain organoids (Line 58). I agree that this is a substantial limitation in the field but I am not convinced that this variability has been addressed or properly controlled for. While the authors demonstrate that a homogeneous distribution of gRNAs is maintained from plasmid, to hESCs, to embryoid bodies (Fig 1d) they do not show any such analysis at the level of the organoids. Ext. Data Fig. 5e shows that the distribution of gRNAs in the 8 libraries is very biased which has important implications for all of the downstream analysis. Moreover, each library is comprised of multiple organoids (3 to 4 on average) and it's impossible to know how many of the cells in a given library were derived from a single organoid (i.e. jackpotting of results). If I had to guess, fewer than 100 cells were captured for most gene targets per library. At that level, especially spreading the results across many cell types, I worry that the results are not robust. As best I could tell, the raw and processed data was not made available to reviewers and the manuscript did not include a table that shows how many cells mapped to each gene target. Though admittedly the resolution of the figure is not great, it seems that, even for ARID1B (a key focus of the manuscript), most organoids have very few representative cells and approximately half of the total ARID1B cells come from a single library and potentially even a single organoid given the points above. Thus, the phenotypic and cellular variability of cerebral organoids seems like a challenge that has not been adequately addressed.

6) Some key figures, for example Figure 2a,d,g show only the significance of a result and do not show the magnitude of the result. In my mind, this would be akin to showing a volcano plot without the x axis to show the fold change of the comparison. This seems to severely limit my ability to interpret the results.

7) The differential expression analyses (Fig. 3) are performed at the level of dorsal and ventral. Given that the authors have already shown that there are clear differences in cell type abundance within these categories, any DE analysis will likely be capturing those differences in cell type abundance rather than any difference in actual gene expression.

8) Why weren't the patient-derived organoids additionally subjected to single cell multi-omics? Instead 4g shows that they were only analyzed via IF. Why not recapitulate these loss-of-function experiments with ARID1B organoids they performed previously with actual patient derived mutations?

9) Aside from the GRN, there is almost no utilization of the multi-omic nature of the data. There is also essentially no QC information presented on the multi-omic data.

10) It appears, at least in my reading of the manuscript, that the SAG and IWP2 stimulation protocol used to generate the patient-derived organoids is different than the protocols used earlier in the paper. If not, this should be described earlier in the paper; if so, this should be justified as to why it was changed.

Minor suggestions:

11) How does 4 months of organoid development in vitro track with in vivo human development? What is known about ASD changes in the brain at this stage? Presumably something is known and this is highly relevant to this study if the authors are trying to convince us that these developmental changes are remotely biological and not artificial. Relatedly, it would be helpful if the authors could cite literature supporting the idea that ASD starts in utero.

12) Line 57 – The authors fail to acknowledge in the introduction that cerebral organoids do not produce the diversity of cell types found in the developing brain. Instead, the introduction makes it seem as if organoids are equivalent to little mini brains. However, in reality they lack many cell types and instead only include those from the RGC lineage really. Given that the first wave of entry of microglia into the developing brain precedes the peak of neurogenesis and the organoids do not have microglia, this seems like an important caveat to discuss.

13) The introduction or perhaps the discussion could benefit from more information on what is known about the role of the BAF complex in ASD.

14) Figure 1e – The title of this plot is not very clear and the authors need to add a Y axis. Also the authors do not discuss what “Control” in this instance?

15) Lines 157-160: I think it is an overstatement to attempt to connect perturbation of cell type proportions with a ASD symptom; this could be and likely is caused by much more complex biology and physiology.

16) Fig. 2b,e – I don't find this presentation to be useful. I am not able to interpret this figure.

17) I was not immediately convinced by the various enrichment analyses testing for enrichment in the SFARI database because the authors did not mention what the background set of genes was. I initially assumed the authors were using the whole transcriptome as the background but the methods show this is not the case. I would state the background set of genes in the main text.

18) Extended Data Figure 6 is nearly uninterpretable at the current scale and lacking labels of cell types/trajectories in the plots

19) The authors should at least discuss the caveat that these mosaic screens allow for but cannot assess effects caused by heterogeneous cell-to-cell interactions that might (for ex.) lead a cell with one knockout to affect a neighboring cell.

20) The prenatal imaging data in Fig 4h is a unique and interesting addition but I struggled to interpret the biological relevance of these figures and was forced to take them at face value with the explanation from the text.

21) The model shown in Fig. 4i is not very clear. The differences between the WT and ARID1B perturbation (arrow size, oval size) are not striking enough to draw my eye for immediate interpretation of the meaning.

Referee #2 (Remarks to the Author):

The work by Li et al. is timely and important for advancing our understanding of the genetic and developmental architecture of autism spectrum disorders (ASD).

In particular, leveraging CRISPR-based perturbations, brain organoids and single cell analysis, the authors set out to identify cell type-specific transcriptional endophenotypes as well as the impact on neurodevelopmental trajectories and gene regulation of the loss-of-function of 36 high-risk ASD genes. As a next step, they related the impact observed in the screening of ARID1B knock-out to the defects observed in cerebral organoids from iPSC of ASD patients harboring heterozygous ARID1B mutations.

These results are thus potentially very relevant, both for shedding new light on the neurobiological basis of ASD and for the novelty of the technological implementation of CRISPR-based perturbation in cerebral organoids. Unfortunately, however, major flaws in the experimental design and in the analytical strategy do not allow to support the key knowledge claims of the paper, at least in its current state.

A thorough revision of the design is thus necessary to tackle the major concerns on the validity and accuracy of the findings, so as to render the work both robust and reproducible and meet the ambitious and laudable goal it set out to achieve.

Below is a list of the key issues, in the hope that it will be useful to guide authors in the reassessment of their strategy and the implementation of the new experiments.

Major concerns:

- The numbers of cells for each perturbation (Extended Data Figures 5, in particular panel e, and 6) show a heavy imbalance: few perturbations in the mosaic organoids are over-represented, while several others are under-represented (including control cells that are under-represented or absent in several of the 8 libraries). In particular, a main concern emerges from the fact that the 2 perturbations with the highest number of recovered cells (KMT2C and LEO1) are differentially enriched, respectively, in the dorsal and ventral regions of the UMAP. This is a major issue because the over-representation of cells mutated for these 2 genes is in all likelihood the main driver of the observed dimensionality reduction where the authors identify the 2 regions of dorsal and ventral cell populations. Thus, a UMAP computed only on control cells, as well as a UMAP where the dataset is downsampled to have the same number of cells per perturbation, are both necessary to properly interpret the

validity of cell type annotation and, as a consequence, of all the downstream analysis on the effects of the individual perturbations.

The imbalanced representation of perturbations, together with the impossibility in a mosaic design to discern cell-autonomous from non-cell-autonomous effects derived from the interplay of different perturbations, require the main results of the study to be replicated with new experiments. As minimal requirement, for genes showing a significant change of cell abundance, the results should be validated in homogeneous (ie. non mosaic) organoids carrying only that specific perturbation; this would allow both to confirm the observations with a larger number of cells, and to discern cell autonomous effects from potential non-autonomous ones. Given the ambition of the work, and the scalability of the employed assays, this approach should ideally be employed for all the 36 genes, and it appears somewhat surprising why, also in light of the effective gRNA construct

design that was chosen, the systematic assay of all individual perturbations separately was not adopted as the baseline. When growing new batches of organoids, one suggestion to back the idea that perturbations are selected would be to track in time, at known crucial stages of organoid generation (or every 15/30 days), gRNA compositions of small pools of organoids (e.g. 3/5 organoids per pool, 3/5 pools per time point).

The experimental design followed by the authors is based on the pooling of 3-6 organoids for each of the 8 chromium controller runs. There are thus no independent replicates that can be used in the analysis to evaluate the reproducibility of the observed phenotypes. Indeed, it is truly remarkable that the word “replicate” never appears in the paper. All results in figures 2 and 3 (and associated extended data figures) about the effects of the perturbations in terms of ratio of dorsal to ventral cell populations, specific enrichment/depletion for a particular cell type, and differential densities along pseudotemporal axis need replicates to be interpreted.

- The two above issues imply that it is not possible to evaluate the validity and accuracy of all the main results of the paper: figure 2 and 3.
- The authors presented an elegant and novel strategy to generate efficient and controlled pooled systems with high clonal complexity in pluripotent stem cells, not in organoids as they claim, since there is no data showing the unique clone barcode distribution after differentiation of the organoids. This is instead a result of obvious relevance that should be shown.
- Results from differential abundance analysis should be strengthened by applying algorithms that have been developed for scRNASeq frameworks such as scCODA that account for the inherent bias present in cell-type compositions with a Bayesian approach for cell-type composition to address the low replicate issue, and/or MILO, thus does not rely on discrete clusters as input when testing for differential abundance among experimental conditions, thus improving the detection of differences in continuous trajectories. In figure 2A, it would be useful to have relevant about the magnitude of the enrichment in addition to the significance.
- For the genes that show an impact on the abundance of specific cell populations, it would be interesting to check if the detected effect is coherent with the gene expression patterns across cell types in the organoid model in study as well as in datasets from fetal brain at comparable stages.
- Have the expression levels of targeted genes been checked in the results of the differential expression analysis? For each guide the expression level of the target gene should be shown for the perturbed cells vs the others. This would be instrumental also for assessing the relevance of the dosage difference between the ARID1B knock-out and the ARID1B haploinsufficient backgrounds, and hence properly evaluating this validation aspect of the work (Fig. 4).
- From the Manhattan plot in Figure 2, the number of DEGs is quite low for most of the examined conditions. This is quite puzzling when compared to the results about cell type abundance. How do the authors explain this striking discrepancy?
- Is the statement about the involvement of the ubiquitin-proteasome pathway substantiated by a functional enrichment analysis or other analytical techniques?
- The CHD8 perturbation does not show the impact on cell population abundance that was reported by several papers in similar organoid models and in vivo (e.g Paulsen et al. 2022; Villa et al. 2022; Jin et al. 2020). What is the explanation for this (which should anyway be discussed in the paper)?
- Finally, most observations are solely grounded on differential expression analyses with no significant validation of at least key findings. At least basic biochemical assays are required, for specific perturbations, to corroborate/support claims and observations such those on mitochondrial and proteasome pathways dysregulation.

Referee #3 (Remarks to the Author):

The Manuscript entitled "Single Cell CRISPR Screening in Organoids Identifies the Origins of Autism" submitted by Li and colleagues aims to interrogate the neurodevelopmental consequences after perturbation of 36 Autism Spectrum Disorder (ASD) risk genes. The authors developed a pooled loss-of-function CRISPR screening in system mosaic organoids, allowing for the simultaneous analysis of all perturbed genes within a single organoid. Using a single-cell multi-omics approach the authors discovered dysregulation of several developmental pathways following perturbation, mainly converging on the developmental trajectories of dorsally derived intermediate progenitor cells and ventrally derived radial glia. This effect was most obvious within cells that received the loss of function perturbation for ARID1B, a member of the BAF chromatin remodeling complex. Further analysis of the single-cell data set revealed a propensity of these cells more frequently giving rise to oligodendrocyte progenitor cells (OPCs) as opposed to neurons. This finding was supported by the generation and analysis of ARID1B patient-derived ventrally patterned organoids which showed a greater expression of Olig2 at DIV 40. This was further validated by an in utero MRI scan of the ARID1B patient which allowed for volumetric measurement of the CGE and LGE, two germinal zones that give rise to OPCs. Overall the manuscript has the merit of focusing on an approach that aims at tackling the critical question of whether ASD converges on the development of specific cell types during fetal development. However, while the aspirational goal of this study is very relevant to the field, the current execution of this idea has significant flaws that limit the relevance and accuracy of the results. Therefore, the general sense is that the idea is interesting, but that the results should be validated in at least another line with different genetic backgrounds and in NON-mosaic organoids where separate genes are downregulated in individual organoids.

General Notes

- The title of the paper is exaggerated and inaccurate.
- There are big concerns about the fundamentals of the pooled CRISPR screen method - perturbing so many genes within a single system, will lead to significant crosstalk and feedback between populations. Observed phenotypes may not be the direct result of the mutation but a downstream effect. It is incredibly difficult and inaccurate to parse apart the direct cause and effect relationships in this system.
- The timepoints shown for the organoid experiments are not consistent (single-cell at 4 months - immunohistochemistry at day 40, organoid validation up to 27 days)

Figure 1

- It seems odd that despite the generation of LGE, CGE, and dorsal forebrain there was no generation of MGE which is a major contributor to interneuron populations
- There is no proteomic validation of the knockouts, this is a necessity.
- These mutations in patients are mainly heterozygous, the mutations used in the screen are homozygous. The findings from this screen may then have been largely exacerbated by this genetic difference.
- There is no clonal bias at the iPSC stage, but in supplemental fig. 5 and 6 a clear bias in the screen is seen.
- There is no reference to the clonal barcode - is one clone contributing more than others?

-There is a confounding variable of neighboring knockout cells influencing the differentiation trajectory of each other.

- The screen should be repeated in other lines to determine if any effects are cell line-specific and background specific.

Figure 3

-Showing dysregulation of gene expression and regulatory networks after genetic perturbation of key regulating genes seems redundant and uninformative.

Figure 4

-There are weak evidence showing transitional changes at the day 40 timepoint while all other experiments were done at 4 months.

-The increased number of cells could be due to increased vulnerability of INPs (ie. cell death), so potentially no cell fate decision is affected.

-There is no reference to the Paulsen et al. Nature 2022 that shows the same phenotype in ARID1B line, using single-cell rna-seq (stronger evidence compared to only using 2 markers).

-The data obtained from in utero MRI's is interesting but the LGE/CGE volume needs to be normalized to total brain volume in order to be relevant.

-It is again surprising that no MGE was produced using the Bagley 2017 protocol which was shown to robustly generate this region. There is no explanation for this in the manuscript.

We would like to thank all the reviewers for their thoughtful and constructive comments and suggestions. We have now followed the recommendations and performed comprehensive revision experiments to address all the comments and concerns, which have further substantiated the significance of our findings. Major experiments include:

- 1) Replication of the entire single-cell perturbation screening on 4-month-old brain organoids with two additional batches;
- 2) Re-analysis of all scRNA-seq experiments with newly integrated datasets on cell type-specific abundance, differential gene expression and genetic regulatory network inference;
- 3) Systematic determination of CHOOSE screening system on gRNA representation and clonal diversity;
- 4) Validation of key findings for perturbations that lead to overall cell abundance changes, depletion of intermediate progenitors or enrichment of interneuron precursors by generating individual non-mosaic brain organoids for each gene.

Together with all other experiments, we have generated a substantial amount of new data, including **43 new data panels** and **15 updated data panels**, which resulted in **4 updated main figures**, **6 updated Extended data figures** and **7 completely new Extended data figures**. We have also included **5 reviewer figures**.

As the revision is extensive, here we include a summary list of data figure changes presented in the revision (**Table 1**), and a detailed point-by-point response to all the issues raised. Major revisions in the results, methods and figure legends sections are marked in blue font to highlight the new data.

Table 1: Summary list of changes to the figures.

Figures	Changes/ Additions	Experiments	Responding to Reviewer
Figure 1	1f, g	Replication of the entire screen; Integration of new data set; Annotation computed on control cells.	Reviewer 2
Figure 2	2a, b (new)	gRNA representation analysis; Cell proliferation and depletion.	Reviewer 1, 2
Figure 2	2c	Repeat cell type abundance testing with newly integrated dataset	Reviewer 2
Figure 3	3a, b, c, d (new), e, f, g	Repeat differential expression gene analysis; Repeat GRN analysis; New GO enrichment analysis.	Reviewer 2
Figure 4	4h	Update GE measurement	Reviewer 3
Figure 4	4i	Modify working model	Reviewer 1

Ext. Data Fig. 2	New	Protein expression of perturbed genes	Reviewer 3
Ext. Data Fig. 4	c-h	Clone complexity in mosaic organoids	Reviewer 1, 2, 3
Ext. Data Fig. 5	d-f	eCas9 induction in mosaic organoids (cell-cell interaction)	Reviewer 1, 2, 3
Ext. Data Fig. 6	a-g	Repeat, integration and update of scRNAseq analysis	Reviewer 2
Ext. Data Fig. 6	i, j	ASD risk gene expression pattern in organoids and in vivo	Reviewer 1, 2
Ext. Data Fig. 7	New	gRNA representation in mosaic organoids	Reviewer 1, 2, 3
Ext. Data Fig. 8	New	Validation of depletion and over-proliferation phenotypes in organoids with individually perturbed genes	Reviewer 1, 2
Ext. Data Fig. 9	a, b	Integration and update of local cell composition analysis	Reviewer 1
Ext. Data Fig. 10	New	Consistency testing of scRNA-seq library replicates	Reviewer 2
Ext. Data Fig. 11	New	Validation of intermediate progenitor phenotypes in organoids with individually perturbed genes	Reviewer 2
Ext. Data Fig. 12	New	Validation of ventral progenitor cell phenotypes in organoids with individually perturbed genes	Reviewer 2
Ext. Data Fig. 14	New	DEG and GO enrichment analysis	Reviewer 2
Ext. Data Fig. 15	b-d	scMulti-omic QC	Reviewer 2
Ext. Dat Fig. 16	c-g	GE measurement and normalization	Reviewer 3
Reviewer Fig. 1		scDNA seq	Reviewer 1
Reviewer Fig. 2		ARID1B perturbed cells	Reviewer 1
Reviewer Fig. 3		scRNAseq of ARID1B patient organoids	Reviewer 1
Reviewer Fig. 4		Cell abundance testing using MILO	Reviewer 2
Reviewer Fig. 5		DE analysis of perturbed genes	Reviewer 2

Point-to-point response to reviewers' comments

Reviewer #1 (Remarks to the Author):

Here, Li and Fleck et al. employ brain organoids and functional genomics to tackle a critical gap in knowledge of our understanding of Autism Spectrum Disorder (ASD) - how do genetic risk factors driver pathology during human neurodevelopment? First, they develop and optimize a CRISPR/Cas9 screening platform for efficient knockout of risk genes in mosaic organoids modeling ASD relevant brain regions, of which they dub the CRISPR-human organoids-scRNA-seq method (CHOOSE). Utilizing this method, they target 36 known ASD-risk genes in a loss-of-function screen and subsequently employ single cell multi-omics to characterize: 1) the effects of these perturbations on cell fate determination; and 2) differential developmental progression across pseudo-time trajectories. Following this, with Gene Regulatory Network (GRN) analysis via use of their tool Pando, they find that many risk genes cause dysregulation of shared disease-relevant pathways, recapitulating much known biology in the field. Their main findings include the casual depletion of IPCS, enrichment of Radial Glial Cells, and more prominently the significant role that ARID1B loss-of-function plays in these pathological changes.

In my opinion, the major strength of the paper comes from the experiments shown in Figure 4 where the authors demonstrate a recapitulation of their ARID1B hypothesis in patient-derived iPSC lines and a reversal of the phenotype with mutation correction. I find this result to be individually compelling and I commend the authors for performing this experiment and including the corrected mutation as a negative control.

However, while I feel that the authors have identified an important problem and employed interesting and cutting-edge methods, I have significant reservations with respect to the logic of the study and the relevance of the system to human disease. The novelty of both their CHOOSE technique and the biological insight to ASD causal drivers appears questionable. Compounding this, I find the organization and presentation of data to be unintuitive and feel that the authors have overstated the impact of the work in certain places.

I have outlined my suggestions for improvement below, highlighting both major and minor suggestions.

We thank the reviewer for recognizing the significance of our study and the positive comments on the ARID1B characterizations. We have now followed the reviewer's suggestions and addressed each comment with additional experiments and data (see below). As a result, we hope you agree that the manuscript is greatly improved and strengthened.

Major suggestions:

1) The title, claiming to identify the "origins of autism" is very overstated.

We appreciate the reviewer's criticism. We have changed the title to '*Single-cell brain organoid screening identifies developmental defects in autism*'.

2) The authors focus on 36 high-risk ASD genes but do not include any real controls in this set. Many of these genes are epigenetic regulators with broad activities across nearly every biological function. Because no controls are included that target similar genes that have not been implicated in ASD, its challenging to understand whether the results obtained are related to ASD or just the consequence of knocking out important genes. A cynical reader

might expect that knockout of these genes in any culture system would produce marked effects. At least in other systems (for ex. the hematopoietic system) it is well understood that perturbation of epigenetic regulators leads to defects in differentiation.

We thank the reviewer for bringing up this interesting point. We agree that epigenetic regulators (19 genes included in the screen) are important for many biological functions. On the other hand, we would also like to emphasize that epigenetic regulators can cause very specific disease phenotypes and that their analysis in organoids can reveal specific functions as exemplified by a beautiful recent study from the Arlotta lab (Paulsen et al., Nature 2022). Epigenetic regulators have been implicated in diverse yet specific processes including neural progenitor proliferation, neural and glial cell fate determination, as well as neuronal maturation¹. The fact that chromatin genes are among one of the three major groups of genes specifically associated with ASD^{2,3} intrigued us to systematically test and compare them in one system at the same time.

The CHOOSE system uses single cell transcriptomes as a high-content readout to dissect molecular changes in a cell-type specific manner. We want to emphasize that our findings do not necessarily suggest a common defect in differentiation, but rather we identified specific cell types that are more vulnerable to certain genetic perturbations (New Fig. 2c). For example, *KMT2A* perturbation causes an enrichment of L2/3 excitatory neurons (EN); and *POGZ* perturbation leads to an enrichment of radial glial (RG) cells. Meanwhile, some perturbations lead to changes of broader cell types, such as *KMT2C* perturbation which causes depletions of multiple dorsal cell populations. While perturbations of several genes, such as *KDM6A*, lead to an overall cell depletion without affecting cell differentiation.

To further investigate the molecular changes, we performed differentially expressed gene (DEG) analysis followed by new GO term enrichment analysis (**New Fig. 3a-c, New Extended Data Fig. 14a, b**). We now identified specific molecular pathways that are dysregulated in several perturbations. Thus, despite the expectation that epigenetic regulators have broad activities, we are very excited to see that these ASD-associated epigenetic regulators act at different and specific processes to govern proper brain development. In addition, we have checked the expression patterns of all perturbed epigenetic regulators in brain organoids (**New Extended Data Fig. 6i**). Interestingly, we found that several genes do have cell-type specific enrichment of expression (e.g., *KMT2C*, RG cells; *BAZ2B*, intermediate progenitors (IP); *BCL11A*, L2/3 EN; and *KAT2B*, ventral radial glia (vRG) cells), which further support a cell type-specific requirement of these epigenetic regulators during brain development.

We agree it will be interesting to test another set of epigenetic regulators which do not cause ASD but only other forms of neurodevelopmental disorders. Unfortunately, such a gene list is not currently available. In addition, many genes that cause ASD are also implicated in other defects like intellectual disability. Prominent examples include *MECP2*, which causes Rett syndrome, but also *ARID1B*, *CHD2*, and *CHD8*. Additionally, we cannot find an epigenetic gene set to use as ‘controls’ with a statistical power for testing, since such neurodevelopmental disorders-associated genes should not cause ASD, or any differentiation phenotypes or molecular changes as we observed in our screen. We hope that with the rapid expansion of gene discovery studies, such a gene list could emerge in the near future and the CHOOSE system would then be an ideal platform to test and compare a complete list of epigenetic regulators associated with human diseases.

We have now added new data including the expression patterns of our targeted genes in organoids (**New Extended Data Fig. 6i**), perturbation induced GO term enrichment analysis (**Fig. 3c, Extended Fig. 14b**), and a new paragraph in the discussion (Page 12, highlighted in blue) to further emphasize the insights provided by our findings.

3) My understanding is that the vast majority if not all of the genes being studied are mutated in ASD causing haploinsufficiency. To me, there is a big difference between the consequences of loss of one copy of a gene versus complete knockout. Since I do not believe the authors are able to determine the consequences of the sgRNAs on each individual cell, it seems impossible to know what percent of cells are null and what percent have only lost one copy. My intuition is that most cells with successful editing would be null which complicates the interpretability and disease relevance of the results. This causes major issues for the interpretation of these results and the extrapolation of these results to the human condition.

We thank the reviewer for bringing up this important point. The reviewer is correct that from current medical genetic studies, many ASD patients identified with mutations are determined to be heterozygous. It is worth noting, though, that homozygous genetic mutations have also been documented in ASD patients⁴⁻⁹. Recessive inheritance patterns are not always captured in ASD, probably due to the lethal effects of complete protein loss and detrimental consequences in multiple organs. This has been largely supported by ASD rodent models, where a full knockout of an ASD gene does not produce a viable animal. In fact, functional studies of ASD genes were performed not only in heterozygous animals but also complemented by conditional homozygous KO in brain tissues.

The concern stems from the possibility that a homozygous mutation might have different molecular and cellular effects than heterozygous ones. However, in some cases, the same mechanisms are affected but become exacerbated in the homozygous condition due to further depletion of functional proteins. Both scenarios can occur, although the latter is more likely when genes function in a dose-dependent manner. Most mutations for the ASD genes chosen for our screen are indeed protein damaging loss-of-function mutations acting in a haploinsufficient manner¹⁰. In such cases, the homozygous condition might even provide greater power to observe the same cellular and molecular effects than the heterozygous, which is the strength of CRISPR KO screens.

Consistent with this expectation, heterozygous and homozygous mutants do present the same disrupted cellular and molecular processes in many cases with more severe phenotypes observed in the homozygous context. Below we list some key findings made from functional studies of several prominent ASD genes.

TBR1: *Same phenotypes observed in both heterozygous and homozygous mutant animals at molecular, cellular and neuronal activity level.* It has been shown¹⁰ that both heterozygous and homozygous KO of *Tbr1* lead to the dysregulation of the same regulators/markers of layer 6 EN, including *Nr4a2*, *Wnt7b* and *Bcl11a*. Dendritic morphology and synaptic density were also affected in the same manner in both heterozygous and homozygous mutants. More importantly, the same phenotypes were observed even at the neuronal activity level, including abnormal spontaneous excitatory/inhibitory postsynaptic currents.

FOXP1: *ASD-like behaviors were only observed in homozygous, but not in heterozygous *Foxp1* mutant mice.* Mice with brain-specific homozygous *Foxp1* KO present hyperactivity, impaired short-term memory, increased repetitive behavior, impaired social behavior, as well as reduced anxiety¹¹. Surprisingly, *Foxp1* heterozygous mice do not present any deficits in all behavior tests performed.

ARID1B: *Same altered gene families observed in both *Arid1b* heterozygous and homozygous mutant mice.* It has been shown cortical interneuron development was disrupted in *Arid1b* heterozygous mutant mice¹². Consistently, β -catenin, β -catenin target genes, H3K9ac and several synaptic molecules were all dysregulated in the same manner in both heterozygous

and homozygous KO mice. Especially, homozygous mice show larger expression fold changes compared to heterozygous.

These data, together with many more examples (*CHD8*^{13,14}, *MECP2*^{15,16}) suggest that homozygous models in many cases provide insightful information for ASD gene functions and the phenotypes observed in these models can be highly relevant to ASD pathology. Phenotypes can often be mild in heterozygous mutants and require experimental approaches with higher sensitivity and detection power. Thus, we do not believe generating homozygous KO is an issue in our system. Instead, similar to the conditional knockout situation, we believe our system offers a unique opportunity to analyze their phenotypic changes in homozygous mutant cells if they exist.

Single cell DNA sequencing:

Nonetheless, we share the same interests with the reviewer in finding out the ground truth of the zygosity of each single perturbed cell. We turned to a recently established, **microfluidic-based Tapetri system (Mission bio) to genotype single cells (Reviewer Fig. 1a)**.

Reviewer Fig. 1 scDNAseq of CHOOSE mosaic organoids

a, Workflow of scDNAseq using the Tapetri platform. **b**, H9 cell line-specific SNP recovery. **c**, ADO frequencies calculated using H9 cell line-specific SNPs. ADO = cells with 1 allele recovered/cells with 1 and 2 alleles recovered. **d**, gRNA editing outcomes of two targeted loci for each gene. Rarely, a third sequence was recovered, might be due to doublets. **e**, Percentage of cells with confident KO for each perturbation.

We first designed an oligo panel including paired primers to amplify 72 gRNA targeting loci (two gRNAs for each gene) and primers to amplify gRNA sequences from genomic DNA.

CHOOSE mosaic organoids were dissociated and GFP+/dTomato+ cells were sorted and subjected to the Tapestry pipeline, followed by simultaneous amplification of gRNA targeted loci from both alleles within each cell using barcoded primers. The amplicon library was then subjected to NGS and bioinformatic analysis. Ideally, with this experiment we could recover each loci from both alleles to evaluate editing outcomes and zygosity. However, when using a few available H9 cell line-specific SNPs to estimate the quality of the allele recovery frequency, we found that the recovery rates varied at different loci. For example, we recovered chr7_55181370 (SNP) information in less than 50% of the cells (**Reviewer Fig. 1b**). Additionally, we found that the allele dropout (ADO, ADO = cells with 1 allele recovered/cells with 1 and 2 alleles recovered) frequencies range from 20% to 80% (**Reviewer Fig. 1c**). For these two reasons, we could not confidently define the zygosity for every locus.

We could however analyze the editing outcomes from the recovered regions. For the majority, we identified at least one allele that was successfully edited (**Reviewer Fig. 1d**). We calculated the confident homozygous knockout percentages (2 mutant sequences for at least one locus for each gene) (**Reviewer Fig. 1e**) and observed an average of 50% homozygous KO, although this varies substantially for each perturbation (**Reviewer Fig. 1e**). Thus, we decided to not include this data in the manuscript, as we feel it does not provide a clear and complete picture of the hetero-/homozygosity of each perturbed cell, and may be confusing to the reader. We would be happy to include the data, however, upon the reviewer's explicit request.

CRISPR screenings have contributed enormously to biological discovery, biomedical research and drug development, although techniques for precisely generating heterozygous and homozygous mutations specifically are so far not available. We have included additional text in the **discussion** to emphasize the relevance of our studies and acknowledge the limitations of the technology.

Methods for scDNA-seq bioinformatic analysis. Raw reads were run through the Tapestry pipeline (Mission bio). Aligned sequences which are assigned to a cell were processed further. Target gene assignment to a cell was assessed from guide 1 and 2 amplicons analogous to scRNA analysis. CRISPRessoWGS predicted allele sequences for a 60 nt region around the expected cut sites of the cell's target gene¹⁷. Resulting sequences supported by more than 1 read were error corrected with genBaRcode (Levenshtein distance = 1, connectivity based)¹⁸. Allele sequences were filtered for amplicons with a read depth of at least 20x and an allele frequency higher than 10%. Full functional knockouts are cells with 2 mutated sequences in one of the target amplicons. Allele dropout rate was calculated on heterozygous SNPs of H9 cells (SRR6377128) overlapping the amplicons. Therefore, the cell genotype was derived from Tapestry prediction.

4) I'm not convinced that the CHOOSE system is novel. It is my understanding that this system was nearly fully developed previously (by the authors – PMID 33122427) and that the primary novelty is the addition of a second guide. If the authors disagree with this interpretation, then I would encourage them to dedicate time in the manuscript to describing how the CHOOSE system is important and novel in the context of previously established CRISPR mosaic tools in organoids.

We thank the reviewer for pointing this out and apologize for missing the opportunity to highlight the novelty of the CHOOSE system. The key difference between the CHOOSE system and the CRISPR-LICHT system is that **the CHOOSE system uses single cell transcriptomics as a phenotypic readout**. This was not possible in the CRISPR-LICHT

system which uses gRNA rank of the pooled cells as a readout to assess cell number changes. The CHOOSE system allows for simultaneous **detection of gRNAs that introduce a genetic perturbation** and the corresponding **transcriptomic profiles in single cells**. This provides a data-rich readout and is powerful to characterize complex biological phenotypes such as changes in cell types, cell states, as well as molecular pathways.

As described in the manuscript, to establish the CHOOSE system we engineered the 3' LTR of the lentivirus based on the CROP-seq design to directly read gRNA sequences from scRNAseq¹⁹. So far, CROP-seq or other single cell based perturb-seq has mainly been used in 2D culture systems. This can be challenging for 3D tissue CRISPR screening, as organoids undergo long-term differentiation, drastic epigenetic changes (which often causes lentiviral constructs silencing), and present diverse cell types. Our study demonstrated the feasibility of tissue-based single-cell perturbation screening and provides a complete framework including gRNA library design and delivery to stem cells, organoids culture, tissue preparation and single cell sequencing, and bioinformatic pipeline. Until the study presented here, human organoid-based screening with single-cell transcriptomic readout have not achieved similar resolution in terms of the differentiation length (4 months) and the complexity of the tissue (16 cell types covering diverse progenitors, neuronal and glial cell populations of the developing brain). We have now modified the text in the introduction to further highlight the novelty of the CHOOSE system (Page 2, 3). We also modified the workflow in **New Fig. 1a** to better reveal our experimental design.

5) In the introduction, the authors highlight the “phenotypic variability” of brain organoids (Line 58). I agree that this is a substantial limitation in the field but I am not convinced that this variability has been addressed or properly controlled for. While the authors demonstrate that a homogeneous distribution of gRNAs is maintained from plasmid, to hESCs, to embryoid bodies (Fig 1d) they do not show any such analysis at the level of the organoids. Ext. Data Fig. 5e shows that the distribution of gRNAs in the 8 libraries is very biased which has important implications for all of the downstream analysis. Moreover, each library is comprised of multiple organoids (3 to 4 on average) and it's impossible to know how many of the cells in a given library were derived from a single organoid (i.e. jackpotting of results). If I had to guess, fewer than 100 cells were captured for most gene targets per library. At that level, especially spreading the results across many cell types, I worry that the results are not robust. As best I could tell, the raw and processed data was not made available to reviewers and the manuscript did not include a table that shows how many cells mapped to each gene target. Though admittedly the resolution of the figure is not great, it seems that, even for ARID1B (a key focus of the manuscript), most organoids have very few representative cells and approximately half of the total ARID1B cells come from a single library and potentially even a single organoid given the points above. Thus, the phenotypic and cellular variability of cerebral organoids seems like a challenge that has not been adequately addressed.

We thank the reviewer for pointing this out. We have followed the reviewer's comment and now added a substantial amount of new data to test the robustness of our results.

1. gRNA representation analysis, with more replicates from independent batches.

- 1) We generated another 3 independent batches (**New Extended Data Fig. 7**, Batch 2-4) of 4-month-old organoids for scRNAseq (5-10k cells per library) or bulk analysis of gRNAs recovered from genomic DNA (50-150k cells per sample). In total, these experiments resulted in additional 6 scRNAseq libraries of eCas9-induced cells, 3 scRNAseq libraries of eCas9 uninduced cells from 4-OHT treated organoids, and 5 scRNAseq libraries of eCas9-uninduced cells from 4-OHT untreated organoids as an alternative control. Each library was prepared from an independent pool of organoids

as a replicate. The bulk analyses were performed on another 24 samples (sorted from 24 independent pools of organoids) from the three different conditions.

From both scRNAseq and bulk analysis, we consistently see over-representation of some gRNAs (e.g. *LEO1* and *KMT2C*) and under-representation of other gRNAs (e.g. *ADNP* and *WAC*). This was not the case in eCas9-uninduced cells as gRNAs are homogeneously represented in organoids with or without 4-OHT treatment (**New Extended Data Fig. 7**). In fact, the data suggest an over-proliferation or depletion phenotype. We thank the reviewer for pointing this out which led us to an exciting discovery of additional phenotypes from the screen (**New Fig. 2a, b**).

- 2) The over-proliferation/depletion phenotypes were further supported by additional 2 lines of experiments.

2.1) As also suggested by reviewer 2, we checked gRNA representation at different development stages (Day 20, 40, 60, 80) when growing a new batch of organoids (**New Extended Data Fig. 7b**). This has provided additional information on the over-proliferation/depletion phenotypes. For example, when comparing to uninduced cells sorted from the same pool, we found that percentages of some genes, such as *KMT2C* and *LEO1* started to increase already at day 20.

2.2) The phenotypes were also validated with a FACS-based quantitative approach by generating non-mosaic organoids for individual genes (**New Extended Data Fig. 8**) at two different time points.

2. Raw cell numbers from each library.

We replaced the low resolution of the figure on the cell numbers and have added a new table (**New Supplementary Data 1**) to list raw cell number data for each perturbation. For *ARID1B*, we made two plots (cell numbers and percentages recovered from 11 independent pools of organoids as replicates from 2 batches) which show that *ARID1B* perturbed cells were recovered with 1-4% of the total cells within each library (**Reviewer Fig. 2**).

Reviewer Fig. 2 Numbers and percentages of recovered *ARID1B* perturbed cells per library.

3. Statistic approaches to test the consistency of the cell type abundance changes

We have reanalyzed the data including 3 additional replicates from an independently grown

batch (in total 11 pools of organoids as replicates, 2 Batches) by testing (as we did previously) on a single cell level, using replicate as a covariate to control for biases stemming from differential cell numbers between replicates (**New Fig. 2c**).

Furthermore, this new data now allows us to perform an additional analysis to evaluate the consistency of effects across independent pools of organoids from 3 batches (in total 14 library replicates) to complement the analysis performed on the single-cell level (**New Extended Data Fig.10**). Here, we treated every pool of organoids as an independent sample (rather than every cell) and performed a t-test on the per-organoid pool fold enrichment of each guide. This approach can be used to evaluate the variability of the effect across library replicates from different batches, but it has less power compared to the testing on the single-cell level. We are glad to see that many of the effects highlighted in the study (e.g., depletion of IP in *KMT2C*, *BCL11A*, *CHD2*, *KDM5B* etc. perturbations, enrichment of INP in *CIC*, *IRF2BPFL*, *MED13*, *PHF3*, *TBL1XR1*, *TCF20* etc. perturbations, vRG enrichment in *ARID1B* perturbation) were significant also in this test, which supports their strong consistency across independently grown organoids. Although it is worth noting that if effects were detected on a single-cell level but not on the level of pools of organoids this might stem from higher variability of the effect across organoids pools but does not invalidate the effect seen on the single cell level.

Together, these newly added data and analyses have greatly supported the reproducibility of our findings and even led us to the discovery of new phenotypes from the screen. We are grateful to the reviewer's comment.

6) Some key figures, for example Figure 2a,d,g show only the significance of a result and do not show the magnitude of the result. In my mind, this would be akin to showing a volcano plot without the x axis to show the fold change of the comparison. This seems to severely limit my ability to interpret the results.

In the original manuscript we sought to highlight the most significant results and therefore chose to plot significance instead of fold change. We agree that adding a measure of effect size to the plot will enhance the readers ability to interpret the figures and have added it to all relevant plots, including **New Fig. 2b, c**, and **New Fig. 3a**.

7) The differential expression analyses (Fig. 3) are performed at the level of dorsal and ventral. Given that the authors have already shown that there are clear differences in cell type abundance within these categories, any DE analysis will likely be capturing those differences in cell type abundance rather than any difference in actual gene expression.

We agree with the reviewer that compositional changes can bias DE analysis. To mitigate this bias we have controlled for cell type as a covariate during DE analysis. As a result, the analysis should capture genes that are differentially expressed independent of cell type. We did not see many markers of differentially abundant cell types among the DEGs, which supports the effectiveness of this strategy.

8) Why weren't the patient-derived organoids additionally subjected to single cell multi-omics? Instead 4g shows that they were only analyzed via IF. Why not recapitulate these loss-of-function experiments with *ARID1B* organoids they performed previously with actual patient derived mutations?

We have followed the reviewer's comment and performed single cell RNA sequencing on 4-month-old organoids derived from the following 5 cell lines using the same protocol as the screen (**Reviewer Fig. 3**):

- 1) Two patient cell lines including Patient 1 (6q25.3del) and Patient 2 (c. 2201dupG).
- 2) The Patient 2 cell line with corrected mutation.
- 3) A new iPSC line from a healthy donor (HDon.1 ARID1B^{+/+}) and a new engineered ARID1B heterozygous mutant cell line using the same genetic background (HDon.1 ARID1B^{+/-}).

We consistently observed increased percentages of interneuron and OPC clusters across all three ARID1B mutant cell lines (**Reviewer Fig. 3c, d**), which is in agreement with our findings from the primary screen. Our results are also consistent with a recent study in which ARID1B mutant organoids generate increased interneuron (DLX2+) populations²⁰. All these data strongly support the robustness and reproducibility of our results. However, we feel such analyses have already been reported and we are concerned of the amount of space including them in our manuscript would occupy. We, therefore, propose to cite this study in our text and only include this scRNAseq dataset as a reviewer figure. In addition, we believe that subjecting these tissues to additional single-cell multi-omics experiments is far beyond the scope of the current study, and it would be perhaps more suitable to have a mechanistic follow-up study to thoroughly investigate the chromatin changes caused by ARID1B mutation instead.

Reviewer Fig. 3 scRNAseq of ARID1B patient and engineered iPSCs derived organoids.

a, UMAP embedding of the scRNA-seq dataset of brain organoids from 5 cell lines. **b**, Marker gene expression patterns. **c**, UMAP embedding separated by ARID1B wildtype and mutant cell lines. **d**, Percentages of INs and OPCs from ARID1B mutant and wildtype organoids.

9) Aside from the GRN, there is almost no utilization of the multi-omic nature of the data. There is also essentially no QC information presented on the multi-omic data.

We specifically collected multi-omic data with the goal to infer a GRN underlying neurogenesis at the presented developmental stage in organoids. GRN inference

encompasses a number of analysis steps that leverage joint RNA and ATAC measurements, including selection of candidate regulatory regions per gene, TF binding site prediction, and region-to-gene linkage. We did not present the results of each individual step, as is often done, since we felt that discussing the GRN as a holistic representation of these analyses was more meaningful and interesting in the context of our screen results. Furthermore, we apologize if the presented QC metrics were not sufficient to evaluate the quality of the data, we have added plots showing additional metrics, such as TSS enrichment, and fragment size distribution (**Extended Data Fig. 15 b-d**). We think these metrics show clearly that the multi-omic data is of high quality.

10) It appears, at least in my reading of the manuscript, that the SAG and IWP2 stimulation protocol used to generate the patient-derived organoids is different than the protocols used earlier in the paper. If not, this should be described earlier in the paper; if so, this should be justified as to why it was changed.

The reviewer is correct that SAG and IWP were used to enrich ventral telencephalon tissues according to previously published protocols^{21,22}. The reason for this is that we specifically wanted to examine the changes of the ventral progenitors (interneuron precursor cells and oligodendrocyte precursor cells) as observed in the screen. Thus, a ventralized protocol would be more suitable for us to check the behavior of these cell populations at an earlier stage. Without ventralization, ventral lineages are populated at a later stage in organoids²³. We have added **additional** justification in the text. We also wanted to highlight that even with the same protocol, the phenotypes were also confirmed using scRNA-seq at Day 120 as mentioned in **comment 8**.

Minor suggestions:

11) How does 4 months of organoid development *in vitro* track with *in vivo* human development? What is known about ASD changes in the brain at this stage? Presumably something is known and this is highly relevant to this study if the authors are trying to convince us that these developmental changes are remotely biological and not artificial. Relatedly, it would be helpful if the authors could cite literature supporting the idea that ASD starts *in utero*.

We thank the reviewer for these interesting comments.

1. Brain organoid development vs *in vivo* fetal brain development. The developmental signatures of cortical organoids have been characterized using systematic transcriptional and epigenetic profiling and compared to *in vivo* fetal brain tissues across different time periods^{24,25}. At the transcription level, cortical organoids between 100-150 days match *in vivo* mid-fetal stages at post-conception weeks (PCW) 16-19. At the epigenetics level, cortical organoids in cultures at 80 to 250 days resemble fetal cortical tissues from mid- to late-fetal stages.

2. ASD brain changes in fetus and abnormal cortical development. ASD is diagnosed based on behavioral symptoms presented in early childhood when the brain development is almost complete. Thus, it is impossible to examine brain changes during prenatal stages in real time. Postmortem brain tissue examinations and gene functions studies suggest abnormal cortical development as one of the key mechanisms contributing to ASD pathology, which is reviewed by Torre-Ubieta et. al²⁶. Specifically, two seminal co-expression networks studies suggest enrichment of ASD genes in several cortical regions and specific cell types during early and mid-fetal stage (e.g., mid-fetal layer 5/6 projection neurons were proposed

to be a key point of convergence for ASD genes)^{27,28}. These data suggest that brain changes could emerge much earlier during development²⁹.

We have added new text (Page 2) and cited the literature in the introduction to further support the developmental defects in ASD.

12) Line 57 – The authors fail to acknowledge in the introduction that cerebral organoids do not produce the diversity of cell types found in the developing brain. Instead, the introduction makes it seem as if organoids are equivalent to little mini brains. However, in reality they lack many cell types and instead only include those from the RGC lineage really. Given that the first wave of entry of microglia into the developing brain precedes the peak of neurogenesis and the organoids do not have microglia, this seems like an important caveat to discuss.

We have added new text on microglia (Page 13) in the discussion to acknowledge the limitations.

13) The introduction or perhaps the discussion could benefit from more information on what is known about the role of the BAF complex in ASD.

We have added additional text (Page 12-13) on the BAF complex in the context of ASD in the discussion.

14) Figure 1e – The title of this plot is not very clear and the authors need to add a Y axis. Also the authors do not discuss what “Control” in this instance?

We thank the reviewer for pointing this out. We have added the title to Y axis and explained control in the figure legend. The control is a non-targeting gRNA control.

15) Lines 157-160: I think it is an overstatement to attempt to connect perturbation of cell type proportions with a ASD symptom; this could be and likely is caused by much more complex biology and physiology.

We agree with the reviewer’s criticism and have removed this statement.

16) Fig. 2b,e – I don’t find this presentation to be useful. I am not able to interpret this figure.

We agree with the reviewer that the analysis presented in the original Fig. 2b, e was partially overlapping with results from Fig. 2a, which might be confusing to the reader. We have therefore removed this analysis from the main figure.

17) I was not immediately convinced by the various enrichment analyses testing for enrichment in the SFARI database because the authors did not mention what the background set of genes was. I initially assumed the authors were using the whole transcriptome as the background but the methods show this is not the case. I would state the background set of genes in the main text.

We regret that this was not clear in our original manuscript, and we agree that the background gene set is crucial for the enrichment analysis. We have clarified this in the main text.

18) Extended Data Figure 6 is nearly uninterpretable at the current scale and lacking labels of cell types/trajectories in the plots.

We apologize for the low resolution of the figure. We have updated the figure with newly integrated data and plotted with higher resolution (**New Extended Data Fig. 9**). We also added a UMAP plot with cell type annotations.

19) The authors should at least discuss the caveat that these mosaic screens allow for but cannot assess effects caused by heterogeneous cell-to-cell interactions that might (for ex.) lead a cell with one knockout to affect a neighboring cell.

We thank the reviewer for bringing up this important point that we have missed in the manuscript. We have addressed this in a similar comment raised by reviewer 2 (see below, response to **comment 1**), **Cell-cell interaction in mosaic organoids**).

20) The prenatal imaging data in Fig 4h is a unique and interesting addition but I struggled to interpret the biological relevance of these figures and was forced to take them at face value with the explanation from the text.

We thank the reviewer for appreciating this precious fetal brain MRI analysis from the patient. Stem cells within the ventral telencephalon (GE) are the common progenitor pool for both interneuron precursors (INPs) and oligodendrocyte precursor cells (OPCs). We think the enlarged GE could be partially due to increased ventral progenitor numbers including both INPs and OPCs. This is consistent with what we see from the screen and the patient cell line derived organoids. Although it is impossible to obtain the patient fetal brain tissue to validate this phenotype, we still think it worthwhile to include the data which clearly shows abnormal GE development. We have added more text to interpret this piece of data.

21) The model shown in Fig. 4i is not very clear. The differences between the WT and ARID1B perturbation (arrow size, oval size) are not striking enough to draw my eye for immediate interpretation of the meaning.

We thank the reviewer for this thoughtful comment. We have modified the disease model to enlarge the arrow size and highlight the enlarged oval for INPs and OPCs (**New Fig. 4i**). We believe the model is now clearer in conveying our message.

Referee #2 (Remarks to the Author):

The work by Li et al. is timely and important for advancing our understanding of the genetic and developmental architecture of autism spectrum disorders (ASD).

In particular, leveraging CRISPR-based perturbations, brain organoids and single cell analysis, the authors set out to identify cell type-specific transcriptional endophenotypes as well as the impact on neurodevelopmental trajectories and gene regulation of the loss-of-function of 36 high-risk ASD genes. As a next step, they related the impact observed in the screening of ARID1B knock-out to the defects observed in cerebral organoids from iPSC of ASD patients harboring heterozygous ARID1B mutations.

These results are thus potentially very relevant, both for shedding new light on the neurobiological basis of ASD and for the novelty of the technological implementation of CRISPR-based perturbation in cerebral organoids.

Unfortunately, however, major flaws in the experimental design and in the analytical strategy do not allow to support the key knowledge claims of the paper, at least in its current state.

A thorough revision of the design is thus necessary to tackle the major concerns on the validity and accuracy of the findings, so as to render the work both robust and reproducible and meet the ambitious and laudable goal it set out to achieve.

Below is a list of the key issues, in the hope that it will be useful to guide authors in the reassessment of their strategy and the implementation of the new experiments.

We thank the reviewer for the enthusiastic and supportive comments. We have followed reviewer's suggestions and performed several lines of experiments, mainly including:

- 1) Repeat of the entire screen on 4-month-old brain organoids from 2 additional batches.
- 2) Systematic determination of gRNA representation and clone barcode on 4 batches of organoids in single cell or bulk experiments.
- 3) Repeat the entire scRNA-seq analysis on figure 2 and 3 with newly integrated data and perform bioinformatic analyses suggested by the reviewer.
- 4) Phenotypic validation of multiple genes in non-mosaic organoids with different experimental approaches.

These experiments resulted in a substantial amount of new data which greatly improved the accuracy of the findings and further demonstrated the robustness of the system.

Major concerns:

1) The numbers of cells for each perturbation (Extended Data Figures 5, in particular panel e, and 6) show a heavy imbalance: few perturbations in the mosaic organoids are over-represented, while several others are under-represented (including control cells that are under-represented or absent in several of the 8 libraries). In particular, a main concern emerges from the fact that the 2 perturbations with the highest number of recovered cells (KMT2C and LEO1) are differentially enriched, respectively, in the dorsal and ventral regions of the UMAP. This is a major issue because the over-representation of cells mutated for these 2 genes is in all likelihood the main driver of the observed dimensionality reduction where the authors identify the 2 regions of dorsal and ventral cell populations. Thus, a UMAP computed only on control cells, as well as a UMAP where the dataset is downsampled to have the same number of cells per perturbation, are both necessary to properly interpret the validity of cell type annotation and, as a consequence, of all the downstream analysis on the effects of the individual perturbations.

The imbalanced representation of perturbations, together with the impossibility in a mosaic design to discern cell-autonomous from non-cell-autonomous effects derived from the interplay of different perturbations, require the main results of the study to be replicated with new experiments. As minimal requirement, for genes showing a significant change of cell abundance, the results should be validated in homogeneous (ie. non mosaic) organoids carrying only that specific perturbation; this would allow both to confirm the observations with a larger number of cells, and to discern cell autonomous effects from potential non-autonomous ones. Given the ambition of the work, and the scalability of the employed assays, this approach should ideally be employed for all the 36 genes, and it appears somewhat surprising why, also in light of the effective gRNA construct design that was chosen, the systematic assay of all individual perturbations separately was not adopted as the baseline. When growing new batches of organoids, one suggestion to back the idea that perturbations are selected would be to track in time, at known crucial stages of organoid generation (or every 15/30 days), gRNA compositions of small pools of organoids (e.g. 3/5 organoids per pool, 3/5 pools per time point).

We thank the reviewer for this thoughtful comment and have addressed in the following 4 sections.

1. Cell-cell interaction in mosaic organoids.

Since a similar point has also been raised by **reviewer 1 (comment 19)** and **reviewer 3**, and is highly relevant to the reviewer's remaining comment, we would like to address this upfront.

4-OHT was added to induce eCas9 (dToma⁺) expression in 5 days old EBs. In our system, 4-OHT could not fully penetrate tissues. We have systematically titrated 4-OHT and found that when applied at a concentration between 0.3 ug/ml-0.6 ug/ml, 10-20% of the cells were induced (Day 30, GFP⁺/dToma⁺, **New Extended Data Fig. 5d**). We also tracked the mutant population through development at day 20, 40, 60, 80, 100 and 120 (0.3 ug/ml 4-OHT) and observed stable, low percentages (~20%) of mutant populations across all stages (**New Extended Data Fig. 5e**). At day 120, we observed an average of 21.8% mutant population (**New Extended Data Fig. 5f**). This data suggests that most of the cells in the mosaic organoids are wild-type, which greatly limits mutant cell-cell interaction. In addition, samples were prepared from multiple pools of organoids and every mosaic organoid consists of 36 mutants. Thus, the chance that a specific mutant always neighbors another same mutant cell is extremely low. Such neighboring effects will also be averaged out when analyzing hundreds and thousands of cells for each perturbation and comparing them to the internal control cells. We have now included additional clarification in the revised manuscript.

2. gRNA representation analysis

To address this point and a similar point raised by reviewer 1 (see detailed response to **reviewer 1, comment 5**), we carried out additional single-cell (14 pools of organoids) and bulk experiments (24 pools of organoids) from 3 independent batches. We analyzed gRNA representation in three conditions (eCas9-induced cells, eCas9 uninduced cells from either 4-OHT treated or untreated organoids) (**New Extended Data Fig. 7a**). We consistently see over- and under- represented gRNAs in eCas9-induced cells but not in uninduced cells at day 120. This led to an interesting discovery of the over-proliferation (e.g., *LEO1*, *KMT2C*) and depletion (e.g. *ADNP*, *WAC*) phenotypes from the screen (**New Fig. 2a, b**). As suggested, gRNA representation was also examined at different development stages (Day 20, 40, 60, 80) and compared to internal uninduced cells sorted (**New Extended Data Fig. 7b**). We found that percentages of some genes, such as *KMT2C* and *LEO1* already started to increase at day 20. The findings of the over-proliferation/depletion phenotypes were further validated by generating non-mosaic organoids individually for several perturbations (**New Extended Data Fig. 8**).

3. Computing new control UMAP for single cell analysis.

We agree that annotating the cells based on the full dataset might have been biased by perturbation effects. In the revised manuscript, we have therefore adopted a strategy similar to the one proposed by the reviewer with the aim to mitigate these biases. Manual annotation of cells is now purely based on control and uninduced cells (35,203 cells) (**New Extended Data Fig. 6d**), and cell type labels for the full dataset are derived from a label transfer. We found that perturbed cells generally adopted cell states resembling those found in the control, albeit with varying proportions.

4. Individual gene validation in non-mosaic organoids.

The reviewer is concerned about potential non-cell autonomous effects in the mosaic systems and thus suggests validating the findings in non-mosaic organoids. In our response to the reviewer's comment above, we describe that in the CHOOSE system, low percentages of cells (~20%) are eCas9-induced which greatly minimizes such effects.

We would like to emphasize that generating non-mosaic organoids with individual genes perturbed in an arrayed fashion is a completely different approach compared to the pooled CRISPR screening system. The arrayed screen is limited by scalability, although it offers opportunities for mechanistic follow-up studies. However, high-content single-cell screening can already identify complex biological phenotypes³⁰, which is a key strength of the CHOOSE system. Thus, we think that re-examining all perturbations separately extends beyond the scope of the current study. Especially since our system offers controlled setting for all perturbations during lentivirus production, infection of stem cells and collection, organoids generation, sample preparation and scRNA-seq, once gRNA plasmid library has been constructed. Mutant cells could, thus, be compared to the controls bearing the same environment, which allows us to isolate the phenotypes that are purely contributed by perturbations.

To further support our findings, we set out to validate the key phenotypes for several perturbations in non-mosaic organoids as suggested by the reviewer. For this, we generated individually perturbed organoids (**New Extended Data Fig. 8, 11, and 12**) and carried out three lines of phenotypic validation experiments:

- 1) We verified the over-proliferation and depletion phenotypes seen upon perturbation of *LEO1*, *KMTC2*, *ADNP* and *WAC*. We used a FACS-based approach to quantitatively analyze the ratio of mutant cells (GFP+/dTomato+) to non-mutant cells (GFP+/dTomato-) that were isolated from the same organoids at two development stages. The ratio was then compared to a non-targeting control gRNA. Over-proliferation of *LEO1* and *KMT2C*, and depletion of *ADNP* and *WAC* were consistently observed at both time points (**New Extended Data Fig. 8**).
- 2) We verified the depletion of intermediate progenitors (IP) upon perturbation of *KMT2C* and *PHF3*. We performed immunohistochemistry (IHC) analyses of EOMES positive (a marker for IPs) cells at day-60-old organoid tissues. At this stage, the organoids present an organized cytoarchitecture, with clearly separated ventricular zone (VZ), subventricular zone (SVZ) and cortical plate (CP), which allows reliable IHC measurement. Percentages of IPs in mutant cells-populated areas were quantified and compared to organoids infected with control gRNA (**New Extended Data Fig. 11**). We observed significant depletion of IPs in *KMT2C* and *PHF3* mutants, consistent with our findings from the screen.
- 3) We verified the increase of ventral progenitors upon perturbation of *KMT2C*, *MED13*, *PHF3* and *TBL1XR1*. We performed IHC analyses of DLX2 positive cells, a marker labels interneuron precursors³¹. Our protocol presents both dorsal and ventral regions at the early stages^{32,33}, and allows us to examine INPs already at Day 60. We found significantly increased DLX2 positive cells in the mutant rosettes area for perturbations of *KMT2C*, *MED13*, *PHF3* and *TBL1XR1* (**New Extended Data Fig. 12**).

2) The experimental design followed by the authors is based on the pooling of 3-6 organoids for each of the 8 chromium controller runs. There are thus no independent replicates that can be used in the analysis to evaluate the reproducibility of the observed phenotypes are. Indeed, it is truly remarkable that the word “replicate” never appears in the paper. All results in figures 2 and 3 (and associated extended data figures) about the effects of the perturbations in terms of ratio of dorsal to ventral cell populations, specific enrichment/depletion for a particular cell type, and differential densities along pseudotemporal axis need replicates to be interpreted.

- The two above issues imply that it is not possible to evaluate the validity and accuracy of all the main results of the paper: figure 2 and 3.

We thank the reviewer for pointing this out. We agree that multiple biological replicates are crucial for validating our results. In the original manuscript, we had performed the screen in 8 independent pools of organoids, which we consider to be biological replicates. In the revised manuscript, we first have added 3 more replicates from Batch 2, an independently grown batch of organoids (**New Extended Data Fig. 7a**, eCas9 induced-cells, pink). We have reanalyzed the data including the newly generated batch by testing (as we did previously) on a single cell level, using replicate as a covariate to control for biases stemming from differential cell numbers between replicates. We have updated all relevant main and supplemental figures including cell type abundance testing, differential gene expression analysis as well as GRN enrichment.

Furthermore, this newly integrated data now allowed us to evaluate the consistency of effects across independent pools of organoids (14 biological replicates, 3 batches) from multiple batches (including another 3 replicates from a third independent batch, using non-induced cells as an alternative control) to complement the analysis performed on the single-cell level (**New Extended Data Fig. 10**). Here, we treated every pool of organoids as an independent sample (rather than every cell) and performed a t-test on the per-organoid pool fold enrichment of each guide. This approach can be used to evaluate the variability of the effect across library replicates from different batches, but it has less power compared to the testing on the single-cell level. We are glad to see that many of the effects highlighted in the study (e.g., depletion of IP in *KMT2C*, *BCL11A*, *CHD2*, *KDM5B* etc. perturbations, enrichment of INP in *CIC*, *IRF2BPFL*, *MED13*, *PHF3*, *TBL1XR1*, *TCF20* etc. perturbations, vRG enrichment in *ARID1B* perturbation) were significant also in this test, which supports their strong consistency across independently grown organoids. Although it is worth noting that if effects were detected on a single-cell level but not on the level of pools of organoids this might stem from higher variability of the effect across organoids pools but does not invalidate the effect seen on the single cell level. Overall, we found that the major effects on cell type abundance changes highlighted in the original manuscript were corroborated by both analyses.

3) The authors presented an elegant and novel strategy to generate efficient and controlled pooled systems with high clonal complexity in pluripotent stem cells, not in organoids as they claim, since there is no data showing the unique clone barcode distribution after differentiation of the organoids. This is instead a result of obvious relevance that should be shown.

We thank the reviewer for appreciating the barcoded system and apologize for missing this important aspect of the CHOOSE system. We have performed targeted amplification and retrieved clone information from both scRNAseq experiments as well as additional bulk genomic DNA analysis (**New Extended Data Fig. 4c-h**). For scRNA-seq, we have recovered on average 125 clones for each perturbation (median: 106) and the clones are distributed across all libraries (**New Extended Data Fig. 4c, d**). We also analyzed the size of each clone. The mean average cell number in each clone per perturbation is 4.4 (with median of 3.6; **New Extended Data Fig. 4 e, f**). These data suggest that cells captured in the CHOOSE screen came from diverse and relatively small clones, which is important for eliminating dominant clonal effects. Bulk analysis of the genomic DNA with a much higher cell number input (50-150K) reveals the high clonal complexity in both eCas9-uninduced and induced cells, with a homogenous distribution in uninduced cells as expected (**New Extended Data Fig. 4g, h**). Interestingly, over-proliferation/depletion phenotypes could also be reflected by clone

analysis, as the gene LEO1 presents the highest clone number and averaged clone size. Together, these data underscore the great advantage of barcoding in tissue-based screening.

4) Results from differential abundance analysis should be strengthened by applying algorithms that have been developed for scRNASeq frameworks such as scCODA that account for the inherent bias present in cell-type compositions with a Bayesian approach for cell-type composition to address the low replicate issue, and/or MILO, thus does not rely on discrete clusters as input when testing for differential abundance among experimental conditions, thus improving the detection of differences in continuous trajectories.

We thank the reviewer for alerting us to these recently developed tools for differential abundance analysis. In the manuscript we employed a Cochran–Mantel–Haenszel statistic which allowed us to control for biases stemming from differential cell numbers and perturbation gRNA representations between replicates. We were curious to see if the results of our analysis would be confirmed by MILO which was specifically developed for this purpose. While we noticed that pooled CRISPR screens violated the 1 observation = 1 experimental sample assumption made by both MILO and scCODA, we were able to apply MILO to our data using workarounds suggested by the authors (<https://github.com/MarioniLab/miloR/issues/259>). We found that the analysis with MILO confirmed many of the results that we had seen previously, including extensive depletion of IPs in multiple perturbations, enrichment of OPCs in ARID1B, enrichment astrocytes in CIC as well as a widespread depletion of dorsal telencephalon and/or enrichment of ventral telencephalon fates (e.g., *KMT2C*, *BCL11A*, *KAT2B*) (**Reviewer Fig. 4**). While we found that these results strengthened the confidence in our previous analysis, we reasoned that including them in the manuscript could be confusing to the reader. Thus, we include this as a reviewer figure.

Reviewer Fig. 4 Differential abundance testing using MILO. **a**, UMAP embedding showing cell type annotations. **b**, MILO analysis performed with library (replicate) as a covariate as suggested by the authors via GitHub. The results of MILO for 3 target genes with strong effects in our analysis are displayed as UMAP embeddings showing differential abundance in individual neighborhoods (top) and barplots with fold changes aggregated to cell type (bottom).

5) In figure 2A, it would be useful to have relevant about the magnitude of the enrichment in addition to the significance.

As suggested, we have changed the plots throughout the manuscript (**New Fig. 2c, 3a**) to better display the effect size of both composition changes and differential expression.

6) For the genes that show an impact on the abundance of specific cell populations, it would be interesting to check if the detected effect is coherent with the gene expression patterns across cell types in the organoid model in study as well as in datasets from fetal brain at comparable stages.

We thank the reviewer for bringing up this interesting point. We checked the expression of the perturbed genes in both our organoids dataset and a recently published fetal brain development dataset at 5-14 pcw³⁴ (**New Extended Data Fig. 6i, j**). Indeed, we see several perturbed genes with cell-type specific enrichment, especially in organoids. For example, the three BAF complex members have very distinct expression pattern: *ARID1B* is highly expressed in all three types of interneurons as well as L23 EN; *BCL11A* is highly expressed in IP, L6_ThPN and L23 EN; and *SMARCC2* expresses at low levels in all cell types. This probably suggests a cell-type specific requirement of individual BAF complex members during neural stem cell lineage progression³⁵. Indeed, *ARID1B* and *BCL11A* have larger effects on multiple cell types including IP, EN and INs, while *SMARCC2* perturbation only affects vRG.

7) Have the expression levels of targeted genes been checked in the results of the differential expression analysis? For each guide the expression level of the target gene should be shown for the perturbed cells vs the others. This would be instrumental also for assessing the relevance of the dosage difference between the *ARID1B* knock-out and the *ARID1B* haploinsufficient backgrounds, and hence properly evaluating this validation aspect of the work (Fig. 4)

We thank the reviewer for the comment. Using gene expression levels to reflect CRISPR editing outcomes can be misleading. This is because edited gene sequences can still be transcribed to mRNAs. Although it is also true that in many cases, CRISPR editing results in a premature stop codon, which may trigger nonsense-mediated mRNA decay. As suggested, we checked DE for each perturbed gene in dorsal and ventral populations (**Reviewer Fig. 5**). As expected, we found dysregulated genes in some but not all perturbations.

Reviewer Fig. 5

DE analysis of perturbed genes. Dot plot showing differential expression of the gRNA target gene in perturbed cells for each the dorsal and ventral telencephalon developmental trajectory. Size indicated $-\log_{10}(p\text{-value})$, color indicates log fold change.

To check the protein levels of *ARID1B*, we generated organoids with *ARID1B* perturbation and measured by IHC (**New Extended Data Fig. 2**). Our results suggest an average of 38.6% reduction of protein levels in eCa9-induced cells compared to eCas9-uninduced cells. This reduction is not caused by eCas9 expression as similar protein expression levels are detected in eCas9-induced and uninduced cells from organoids infected with non-targeting control gRNA. This data highlights the relevance of the phenotypes observed in the screen compared to organoids derived from *ARID1B* heterozygous patient iPSCs.

8) From the Manhattan plot in Figure 2, the number of DEGs is quite low for most of the examined conditions. This is quite puzzling when compared to the results about cell type abundance. How do the authors explain this striking discrepancy?

We also found it interesting that the gene expression changes induced by the perturbations were rather subtle compared to changes in cell type composition. The ability to detect such subtle differential expression changes in our screen is influenced by the number of cells we detect per target gene and cell state and are aware that our statistical power is limited for some targets. We were therefore pleased to see that the inclusion of more replicates almost doubled the number of detected DEG. Overall, we detect DE genes for all perturbed targets and show that they are involved in important pathways and regulomes (**New Extended Data Fig. 14c, Fig. 3a-c, f, Supplementary Data 2-4, 7**).

Furthermore, we think that strong compositional changes need not correspond to strong expression changes in the differentiated cell type. For instance, a perturbation could induce small alterations in gene expression during development thereby impacting the outcome of differentiation and cell fate decision events, but which are not maintained in the terminally differentiated cell type.

9) Is the statement about the involvement of the ubiquitin-proteasome pathway substantiated by a functional enrichment analysis or other analytical techniques?

Finally, most observations are solely grounded on differential expression analyses with no significant validation of at least key findings. At least basic biochemical assays are required, for specific perturbations, to corroborate/support claims and observations such those on mitochondrial and proteasome pathways dysregulation.

We thank the reviewer for the comments. As they are related, we would like to address them here together. As suggested, we now performed systematic GO enrichment analysis with dorsal and ventral DEGs separately for each perturbation (**New Extended Data Fig. 14c**, full list in **Supplementary Data 2**). We are glad to see that pathways involved in protein misfolding and mitochondria function are enriched in multiple perturbations, including *ASHL1* (dorsal), *DDX3X* (dorsal), *FOXP1* (dorsal), *KDM5B* (ventral), *KMT2A* (ventral). Excitingly, we also observed many enriched GO terms covering diverse functions, many of which are confirming previous studies, such as ribosome assembly (*SETD5* perturbation)³⁶, forebrain development (*TBR1* perturbation)³⁷, mitochondrion organization (*FOXP1* perturbation)³⁸, lipid homeostasis (*IRF2BPL* perturbation)³⁹, autophagosome maturation (*KAT2B* perturbation)⁴⁰, and cilium development and histone demethylation (*MECP2* perturbation)⁴¹, which further support the power of detecting complex biological phenotypes in the CHOOSE system. We thus decide to not only focus on the mitochondrial and proteasome pathways, but rather highlight all these consistent observations and provide a comprehensive GO enrichment list which can serve as a resource for future studies of under-explored genes. We thank the reviewer for this helpful suggestion and have added this new analysis and edited the text in the section of 'CHOOSE identifies dorsal and ventral telencephalon-specific dysregulated genes' (Page 8-9, highlighted in blue).

10) The CHD8 perturbation does not show the impact on cell population abundance that was reported by several papers in similar organoid models and in vivo (e.g Paulsen et al. 2022; Villa et al. 2022; Jin et al. 2020). What is the explanation for this (which should anyway be discussed in the paper)?

The Paulsen et al. paper identified increased interneurons and interneuron progenitors at 3.5-month-old organoids in the HUES66 cell line but not in two other cell lines (GM08330 and H1). In their *in vivo* mouse studies (Jin et al. 2020), perturbation of CHD8 does not lead to any cell type compositional changes, but a gene expression module associated with oligodendrocyte progenitor (ODC1). Olig1 is also dysregulated in CHD8 perturbation in our study (**Supplementary Data 2**). A more thorough study on CHD8 from Villa et.al 2022 identified increased interneuron generation at D60, but not at D120 in two cell lines. These results (no interneuron changes at D120 in 4 out of 5 cell lines, and no changes in *in vivo* studies) are in fact very much in line with our observation of a negative result. In addition, the Villa et.al paper very nicely confirmed that phenotypes such as enlarged organoids size and proliferation associated with macrocephaly are human mutation-specific. In our screen, we observed a trend of increased interneuron population, as well as cycling ventral radial glia cells, but it is not significant. Given these results, we think the fact that we did not capture cell abundance changes can be due to a small effect size of CHD8 on cell fate determination or to the phenotypes being more mutation-specific, although it can also be due to the power of our experimental approach.

Referee #3 (Remarks to the Author):

The Manuscript entitled "Single Cell CRISPR Screening in Organoids Identifies the Origins of Autism" submitted by Li and colleagues aims to interrogate the neurodevelopmental consequences after perturbation of 36 Autism Spectrum Disorder (ASD) risk genes. The authors developed a pooled loss-of-function CRISPR screening in system mosaic organoids, allowing for the simultaneous analysis of all perturbed genes within a single organoid. Using a single-cell multi-omics approach the authors discovered dysregulation of several developmental pathways following perturbation, mainly converging on the developmental trajectories of dorsally derived intermediate progenitor cells and ventrally derived radial glia. This effect was most obvious within cells that received the loss of function perturbation for ARID1B, a member of the BAF chromatin remodeling complex. Further analysis of the single-cell data set revealed a propensity of these cells more frequently giving rise to oligodendrocyte progenitor cells (OPCs) as opposed to neurons. This finding was supported by the generation and analysis of ARID1B patient-derived ventrally patterned organoids which showed a greater expression of Olig2 at DIV 40. This was further validated by an in utero MRI scan of the ARID1B patient which allowed for volumetric measurement of the CGE and LGE, two germinal zones that give rise to OPCs.

Overall the manuscript has the merit of focusing on an approach that aims at tackling the critical question of whether ASD converges on the development of specific cell types during fetal development. However, while the aspirational goal of this study is very relevant to the field, the current execution of this idea has significant flaws that limit the relevance and accuracy of the results. Therefore, the general sense is that the idea is interesting, but that the results should be validated in at least another line with different genetic backgrounds and in NON-mosaic organoids where separate genes are downregulated in individual organoids.

General Notes

1) The title of the paper is exaggerated and inaccurate.

We agree with the reviewer's criticism. We have now changed the title to '*Single-cell brain organoid screening identifies developmental defects in autism*'.

2) There are big concerns about the fundamentals of the pooled CRISPR screen method - perturbing so many genes within a single system, will lead to significant crosstalk and feedback between populations. Observed phenotypes may not be the direct result of the mutation but a downstream effect. It is incredibly difficult and inaccurate to parse apart the direct cause and effect relationships in this system.

We thank and are grateful to the reviewer for bringing up this important point and agree that potential cell-cell interaction could be a confounding factor in general in pooled CRISPR screens. We apologize for missing an important clarification for the CHOOSE system (see detailed response to reviewer 2, **comment 1, Cell-cell interaction in mosaic organoids**) that in our system, eCas9-induced cell populations are controlled at a low amount, with 21.8% at day 120 and ~ 20% throughout the development (**New Extended Data Fig. 5d-f**). This largely minimizes the cell-cell interactions in mosaic organoids. Besides, samples were prepared from multiple pools of organoids and neighboring effects will also be averaged out when analyzing hundreds and thousands of cells for each perturbation and comparing them to the internal control cells.

In addition, we have included more replicates from an additional independent batch for single cell analysis (11 pools of organoids, **New Fig. 2c**), a thorough consistency test on 3 independent batches (14 pools of organoids, **New Extended Data Fig. 10**), and performed several lines of validation experiments in non-mosaic organoids with individually perturbed genes including over-proliferation/depletion phenotypes (**New Extended Data Fig. 8**), depletion of INPs (**New Extended Data Fig. 11**), enrichment of INPs (**New Extended Data Fig. 12**). These new results together substantially improve the accuracy of our findings.

3) The timepoints shown for the organoid experiments are not consistent (single-cell at 4 months - immunohistochemistry at day 40, organoid validation up to 27 days)

We thank the reviewer for the comment. From the primary screen we found that *ARID1B* perturbation causes significant enrichment of ventral telencephalon progenitors including OPCs and INPs. We thus used a previously published ventral protocol aiming to specifically enrich ventral regions and examine affected progenitors at an earlier time point^{21,22}. Without ventralization, ventral lineages are populated at a later stage as previously reported²³. We have now also confirmed at day 120 without adding SAG and IWP2, increased INP and OPC populations consistently present in 2 patient cell lines and an additional engineered cell line (See details in responses to reviewer 1, **comment 8**).

With respect to 'organoid validation up to 27 days', we want to clarify that the microscopic images of overall organoids morphology at day 27 and the IHC staining at day 24 are not validations of mutant phenotypes, but rather representations of organoids showing tissue quality and identity of CHOOSE organoids at early stages, including successful eCas9 induction (dTomato+) and forebrain patterning (FOXG1+). These quality and identity measurements are important to perform at earlier time points.

Figure 1

1) It seems odd that despite the generation of LGE, CGE, and dorsal forebrain there was no generation of MGE which is a major contributor to interneuron populations

We agree with the reviewer that it would be ideal to have all three GE regions together with dorsal forebrain in one protocol. However, organoids with different patterning protocols are always enriched with different regions^{21,22,32,33,42-45}. With a focus on cortical development, we used a protocol established by our group which produces dorsal and a subset of ventral

regions^{32,33}. Especially, consistent with previous observation, MGE region is missing from this protocol (Figure S1, S2³³). This was also reported by another comprehensive single cell analysis of cortical organoids⁴⁵. We have now added new text in the discussion with respect to this protocol limitation.

2) There is no proteomic validation of the knockouts, this is a necessity.

We are not entirely sure which specific aspect of the proteomic validation the reviewer is referring to, as 'proteomic' study is a broad term including studying protein structure, function, interactions and dynamics⁴⁶. Perhaps the reviewer is asking for the validation of protein expression levels caused by the genome editing. If so, we agree that efficient genome editing is important for the screen. This is also why we have established a FACS-based gRNA editing reporter assay to quantitatively determine and pre-select efficient gRNAs (**Extended Data Fig. 1**). In addition, protein levels will not always reflect editing efficiency, as about 20% of CRISPR-Cas9 mutagenesis produces in-frame mutation⁴⁷. Still, we set out to validate several perturbations with IHC in non-mosaic organoids with individual genes perturbed. Among 6 genes that we have validated, we found 5 (*ARID1B*, *ADNP*, *PHF3*, *SMARCC2*, *KMT2C*) perturbations that cause significant reduction of protein levels (**New Extended Data Fig. 2**). It is worth noting that the detection of proteins using antibodies is also limited by their recognition sites, as most of the antibodies are generated using a protein fragment as an antigen. Antibodies could, therefore, still generate positive signals even if a truncated protein is produced.

3) These mutations in patients are mainly heterozygous, the mutations used in the screen are homozygous. The findings from this screen may then have been largely exacerbated by this genetic difference.

We thank the reviewer for bringing up this important point. We have addressed this in a similar comment raised by Reviewer 1 (**comment 3**).

4) There is no clonal bias at the iPSC stage, but in supplemental fig. 5 and 6 a clear bias in the screen is seen.

We hope we have understood the reviewer correctly that they are referring to gRNA representation bias presented in previous Extended Data Figure 5 and 6. We have systematically tested this and addressed it in a similar comment raised by Reviewer 2 (**comment 1, section 2, gRNA representation analysis**) (**New Fig. 2a, b, Extended Data Fig. 7**).

5) There is no reference to the clonal barcode - is one clone contributing more than others?

We regret missing this important aspect for the CHOOSE system and have addressed this in a similar comment raised by Reviewer 2 (**comment 3**) (**New Extended Data Fig. 4c-h**). We have recovered on average 125 clones for each perturbation (median: 106) and the clones are distributed across all libraries (**New Extended Data Fig. 4c, d**). We also analyzed the size of each clone. The mean average cell number in each clone per perturbation is 4.4 (with median of 3.6; **New Extended Data Fig. 4 e, f**).

6) There is a confounding variable of neighboring knockout cells influencing the differentiation trajectory of each other.

We thank the reviewer for the comment. We have addressed this in a similar comment raised by Reviewer 2 (**comment 1, Cell-cell interaction in mosaic organoids**) (**New Extended Data Fig. 5d-f**).

7) The screen should be repeated in other lines to determine if any effects are cell line-specific and background specific.

The H9 ES cell line used in the current screen is one of the most widely used embryonic stem cell lines in the field and many findings and resources were generated using it (<https://hpscereg.eu/cell-line/WAe009-A>, publications and projects). The biological discoveries made using the H9 cell line have also been widely acknowledged and proven^{33,42,48-52}. We agree that genetic background could contribute to phenotypic variability, a suggestion made by many studies. For this reason, we put huge efforts in recruiting patients and used patient cell lines (two ARID1B mutant cell lines and one corrected cell line with the same patient genetic background) for functional validations. Our results from patient cell line derived organoids are nicely in line with the primary screen. Regrettably, recruiting patients for each of the genes and functionally validating them is impossible and not practical for this current study. Furthermore, we considered the implications of such an attempt in terms of sustainability as well as resource management and reasoned that the value gained would highly likely not be justified. Thus, we feel that repeating the entire screen in a different genetic background adds limited value to the current study, and using patient derived cell lines when applicable would be a more suitable approach.

Figure 3

1) Showing dysregulation of gene expression and regulatory networks after genetic perturbation of key regulating genes seems redundant and uninformative.

Even though it is expected that perturbing regulatory genes will dysregulate GRNs, we would argue that it is not equitable to being uninformative. In our view, it is important to know which genes are dysregulated upon perturbation, as they drive the downstream effects of the perturbation. Thus, DE gene sets and TF modules can give insight about how and why mutations in regulating genes can drive disease phenotypes and cause cell type imbalance as highlighted in our manuscript. For instance, GO enrichment analysis of dysregulated genes revealed general disruption of synapse assembly and fate commitment, as well as processes that were specific to target genes. Furthermore, we found that both dysregulated genes and ASD risk genes are enriched in shared regulomes, indicating ASD-associated regulatory 'hubs'. We think that these findings are interesting and informative since it is not obvious which processes are affected by perturbing key regulators.

Figure 4

1) There are weak evidence showing transitional changes at the day 40 timepoint while all other experiments were done at 4 months.

We thank the reviewer for pointing this out and apologize for not providing enough explanation for the transition. In **Fig. 4 a-e**, analyses were performed using primary screen data at 4 months. Validations using patient iPSC lines in **Fig. 4f** were performed using ventralized organoids by adding SAG and IWP2 according to previously published protocols^{21,22}. The reason is we specifically wanted to examine ventral progenitors (INPs and OPCs). Without ventralization, ventral lineages are populated at a later stage in organoids as previously reported²³. Thus, a ventralized protocol would be more suitable for us to examine the phenotypes in tissues with enriched ventral regions at an earlier time point. We have added additional text to clarify this. We also wanted to point out that without adding ventral

patterning factor, the phenotypes were also confirmed at day 120 using scRNA-seq on three mutant cell lines (See details in response to Reviewer 1, **comment 8**).

2) The increased number of cells could be due to increased vulnerability of INPs (ie. cell death), so potentially no cell fate decision is affected.

We are not sure if we understand the reviewer's comment correctly. If there is increased vulnerability of INPs such as cell death, then we would expect a decreased number, not increased number of INPs.

3) There is no reference to the Paulsen et al. Nature 2022 that shows the same phenotype in ARID1B line, using single-cell rna-seq (stronger evidence compared to only using 2 markers).

We thank the reviewer for the comment. We have now cited the study. In addition, we have performed scRNAseq on the two patient cell lines and another engineered iPSC line (see detailed response to Reviewer 1, **comment 8**). We consistently observed increased OPC and INP populations.

4) The data obtained from in utero MRI's is interesting but the LGE/CGE volume needs to be normalized to total brain volume in order to be relevant.

We thank the reviewer for this great suggestion. We have now normalized GE volume to total brain volume as well as cortical volume (**New Extended Data Fig. 16c-g**). We also included the raw total brain volume and cortical volume for readers. In addition, we have included more controls (5 controls at two different development stages, GW22+23 and GW31), which allow us to further refine our segmentation procedure. As it is challenging to distinguish MGE, CGE and LGE from in utero MRI, we thus used GE volume to be more accurate. The absolute GE volume, as well as normalized GE to total brain and cortex volume are consistently increased at two development stages compared to the averaged control values.

5) It is again surprising that no MGE was produced using the Bagley 2017 protocol which was shown to robustly generate this region. There is no explanation for this in the manuscript.

We thank the reviewer for the comment. We would like to first clarify the plots in **Fig. 4a-e** are generated from sub-clustering and analysis of the ventral telencephalon population in the primary screen, in which the tissues were generated using the Lancaster et al. 2017 protocol and Esk et al. 2021 protocol^{32,33}, thus we did not observe MGE population as addressed in **comment 1** for Figure 1.

The specific validation of ventral progenitors in two patient cell lines were performed in organoids generated using Bagley 2017 protocol, as addressed in **comment 1** for Figure 4. We have modified the text and added justification to avoid confusion.

References

1. Bacon, E. C. *et al.* Rethinking the idea of late autism spectrum disorder onset. *Dev Psychopathol* **30**, 553–569 (2018).
2. Study, T. D. *et al.* Synaptic, transcriptional, and chromatin genes disrupted in autism. *Nature* **515**, 209–215 (2014).
3. Pinto, D. *et al.* Convergence of Genes and Cellular Pathways Dysregulated in Autism Spectrum Disorders. *Am J Hum Genetics* **94**, 677–694 (2014).

4. Graziano, C. et al. A New Homozygous CACNB2 Mutation has Functional Relevance and Supports a Role for Calcium Channels in Autism Spectrum Disorder. *J Autism Dev Disord* **51**, 377–381 (2021).
5. Hnoonual, A., Graidist, P., Kritsaneepaiboon, S. & Limprasert, P. Novel Compound Heterozygous Mutations in the TRAPPC9 Gene in Two Siblings With Autism and Intellectual Disability. *Frontiers Genetics* **10**, 61 (2019).
6. Novarino, G. et al. Mutations in BCKD-kinase Lead to a Potentially Treatable Form of Autism with Epilepsy. *Science* **338**, 394–397 (2012).
7. A., S. K. et al. Recessive Symptomatic Focal Epilepsy and Mutant Contactin-Associated Protein-like 2. *New Engl J Med* **354**, 1370–1377 (2006).
8. Chahrour, M. H. et al. Whole-Exome Sequencing and Homozygosity Analysis Implicate Depolarization-Regulated Neuronal Genes in Autism. *Plos Genet* **8**, e1002635 (2012).
9. Hayek, L. E. et al. KDM5A mutations identified in autism spectrum disorder using forward genetics. *Elife* **9**, e56883 (2020).
10. Darbandi, S. F. et al. Neonatal Tbr1 Dosage Controls Cortical Layer 6 Connectivity. *Neuron* **100**, 831–845.e7 (2018).
11. Bacon, C. et al. Brain-specific Foxp1 deletion impairs neuronal development and causes autistic-like behaviour. *Mol Psychiatr* **20**, 632–639 (2015).
12. Jung, E.-M. et al. Arid1b haploinsufficiency disrupts cortical interneuron development and mouse behavior. *Nat Neurosci* **20**, 1694–1707 (2017).
13. Derafshi, B. H. et al. The autism risk factor CHD8 is a chromatin activator in human neurons and functionally dependent on the ERK-MAPK pathway effector ELK1. *Sci Rep-uk* **12**, 22425 (2022).
14. Ding, S. et al. CHD8 safeguards early neuroectoderm differentiation in human ESCs and protects from apoptosis during neurogenesis. *Cell Death Dis* **12**, 981 (2021).
15. Guy, J., Hendrich, B., Holmes, M., Martin, J. E. & Bird, A. A mouse Mecp2-null mutation causes neurological symptoms that mimic Rett syndrome. *Nat Genet* **27**, 322–326 (2001).
16. Derecki, N. C. et al. Wild-type microglia arrest pathology in a mouse model of Rett syndrome. *Nature* **484**, 105–109 (2012).
17. Clement, K. et al. CRISPResso2 provides accurate and rapid genome editing sequence analysis. *Nat Biotechnol* **37**, 224–226 (2019).
18. Thielecke, L., Cornils, K. & Glauche, I. genBaRcode: a comprehensive R-package for genetic barcode analysis. *Bioinformatics* **36**, 2189–2194 (2019).
19. Datlinger, P. et al. Pooled CRISPR screening with single-cell transcriptome read-out. *Nat Methods* **14**, 297–301 (2017).
20. Paulsen, B. et al. Autism genes converge on asynchronous development of shared neuron classes. *Nature* **602**, 268–273 (2022).
21. Bagley, J. A., Reumann, D., Bian, S., Lévi-Strauss, J. & Knoblich, J. A. Fused cerebral organoids model interactions between brain regions. *Nat Methods* **14**, 743–751 (2017).
22. Bajaj, S. et al. Neurotransmitter signaling regulates distinct phases of multimodal human interneuron migration. *Embo J* **40**, e108714 (2021).
23. Uzquiano, A. et al. Single-cell multiomics atlas of organoid development uncovers longitudinal molecular programs of cellular diversification of the human cerebral cortex. doi:10.1101/2022.03.17.484798.
24. Trevino, A. E. et al. Chromatin accessibility dynamics in a model of human forebrain

development. *Science* **367**, (2020).

25. Gordon, A. et al. Long-term maturation of human cortical organoids matches key early postnatal transitions. *Nat Neurosci* **24**, 331–342 (2021).
26. Torre-Ubieta, L. de la, Won, H., Stein, J. L. & Geschwind, D. H. Advancing the understanding of autism disease mechanisms through genetics. *Nat Med* **22**, 345–361 (2016).
27. Willsey, A. J. et al. Coexpression Networks Implicate Human Midfetal Deep Cortical Projection Neurons in the Pathogenesis of Autism. *Cell* **155**, 997–1007 (2013).
28. Parikhshak, N. N. et al. Integrative Functional Genomic Analyses Implicate Specific Molecular Pathways and Circuits in Autism. *Cell* **155**, 1008–1021 (2013).
29. Hazlett, H. C. et al. Early brain development in infants at high risk for autism spectrum disorder. *Nature* **542**, 348–351 (2017).
30. Bock, C. et al. High-content CRISPR screening. *Nat Rev Methods Primers* **2**, 8 (2022).
31. Petryniak, M. A., Potter, G. B., Rowitch, D. H. & Rubenstein, J. L. R. Dlx1 and Dlx2 Control Neuronal versus Oligodendroglial Cell Fate Acquisition in the Developing Forebrain. *Neuron* **55**, 417–433 (2007).
32. Lancaster, M. A. et al. Guided self-organization and cortical plate formation in human brain organoids. *Nat Biotechnol* **35**, 659–666 (2017).
33. Esk, C. et al. A human tissue screen identifies a regulator of ER secretion as a brain-size determinant. *Science* **370**, 935–941 (2020).
34. Braun, E. et al. Comprehensive cell atlas of the first-trimester developing human brain. *Biorxiv* 2022.10.24.513487 (2022) doi:10.1101/2022.10.24.513487.
35. Sokpor, G., Xie, Y., Rosenbusch, J. & Tuoc, T. Chromatin Remodeling BAF (SWI/SNF) Complexes in Neural Development and Disorders. *Front Mol Neurosci* **10**, 243 (2017).
36. Nakagawa, T. et al. The Autism-Related Protein SETD5 Controls Neural Cell Proliferation through Epigenetic Regulation of rDNA Expression. *IScience* **23**, 101030 (2020).
37. Bedogni, F. et al. Tbr1 regulates regional and laminar identity of postmitotic neurons in developing neocortex. *Proc National Acad Sci* **107**, 13129–13134 (2010).
38. Wang, J. et al. Mitochondrial dysfunction and oxidative stress contribute to cognitive and motor impairment in FOXP1 syndrome. *Proc National Acad Sci* **119**, e2112852119 (2022).
39. Chen, H.-H. et al. IRF2BP2 Reduces Macrophage Inflammation and Susceptibility to Atherosclerosis. *Circ Res* **117**, 671–683 (2015).
40. Jia, Y.-L. et al. P300/CBP-associated factor (PCAF) inhibits the growth of hepatocellular carcinoma by promoting cell autophagy. *Cell Death Dis* **7**, e2400–e2400 (2016).
41. Frasca, A. et al. MECP2 mutations affect ciliogenesis: a novel perspective for Rett syndrome and related disorders. *Embo Mol Med* **12**, e10270 (2020).
42. Lancaster, M. A. et al. Cerebral organoids model human brain development and microcephaly. *Nature* **501**, 10.1038/nature12517 (2013).
43. Velasco, S. et al. Individual brain organoids reproducibly form cell diversity of the human cerebral cortex. *Nature* **570**, 523–527 (2019).
44. Birey, F. et al. Assembly of functionally integrated human forebrain spheroids. *Nature* **545**, 54–59 (2017).
45. Uzquiano, A. et al. Proper acquisition of cell class identity in organoids allows definition of fate specification programs of the human cerebral cortex. *Cell* **185**, 3770–3788.e27 (2022).

46. Jensen, O. N. Interpreting the protein language using proteomics. *Nat Rev Mol Cell Bio* **7**, 391–403 (2006).
47. Michlits, G. et al. Multilayered VBC score predicts sgRNAs that efficiently generate loss-of-function alleles. *Nat Methods* **17**, 708–716 (2020).
48. Wang, Y. et al. Modeling human telencephalic development and autism-associated SHANK3 deficiency using organoids generated from single neural rosettes. *Nat Commun* **13**, 5688 (2022).
49. Pinson, A. et al. Human TKTL1 implies greater neurogenesis in frontal neocortex of modern humans than Neanderthals. *Science* **377**, eabl6422 (2022).
50. Samarasinghe, R. A. et al. Identification of neural oscillations and epileptiform changes in human brain organoids. *Nat Neurosci* **24**, 1488–1500 (2021).
51. Villa, C. E. et al. CHD8 haploinsufficiency links autism to transient alterations in excitatory and inhibitory trajectories. *Cell Reports* **39**, 110615 (2022).
52. Kanton, S. et al. Organoid single-cell genomic atlas uncovers human-specific features of brain development. *Nature* **574**, 418–422 (2019).

Reviewer Reports on the First Revision:

Referees' comments:

Referee #1 (Remarks to the Author):

In this substantial revision, Li & Fleck et al. tempered and expanded their analyses of neurodevelopmental defects underlying Autism Spectrum Disorder (ASD). Major suggested revisions focused on: 1) increasing sample size and independent replicates to bolster findings; 2) validation of findings of mosaic screens in non-mosaic organoids individually targeting one gene at a time; 3) addressing technical limitations of CRISPR-based gRNA, such as guide and clonal diversity after EB formation and heterozygotic vs. homozygotic knockout; and 4) improved organization and presentation of data in streamlined figures. The authors made progress to address numerous criticisms and adjusted the language in the manuscript when necessary to recognize key limitations of the experimental design and model systems used. Through increased and independent replication, their main findings on alteration to cell type proportions due to ARID1B deficiency have been significantly bolstered. Validation and QC of the system, such as quantification of %eCas9 dTomato+ cells per organoid, has clarified the technical limitations. In total, the authors should be commended for performing a substantial and well-organized overhaul of their initial submission. In particular, I'd like to thank the authors for making the presentation of the revised manuscript easier to review given its organization (table of changes, etc) – the effort that went into this is not lost on me. Below I outline areas where I believe minor (but still important) revisions would improve the manuscript, focusing on areas of the rebuttal where I feel that the authors have not adequately addressed my initial concerns.

1. Haploinsufficiency – While I commend the authors for attempting the Tapestri experiments presented as a reviewer figure, I did not find their counterpoints to be very compelling. The cited sources do indicate that in some contexts and for some genes, a knockout might recapitulate some phenotypes of a gene that is normally heterozygously mutated in ASD. However, this shouldn't be taken as proof that most genes will function this way. I think the authors should add a sentence to the results (not discussion!) that clarifies that it is not possible to know whether the introduced mutations are heterozygous or homozygous. This is an important caveat that I think should be presented at the same time as the results so as to not confuse the reader.
2. Per IHC in Extended Data Fig 2a, Extended Data Fig 8d, Extended Data Fig 11a-d, Extended Data Fig 12a, etc. depicting eCas9 expression spatially throughout the organoid, it would appear that this is limited to the exterior of mature organoids where only more mature cell populations lie. As the authors reference in p9 of the rebuttal, the nature of the 4-OHT treatment on 3D cells limits its penetration to all cells in the 3D structure. The authors should recognize this as a limitation in the interpretation of perturbations in the discussion.
3. Related to point #2 above, the concentrated infection of the outermost layer of the organoid also raises an important issue related to cell-cell interactions which were a major topic of discussion in the first round of review. In particular, the authors argue the mutant-mutant interactions are limited because only 20% of the cells are infected. However, these 20% of cells are highly concentrated in the outer-most layer of the organoid. Meaning that the incidence of mutant-mutant cell interactions is much higher than what is presented in the authors' rebuttal. This is of course a non-controllable aspect of the experiment but one that I think the authors need to acknowledge more appropriately in the manuscript. Additionally, the authors should amend Line 135 "limiting the analysis to cell-autonomous phenotypes" as this assumption is not definitively validated.
4. The issue on validating knockdown at the mRNA or protein level remains unresolved. In the rebuttal, the authors argue that such experiments require aren't reliable because of antibodies or the inconsistent relationship between cutting and RNA abundance and protein abundance. I feel that the

authors should quite easily be able to infer how cutting affects both mRNA and/or protein abundance, at least for a top target like ARID1B.

5. Lines 304-306: Figure 3g is referenced only briefly without much additional interpretation that would help the reader.

6. It is unclear to me what the reader should take away from Extended Data Fig 13. It does not seem to provide additional context and no significant interpretation is provided (would be in Lines 241-244). The authors should consider removing or expanding upon this figure.

Referee #2 (Remarks to the Author):

We commend the authors for the extensive work undertaken in the revision, both experimental and computational, that significantly improved the robustness of the results. Also the rebuttal was elaborated in a very detailed and clear format, which helped the evaluation of the work.

While several of our major concerns have been addressed, we list below some observations that still need to be clarified by additional analysis, and some aspects that should be better elaborated in the text.

Referee #2 (Remarks to the Author):

The work by Li et al. is timely and important for advancing our understanding of the genetic and developmental architecture of autism spectrum disorders (ASD).

In particular, leveraging CRISPR-based perturbations, brain organoids and single cell analysis, the authors set out to identify cell type-specific transcriptional endophenotypes as well as the impact on neurodevelopmental trajectories and gene regulation of the loss-of function of 36 high-risk ASD genes. As a next step, they related the impact observed in the screening of ARID1B knock-out to the defects observed in cerebral organoids from iPSC of ASD patients harboring heterozygous ARID1B mutations. These results are thus potentially very relevant, both for shedding new light on the neurobiological basis of ASD and for the novelty of the technological implementation of CRISPR-based perturbation in cerebral organoids.

Unfortunately, however, major flaws in the experimental design and in the analytical strategy do not allow to support the key knowledge claims of the paper, at least in its current state. A thorough revision of the design is thus necessary to tackle the major concerns on the validity and accuracy of the findings, so as to render the work both robust and reproducible and meet the ambitious and laudable goal it set out to achieve.

Below is a list of the key issues, in the hope that it will be useful to guide authors in the reassessment of their strategy and the implementation of the new experiments.

We thank the reviewer for the enthusiastic and supportive comments. We have followed reviewer's suggestions and performed several lines of experiments, mainly including:

- 1) Repeat of the entire screen on 4-month-old brain organoids from 2 additional batches.
- 2) Systematic determination of gRNA representation and clone barcode on 4 batches of organoids in single cell or bulk experiments.
- 3) Repeat the entire scRNA-seq analysis on figure 2 and 3 with newly integrated data and perform bioinformatic analyses suggested by the reviewer.
- 4) Phenotypic validation of multiple genes in non-mosaic organoids with different experimental approaches.

These experiments resulted in a substantial amount of new data which greatly improved the accuracy of the findings and further demonstrated the robustness of the system.

Major concerns:

1) The numbers of cells for each perturbation (Extended Data Figures 5, in particular panel e, and 6) show a heavy imbalance: few perturbations in the mosaic organoids are overrepresented, while several others are under-represented (including control cells that are under-represented or absent in several of the 8 libraries). In particular, a main concern emerges from the fact that the 2 perturbations with the highest number of recovered cells (KMT2C and LEO1) are differentially enriched, respectively, in the dorsal and ventral regions of the UMAP. This is a major issue because the over-representation of cells mutated for these 2 genes is in all likelihood the main driver of the observed dimensionality reduction where the authors identify the 2 regions of dorsal and ventral cell populations. Thus, a UMAP computed only on control cells, as well as a UMAP where the dataset is downsampled to have the same number of cells per perturbation, are both necessary to properly interpret the validity of cell type annotation and, as a consequence, of all the downstream analysis on the effects of the individual perturbations.

The imbalanced representation of perturbations, together with the impossibility in a mosaic design to discern cell-autonomous from non-cell-autonomous effects derived from the interplay of different perturbations, require the main results of the study to be replicated with new experiments. As minimal requirement, for genes showing a significant change of cell abundance, the results should be validated in homogeneous (ie. non mosaic) organoids carrying only that specific perturbation; this would allow both to confirm the observations with a larger number of cells, and to discern cell autonomous effects from potential nonautonomous ones. Given the ambition of the work, and the scalability of the employed assays, this approach should ideally be employed for all the 36 genes, and it appears somewhat surprising why, also in light of the effective gRNA construct design that was chosen, the systematic assay of all individual perturbations separately was not adopted as the baseline. When growing new batches of organoids, one suggestion to back the idea that perturbations are selected would be to track in time, at known crucial stages of organoid generation (or every 15/30 days), gRNA compositions of small pools of organoids (e.g. 3/5 organoids per pool, 3/5 pools per time point).

We thank the reviewer for this thoughtful comment and have addressed in the following 4 sections.

1. Cell-cell interaction in mosaic organoids.

Since a similar point has also been raised by reviewer 1 (comment 19) and reviewer 3, and is highly relevant to the reviewer's remaining comment, we would like to address this upfront.

4-OHT was added to induce eCas9 (dTomato+) expression in 5 days old EBs. In our system, 4-OHT could not fully penetrate tissues. We have systematically titrated 4-OHT and found that when applied at a concentration between 0.3 ug/ml-0.6 ug/ml, 10-20% of the cells were induced (Day 30, GFP+/dTomato+, New Extended Data Fig. 5d). We also tracked the mutant population through development at day 20, 40, 60, 80, 100 and 120 (0.3 ug/ml 4-OHT) and observed stable, low percentages (~20%) of mutant populations across all stages (New Extended Data Fig. 5e). At day 120, we observed an average of 21.8% mutant population (New Extended Data Fig. 5f). This data suggests that most of the cells in the mosaic organoids are wild-type, which greatly limits mutant cell-cell interaction. In addition, samples were prepared from multiple pools of organoids and every mosaic organoid consists of 36 mutants. Thus, the chance that a specific mutant always neighbors another same mutant cell is extremely low. Such neighboring effects will also be averaged out when analyzing hundreds and thousands of cells for each perturbation and comparing them to the internal control cells. We have now included additional clarification in the revised manuscript.

2. gRNA representation analysis

To address this point and a similar point raised by reviewer 1 (see detailed response to reviewer 1, comment 5), we carried out additional single-cell (14 pools of organoids) and bulk experiments (24 pools of organoids) from 3 independent batches. We analyzed gRNA representation in three conditions (eCas9-induced cells, eCas9 uninduced cells from either 4-OHT treated or untreated organoids) (New Extended Data Fig. 7a). We consistently see over- and under- represented gRNAs in eCas9-induced cells but not in uninduced cells at day 120. This led to an interesting discovery of the over-proliferation (e.g., LEO1, KMT2C) and depletion (e.g. ADNP, WAC) phenotypes from the screen (New Fig. 2a, b). As suggested, gRNA representation was also examined at different development stages (Day 20, 40, 60, 80) and compared to internal uninduced cells sorted (New Extended Data Fig. 7b). We found that percentages of some genes, such as KMT2C and LEO1 already started to increase at day 20. The findings of the over-proliferation/depletion phenotypes were further validated by generating non-mosaic organoids individually for several perturbations (New Extended Data Fig. 8).

3. Computing new control UMAP for single cell analysis.

We agree that annotating the cells based on the full dataset might have been biased by perturbation effects. In the revised manuscript, we have therefore adopted a strategy similar to the one proposed by the reviewer with the aim to mitigate these biases. Manual annotation of cells is now purely based on control and uninduced cells (35,203 cells) (New Extended Data Fig. 6d), and cell type labels for the full dataset are derived from a label transfer. We found that perturbed cells generally adopted cell states resembling those found in the control, albeit with varying proportions.

4. Individual gene validation in non-mosaic organoids.

The reviewer is concerned about potential non-cell autonomous effects in the mosaic systems and thus suggests validating the findings in non-mosaic organoids. In our response to the reviewer's comment above, we describe that in the CHOOSE system, low percentages of cells (~20%) are eCas9-induced which greatly minimizes such effects.

We would like to emphasize that generating non-mosaic organoids with individual genes perturbed in an arrayed fashion is a completely different approach compared to the pooled CRISPR screening system. The arrayed screen is limited by scalability, although it offers opportunities for mechanistic follow-up studies. However, high-content single-cell screening can already identify complex biological phenotypes³⁰, which is a key strength of the CHOOSE system. Thus, we think that re-examining all perturbations separately extends beyond the scope of the current study. Especially since our system offers controlled setting for all perturbations during lentivirus production, infection of stem cells and collection, organoids generation, sample preparation and scRNA-seq, once gRNA plasmid library has been constructed. Mutant cells could, thus, be compared to the controls bearing the same environment, which allows us to isolate the phenotypes that are purely contributed by perturbations.

To further support our findings, we set out to validate the key phenotypes for several perturbations in non-mosaic organoids as suggested by the reviewer. For this, we generated individually perturbed organoids (New Extended Data Fig. 8, 11, and 12) and carried out three lines of phenotypic validation experiments:

- 1) We verified the over-proliferation and depletion phenotypes seen upon perturbation of LEO1, KMT2C, ADNP and WAC. We used a FACS-based approach to quantitatively analyze the ratio of mutant cells (GFP+/dTomato+) to non-mutant cells (GFP+/dTomato-) that were isolated from the same organoids at two development stages. The ratio was then compared to a non-targeting control gRNA.

Overproliferation of LEO1 and KMT2C, and depletion of ADNP and WAC were consistently observed at both time points (New Extended Data Fig. 8).

2) We verified the depletion of intermediate progenitors (IP) upon perturbation of KMT2C and PHF3. We performed immunohistochemistry (IHC) analyses of EOMES positive (a marker for IPs) cells at day-60-old organoid tissues. At this stage, the organoids present an organized cytoarchitecture, with clearly separated ventricular zone (VZ), subventricular zone (SVZ) and cortical plate (CP), which allows reliable IHC measurement. Percentages of IPs in mutant cells-populated areas were quantified and compared to organoids infected with control gRNA (New Extended Data Fig. 11). We observed significant depletion of IPs in KMT2C and PHF3 mutants, consistent with our findings from the screen.

3) We verified the increase of ventral progenitors upon perturbation of KMT2C, MED13, PHF3 and TBL1XR1. We performed IHC analyses of DLX2 positive cells, a marker labels interneuron precursors³¹. Our protocol presents both dorsal and ventral regions at the early stages^{32,33}, and allows us to examine INPs already at Day 60. We found significantly increased DLX2 positive cells in the mutant rosettes area for perturbations of KMT2C, MED13, PHF3 and TBL1XR1 (New Extended Data Fig. 12).

Reviewer comment: We commend the authors for the work performed thanks to the revision, which gives stronger robustness to the results, and appreciate the fact that studying several perturbations in non-mosaic organoids allowed them to validate key phenotypes. While the individual gene validation cell-cell interactions and gRNA representation points were better clarified, we are still not able to properly evaluate the annotation of the cells: the authors have manually annotated the embedding only on control and uninduced cells, as suggested, however it is not possible to evaluate this part since they only plotted the expression of marker genes in the UMAP embedding including all cells. The distribution of marker genes, as well as the control gRNA vs uninduced cells labels, and batches labels should be also plotted for the embedding of Extended Data Fig.6d. This is particularly relevant to understand, for example, why the cluster labelled as astrocytes progenitors in Extended Data Fig.6d, which is the most separated from the rest of the embedding, was annotated as astrocytes in the UMAP of fig 1f by label transfer, even if there are only perturbed cells there according to Extended Data Fig.6e.

Finally, while we concur on the utility of pooled CRISPR screening systems, as a different approach relative to the set-up of non-mosaic organoids with individual genes perturbed in an arrayed fashion, we suggest to edit the language of the discussion: “We have developed the CHOOSE system to characterize the loss-of-function phenotypes of high-risk ASD genes across dozens of cell types spanning early brain developmental stages in human cerebral organoids. Our findings provide a developmental and cell-type specific phenotypic database for ASD high-risk gene loss-of-function research, which will shed light on the disease pathogenesis” to clarify that this is a powerful tool for screening when complemented by validation, as indeed the authors did in this study.

2) The experimental design followed by the authors is based on the pooling of 3-6 organoids for each of the 8 chromium controller runs. There are thus no independent replicates that can be used in the analysis to evaluate the reproducibility of the observed phenotypes. Indeed, it is truly remarkable that the word “replicate” never appears in the paper. All results in figures 2 and 3 (and associated extended data figures) about the effects of the perturbations in terms of ratio of dorsal to ventral cell populations, specific enrichment/depletion for a particular cell type, and differential densities along pseudotemporal axis need replicates to be interpreted.

- The two above issues imply that it is not possible to evaluate the validity and accuracy of all the main results of the paper: figure 2 and 3.

We thank the reviewer for pointing this out. We agree that multiple biological replicates are crucial for validating our results. In the original manuscript, we had performed the screen in 8 independent pools of organoids, which we consider to be biological replicates. In the revised manuscript, we first have added 3 more replicates from Batch 2, an independently grown batch of organoids (New Extended Data Fig. 7a, eCas9 induced-cells, pink). We have reanalyzed the data including the newly generated batch by testing (as we did previously) on a single cell level, using replicate as a covariate to control for biases stemming from differential cell numbers between replicates. We have updated all relevant main and supplemental figures including cell type abundance testing, differential gene expression analysis as well as GRN enrichment.

Furthermore, this newly integrated data now allowed us to evaluate the consistency of effects across independent pools of organoids (14 biological replicates, 3 batches) from multiple batches (including another 3 replicates from a third independent batch, using non-induced cells as an alternative control) to complement the analysis performed on the single-cell level (New Extended Data Fig. 10). Here, we treated every pool of organoids as an independent sample (rather than every cell) and performed a t-test on the per-organoid pool fold enrichment of each guide. This approach can be used to evaluate the variability of the effect across library replicates from different batches, but it has less power compared to the testing on the single-cell level. We are glad to see that many of the effects highlighted in the study (e.g., depletion of IP in KMT2C, BCL11A, CHD2, KDM5B etc. perturbations, enrichment of INP in CIC, IRF2BPFL, MED13, PHF3, TBL1XR1, TCF20 etc. perturbations, vRG enrichment in ARID1B perturbation) were significant also in this test, which supports their strong consistency across independently grown organoids. Although it is worth noting that if effects were detected on a single-cell level but not on the level of pools of organoids this might stem from higher variability of the effect across organoids pools but does not invalidate the effect seen on the single cell level. Overall, we found that the major effects on cell type abundance changes highlighted in the original manuscript were corroborated by both analyses.

Reviewer comment: We thank the authors for having clarified the replicates in the study.

3) The authors presented an elegant and novel strategy to generate efficient and controlled pooled systems with high clonal complexity in pluripotent stem cells, not in organoids as they claim, since there is no data showing the unique clone barcode distribution after differentiation of the organoids. This is instead a result of obvious relevance that should be shown.

We thank the reviewer for appreciating the barcoded system and apologize for missing this important aspect of the CHOOSE system. We have performed targeted amplification and retrieved clone information from both scRNAseq experiments as well as additional bulk genomic DNA analysis (New Extended Data Fig. 4c-h). For scRNA-seq, we have recovered on average 125 clones for each perturbation (median: 106) and the clones are distributed across all libraries (New Extended Data Fig. 4c, d). We also analyzed the size of each clone. The mean average cell number in each clone per perturbation is 4.4 (with median of 3.6; New Extended Data Fig. 4 e, f). These data suggest that cells captured in the CHOOSE screen came from diverse and relatively small clones, which is important for eliminating dominant clonal effects. Bulk analysis of the genomic DNA with a much higher cell number input (50150K) reveals the high clonal complexity in both eCas9-uninduced and induced cells, with a homogenous distribution in uninduced cells as expected (New Extended Data Fig. 4g, h). Interestingly, over-proliferation/depletion phenotypes could also be reflected by clone analysis, as the gene LEO1 presents the highest clone number and averaged clone size. Together, these data underscore the great advantage of barcoding in tissue-based screening.

Reviewer comment: We thank the authors for having added data on clone distribution in the organoids. While we understand it is expected not to have a perfect balance of clones after

differentiation, the imbalance observed in Extended Data Fig. 4 between the perturbed genes should be taken into account, for example by downsampling, when performing the downstream molecular analysis (ie differential expression analysis) to understand the transcriptional impact of each perturbation in the scRNASeq data.

Minor note: the authors comment in the rebuttal about LEO1 as the gene with highest clones number, but from the plots KMT2C seems to have the highest number.

4) Results from differential abundance analysis should be strengthened by applying algorithms that have been developed for scRNASeq frameworks such as scCODA that account for the inherent bias present in cell-type compositions with a Bayesian approach for cell-type composition to address the low replicate issue, and/or MILO, thus does not rely on discrete clusters as input when testing for differential abundance among experimental conditions, thus improving the detection of differences in continuous trajectories.

We thank the reviewer for alerting us to these recently developed tools for differential abundance analysis. In the manuscript we employed a Cochran–Mantel–Haenszel statistic which allowed us to control for biases stemming from differential cell numbers and perturbation gRNA representations between replicates. We were curious to see if the results of our analysis would be confirmed by MILO which was specifically developed for this purpose. While we noticed that pooled CRISPR screens violated the 1 observation = 1 experimental sample assumption made by both MILO and scCODA, we were able to apply

MILO to our data using workarounds suggested by the authors (<https://github.com/MarioniLab/miloR/issues/259>). We found that the analysis with MILO confirmed many of the results that we had seen previously, including extensive depletion of IPs in multiple perturbations, enrichment of OPCs in ARID1B, enrichment astrocytes in CIC as well as a widespread depletion of dorsal telencephalon and/or enrichment of ventral telencephalon fates (e.g., KMT2C, BCL11A, KAT2B) (Reviewer Fig. 4). While we found that these results strengthened the confidence in our previous analysis, we reasoned that including them in the manuscript could be confusing to the reader. Thus, we include this as a reviewer figure.

Reviewer Fig. 4 Differential abundance testing using MILO. a, UMAP embedding showing cell type annotations. b, MILO analysis performed with library (replicate) as a covariate as suggested by the authors via GitHub. The results of MILO for 3 target genes with strong effects in our analysis are displayed as UMAP embeddings showing differential abundance in individual neighborhoods (top) and barplots with fold changes aggregated to cell type (bottom).

5) In figure 2A, it would be useful to have relevant about the magnitude of the enrichment in addition to the significance.

As suggested, we have changed the plots throughout the manuscript (New Fig. 2c, 3a) to better display the effect size of both composition changes and differential expression.

6) For the genes that show an impact on the abundance of specific cell populations, it would be interesting to check if the detected effect is coherent with the gene expression patterns across cell types in the organoid model in study as well as in datasets from fetal brain at comparable stages.

We thank the reviewer for bringing up this interesting point. We checked the expression of the perturbed genes in both our organoids dataset and a recently published fetal brain development dataset at 5-14 pcw34 (New Extended Data Fig. 6i, j). Indeed, we see several perturbed genes with

cell-type specific enrichment, especially in organoids. For example, the three BAF complex members have very distinct expression pattern: ARID1B is highly expressed in all three types of interneurons as well as L23 EN; BCL11A is highly expressed in IP, L6_ThPN and L23 EN; and SMARCC2 expresses at low levels in all cell types. This probably suggests a cell-type specific requirement of individual BAF complex members during neural stem cell lineage progression³⁵. Indeed, ARID1B and BCL11A have larger effects on multiple cell types including IP, EN and INs, while SMARCC2 perturbation only affects vRG.

Reviewer comment: We thank the authors for including in the manuscript this new relevant piece of information. We however did not find a description in the methods of how the heatmap values have been calculated.

7) Have the expression levels of targeted genes been checked in the results of the differential expression analysis? For each guide the expression level of the target gene should be shown for the perturbed cells vs the others. This would be instrumental also for assessing the relevance of the dosage difference between the ARID1B knock-out and the ARID1B haploinsufficient backgrounds, and hence properly evaluating this validation aspect of the work (Fig. 4)

We thank the reviewer for the comment. Using gene expression levels to reflect CRISPR editing outcomes can be misleading. This is because edited gene sequences can still be transcribed to mRNAs. Although it is also true that in many cases, CRISPR editing results in a premature stop codon, which may trigger nonsense-mediated mRNA decay. As suggested, we checked DE for each perturbed gene in dorsal and ventral populations (Reviewer Fig. 5). As expected, we found dysregulated genes in some but not all perturbations.

Reviewer Fig. 5

DE analysis of perturbed genes. Dot plot showing differential expression of the gRNA target gene in perturbed cells for each the dorsal and ventral telencephalon developmental trajectory. Size indicated $-\log_{10}(\text{p-value})$, color indicates log fold change.

To check the protein levels of ARID1B, we generated organoids with ARID1B perturbation and measured by IHC (New Extended Data Fig. 2). Our results suggest an average of 38.6% reduction of protein levels in eCa9-induced cells compared to eCas9-uninduced cells. This reduction is not caused by eCas9 expression as similar protein expression levels are detected in eCas9-induced and uninduced cells from organoids infected with non-targeting control gRNA. This data highlights the relevance of the phenotypes observed in the screen compared to organoids derived from ARID1B heterozygous patient iPSCs.

8) From the Manhattan plot in Figure 2, the number of DEGs is quite low for most of the examined conditions. This is quite puzzling when compared to the results about cell type abundance. How do the authors explain this striking discrepancy?

We also found it interesting that the gene expression changes induced by the perturbations were rather subtle compared to changes in cell type composition. The ability to detect such subtle differential expression changes in our screen is influenced by the number of cells we detect per target gene and cell state and are aware that our statistical power is limited for some targets. We were therefore pleased to see that the inclusion of more replicates almost doubled the number of detected DEG. Overall, we detect DE genes for all perturbed targets and show that they are involved in important pathways and regulomes (New Extended Data Fig. 14c, Fig. 3a-c, f, Supplementary Data 2-4, 7).

Furthermore, we think that strong compositional changes need not correspond to strong expression changes in the differentiated cell type. For instance, a perturbation could induce small alterations in gene expression during development thereby impacting the outcome of differentiation and cell fate decision events, but which are not maintained in the terminally differentiated cell type.

9) Is the statement about the involvement of the ubiquitin-proteasome pathway substantiated by a functional enrichment analysis or other analytical techniques?

Finally, most observations are solely grounded on differential expression analyses with no significant validation of at least key findings. At least basic biochemical assays are required, for specific perturbations, to corroborate/support claims and observations such those on mitochondrial and proteasome pathways dysregulation.

We thank the reviewer for the comments. As they are related, we would like to address them here together. As suggested, we now performed systematic GO enrichment analysis with dorsal and ventral DEGs separately for each perturbation (New Extended Data Fig. 14c, full list in Supplementary Data 2). We are glad to see that pathways involved in protein misfolding and mitochondria function are enriched in multiple perturbations, including ASHL1 (dorsal), DDX3X (dorsal), FOXP1 (dorsal), KDM5B (ventral), KMT2A (ventral). Excitingly, we also observed many enriched GO terms covering diverse functions, many of which are confirming previous studies, such as ribosome assembly (SETD5 perturbation)³⁶, forebrain development (TBR1 perturbation)³⁷, mitochondrion organization (FOXP1 perturbation)³⁸, lipid homeostasis (IRF2BPL perturbation)³⁹, autophagosome maturation (KAT2B perturbation)⁴⁰, and cilium development and histone demethylation (MECP2 perturbation)⁴¹, which further support the power of detecting complex biological phenotypes in the CHOOSE system. We thus decide to not only focus on the mitochondrial and proteasome pathways, but rather highlight all these consistent observations and provide a comprehensive GO enrichment list which can serve as a resource for future studies of under-explored genes. We thank the reviewer for this helpful suggestion and have added this new analysis and edited the text in the section of 'CHOOSE identifies dorsal and ventral telencephalon-specific dysregulated genes' (Page 8-9, highlighted in blue).

Reviewer comment: We are glad that our suggestion allowed to include a further interesting resource in the article, even though as suggested before (point 3) the limits of a differential expression analysis performed between groups of very different size should be addressed by downsampling and adequately thematized.

10) The CHD8 perturbation does not show the impact on cell population abundance that was reported by several papers in similar organoid models and in vivo (e.g Paulsen et al. 2022; Villa et al. 2022; Jin et al. 2020). What is the explanation for this (which should anyway be discussed in the paper)?

The Paulsen et al. paper identified increased interneurons and interneuron progenitors at 3.5-month-old organoids in the HUES66 cell line but not in two other cell lines (GM08330 and H1). In their in vivo mouse studies (Jin et al. 2020), perturbation of CHD8 does not lead to any cell type compositional changes, but a gene expression module associated with oligodendrocyte progenitor (ODC1). Olig1 is also dysregulated in CHD8 perturbation in our study (Supplementary Data 2). A more thorough study on CHD8 from Villa et al. 2022 identified increased interneuron generation at D60, but not at D120 in two cell lines. These results (no interneuron changes at D120 in 4 out of 5 cell lines, and no changes in in vivo studies) are in fact very much in line with our observation of a negative result. In addition, the Villa et al. paper very nicely confirmed that phenotypes such as enlarged organoids size and proliferation associated with macrocephaly are human mutation-specific. In our screen, we observed

a trend of increased interneuron population, as well as cycling ventral radial glia cells, but it is not significant. Given these results, we think the fact that we did not capture cell abundance changes can be due to a small effect size of CHD8 on cell fate determination or to the phenotypes being more mutation-specific, although it can also be due to the power of our experimental approach.

Reviewer comment: We concur with the authors that alterations of cell abundance induced by haploinsufficiency or inactivation of ASD-related genes can be transient, manifesting in specific stages of development (as described by several works in the field). Being this an important aspect also for this work, it should be mentioned and discussed in the manuscript.

Referee #3 (Remarks to the Author):

The authors have adequately addressed all concerns.

Comments from Reviewer #1:

In this substantial revision, Li & Fleck et al. tempered and expanded their analyses of neurodevelopmental defects underlying Autism Spectrum Disorder (ASD). Major suggested revisions focused on: 1) increasing sample size and independent replicates to bolster findings; 2) validation of findings of mosaic screens in non-mosaic organoids individually targeting one gene at a time; 3) addressing technical limitations of CRISPR-based gRNA, such as guide and clonal diversity after EB formation and heterozygotic vs. homozygotic knockout; and 4) improved organization and presentation of data in streamlined figures. The authors made progress to address numerous criticisms and adjusted the language in the manuscript when necessary to recognize key limitations of the experimental design and model systems used. Through increased and independent replication, their main findings on alteration to cell type proportions due to ARID1B deficiency have been significantly bolstered. Validation and QC of the system, such as quantification of %eCas9 dTomato+ cells per organoid, has clarified the technical limitations. In total, the authors should be commended for performing a substantial and well-organized overhaul of their initial submission. In particular, I'd like to thank the authors for making the presentation of the revised manuscript easier to review given its organization (table of changes, etc) – the effort that went into this is not lost on me. Below I outline areas where I believe minor (but still important) revisions would improve the manuscript, focusing on areas of the rebuttal where I feel that the authors have not adequately addressed my initial concerns.

We thank the reviewer again for the thorough evaluation and positive comments of our substantial revision! Below we addressed each of the minor comments and we hope the reviewer now finds the manuscript suitable for publication.

1. Haploinsufficiency – While I commend the authors for attempting the Tapestry experiments presented as a reviewer figure, I did not find their counterpoints to be very compelling. The cited sources do indicate that in some contexts and for some genes, a knockout might recapitulate some phenotypes of a gene that is normally heterozygously mutated in ASD. However, this shouldn't be taken as proof that most genes will function this way. I think the authors should add a sentence to the results (not discussion!) that clarifies that it is not possible to know whether the introduced mutations are heterozygous or homozygous. This is an important caveat that I think should be presented at the same time as the results so as to not confuse the reader.

We added a sentence to the results section in addition to the discussion to avoid any potential confusion.

The revised text reads, on page 4, Line 103 – 106:

'Successful genome editing causes frameshift mutations that lead to the loss of BFP fluorescence, allowing quantitative evaluation of gRNA efficiency in a large cell population (Fig. 1c, Extended Data Fig. 1b, c), although it does not allow for the determination of whether a heterozygous or homozygous mutation was introduced.'

2. Per IHC in Extended Data Fig 2a, Extended Data Fig 8d, Extended Data Fig 11a-d, Extended Data Fig 12a, etc. depicting eCas9 expression spatially throughout the organoid, it would appear that this is limited to the exterior of mature organoids where only more mature cell populations lie. As the authors reference in p9 of the rebuttal, the nature of the 4-OHT treatment on 3D cells limits its penetration to all cells in the 3D structure. The authors should recognize this as a limitation in the interpretation of perturbations in the discussion.
3. Related to point #2 above, the concentrated infection of the outermost layer of the organoid also raises an important issue related to cell-cell interactions which were a major topic of discussion in the first round of review. In particular, the authors argue the mutant-mutant interactions are limited because only 20% of the cells are infected. However, these 20% of cells are highly concentrated in the outer-most layer of the organoid. Meaning that the incidence of mutant-mutant cell interactions is much higher than what is presented in the authors' rebuttal. This is of course a non-controllable aspect of the experiment but one that I think the authors need to acknowledge more

appropriately in the manuscript. Additionally, the authors should amend Line 135 “limiting the analysis to cell-autonomous phenotypes” as this assumption is not definitively validated.

During brain organoid development, the tissues undergo rapid and dramatic structural reorganizations by first forming multiple neural rosettes (progenitor centers) within each organoid. The progenitors then proliferate at the ventricular zones, differentiate into neurons, and radially migrate into the cortical plate to form different layers¹. The induction of eCas9 was performed at **5-day-old EB stage**. This is even before the stem cells committed to neuroectoderm, and before the neural rosettes formation. Thus, the penetration nature of eCas9 in EBs will not lead to the mutant cells being ‘concentrated’ in the ‘outermost layer’ of the organoids. Actually, the ‘outermost layer’ neurons are generated from these mutant progenitors at the ventricular zones. Additionally, in the Extended Data Fig. 5c, we have shown that mutant cells (eCas9, dTomato) are distributed across all the rosettes at an early state (day 28). We have now added a zoom in image as an inset to give a clearer presentation, also included here:

We also provide here a high-resolution image (see below) for Extended Data Fig. 8d (Control, dTomato channel). It shows that the mutant cells are present throughout the ventricular zones, subventricular zones as well as the cortical plate, and are not limited to the exterior (zoom in **a'**, **a''**). Although, postmitotic neurons, which are distributed mostly in the cortical plate region, might have higher eCas9 protein levels compared to the progenitors (constantly dividing) at the ventricular zone and subventricular zone.

The penetration nature of 4-OHT in 5-day-old EB could limit the full induction of mutants in the entire tissue. We do not see this as a limitation of our system, as it is possible to achieve higher induction by optimizing 4-OHT dosage and induction timing (e.g., induction at earlier development stages such as day 0 when making EBs). With titration of 4-OHT treatment, we have partially induced cells (~20%), which is an advantage for minimizing cell-cell interaction effects.

Regarding cell-cell interaction, we followed the reviewer's suggestion and revised the sentence in Line 135. The revised text reads, on page 5 , Line 136: *This very likely could limit the mutant cell-cell interactions in the mosaic tissues.*

4. The issue on validating knockdown at the mRNA or protein level remains unresolved. In the rebuttal, the authors argue that such experiments require aren't reliable because of antibodies or the inconsistent relationship between cutting and RNA abundance and protein abundance. I feel that the authors should quite easily be able to infer how cutting affects both mRNA and/or protein abundance, at least for a top target like ARID1B.

We thank the reviewer for the comment. We have actually quantified the mRNA levels from single cell RNA-seq dataset for all gRNAs in previous revision (previous Reviewer Fig. 5). We have also measured the protein levels for several individually perturbed genes (Extended Data Fig. 2), including the top target ARID1B (which has a 38.6% reduction on protein levels in the mutant cells, even though mRNA levels remain unchanged in scRNAseq dataset).

5. Lines 304-306: Figure 3g is referenced only briefly without much additional interpretation that would help the reader.

We thank the reviewer for the comment and now added additional interpretation. The revised text reads, on page 10 , Line 303-309:

“We finally present gene regulatory subnetworks of OLIG1 and EOMES, whose regulomes are both enriched in ASD risk genes and strongly affected by ASD genetic perturbations (Fig. 3g). OLIG1 is preferentially expressed in the ventral telencephalon and is one of the key regulators for interneuron and oligodendrocyte lineages². EOMES is a key transcription factor for the fate specification of intermediate progenitors in the dorsal telencephalon³. The enrichment of OLIG1 and EOMES regulomes suggest potentially vulnerable cell fate specification-related regulatory networks upon ASD genetic perturbations.’

6. It is unclear to me what the reader should take away from Extended Data Fig 13. It does not seem to provide additional context and no significant interpretation is provided (would be in Lines 241-244). The authors should consider removing or expanding upon this figure.

We have removed Extended Data Fig 13.

Comments from Reviewer #2:

We commend the authors for the extensive work undertaken in the revision, both experimental and computational, that significantly improved the robustness of the results. Also the rebuttal was elaborated in a very detailed and clear format, which helped the evaluation of the work. While several of our major concerns have been addressed, we list below some observations that still need to be clarified by additional analysis, and some aspects that should be better elaborated in the text.

We thank the reviewer for the positive comments on our extensive revision. Below we addressed each of the comments with additional analysis and clarification and we hope the reviewer now finds the manuscript suitable for publication.

1) We commend the authors for the work performed thanks to the revision, which gives stronger robustness to the results, and appreciate the fact that studying several perturbations in non-mosaic organoids allowed them to validate key phenotypes. While the individual gene validation cell-cell interactions and gRNA representation points were better clarified, we are still not able to properly evaluate the annotation of the cells: The authors have manually annotated the embedding only on control and uninduced cells, as suggested, however it is not possible to evaluate this part since they only plotted the expression of marker genes in the UMAP embedding including all cells. The distribution of marker genes, as well as the control gRNA vs uninduced cells labels, and batches labels should be also plotted for the embedding of Extended Data Fig.6d. This is particularly relevant to understand, for example, why the cluster labelled as astrocytes progenitors in Extended Data Fig.6d, which is the most separated from the rest of the embedding, was annotated as astrocytes in the UMAP of fig 1f by label transfer, even if there are only perturbed cells there according to Extended Data Fig.6e.

As suggested, we have now added marker gene plots for only control gRNA and uninduced cells (**New Revised Extended Data Fig. 6g**). The expression patterns of these marker genes further support the annotation of the cell clusters. The astrocyte cluster can be confidently annotated with high expression of S100B, APOE and ALDH1L1. In addition, we added the plots with control gRNA/uninduced cells and batch labels (**New Revised Extended Data Fig. 6d-f**).

We apologize for the mistake made in the color and name assigned to the 'astrocyte progenitors' cluster, which should actually be labeled as 'astrocytes'. We have now corrected this.

Finally, while we concur on the utility of pooled CRISPR screening systems, as a different approach relative to the set-up of non-mosaic organoids with individual genes perturbed in an arrayed fashion,

we suggest to edit the language of the discussion: “We have developed the CHOOSE system to characterize the loss-of-function phenotypes of high-risk ASD genes across dozens of cell types spanning early brain developmental stages in human cerebral organoids. Our findings provide a developmental and cell-type specific phenotypic database for ASD high-risk gene loss-of-function research, which will shed light on the disease pathogenesis” to clarify that this is a powerful tool for screening when complemented by validation, as indeed the authors did in this study.

We have changed the text of the discussion. The revised text reads, on page 11-12, Line 357 – 361:

‘We have developed the CHOOSE system to characterize the loss-of-function phenotypes of high-risk ASD genes across dozens of cell types spanning early brain developmental stages in human cerebral organoids. By employing a pooled CRISPR screening system in conjunction with validation, our study provides a developmental and cell-type specific phenotypic database for ASD high-risk gene loss-of-function research, which will shed light on the disease pathogenesis.’

2) We thank the authors for having clarified the replicates in the study.

3) We thank the authors for having added data on clone distribution in the organoids. While we understand it is expected not to have a perfect balance of clones after differentiation, the imbalance observed in Extended Data Fig. 4 between the perturbed genes should be taken into account, for example by downsampling, when performing the downstream molecular analysis (ie differential expression analysis) to understand the transcriptional impact of each perturbation in the scRNASeq data.

Regarding GO analysis: We are glad that our suggestion allowed to include a further interesting resource in the article, even though as suggested before (point 3) the limits of a differential expression analysis performed between groups of very different size should be addressed by downsampling and adequately thematized.

We agree with the reviewer that the imbalance of perturbed cells causes differential sensitivity between perturbations, which affects the number of DEGs that we are able to robustly detect. This makes it challenging to quantify and compare the magnitude of the total perturbation effect in terms of the total number of DEG and we have therefore refrained from doing so in the manuscript.

While downsampling would make the transcriptional impact more comparable, it would also strongly reduce sensitivity for highly sampled perturbations. Instead of downsampling, we have therefore addressed this for the GRN analysis by balancing contribution of DEG across target genes (selecting top 30 DEG based on absolute fold change, defined as TOP-DEG; Methods, Differential expression analysis). In this way we hope to retain the ability to detect differential effects in the most robust and sensitive manner for each perturbation, while accounting for absolute DEG number in otherwise potentially biased downstream analyses (GRN). We realized that we did not explicitly describe this in the results section in our previous version and only mentioned in the figure legend and methods, we have now added this information in the main text to make it clearer for readers (Page 8-9, Line 258-263).

We have also now repeated the global GO enrichment analysis in the same way by using only TOP-DEG (**New Revised Fig. 3c**). For the perturbation-specific GO analysis we have chosen to use all detected DEG per perturbation to maintain sensitivity, as we have successfully confirmed many previous studies.

Minor note: the authors comment in the rebuttal about LEO1 as the gene with highest clones number, but from the plots KMT2C seems to have the highest number.

We thank the reviewer for pointing it out. Gene LEO1 has the *second highest* clone numbers and *highest averaged* clone size.

4) *New Extended Data Fig. 6i, j*

We thank the authors for including in the manuscript this new relevant piece of information. We however did not find a description in the methods of how the heatmap values have been calculated.

We added a description in the methods (Page 22, Line 705-711).

5) We concur with the authors that alterations of cell abundance induced by haploinsufficiency or inactivation of ASD-related genes can be transient, manifesting in specific stages of development (as described by several works in the field). Being this an important aspect also for this work, it should be mentioned and discussed in the manuscript.

We thank the reviewer for this suggestion and have added additional text in the discussion. The revised text reads, on page 13, Line 407 – 410:

'Furthermore, it has been shown that the effects of perturbations of ASD risk genes on cell type abundances can sometimes be transitory'^{11,57}. Consequently, we may not be able to capture these abnormalities during development for certain perturbations.'

Comments from Reviewer #3:

The authors have adequately addressed all concerns.

References:

1. Lancaster, M. A. & Knoblich, J. A. Generation of cerebral organoids from human pluripotent stem cells. *Nat Protoc* 9, 2329–2340 (2014).
2. Silbereis, J. C. *et al.* Olig1 Function Is Required to Repress Dlx1/2 and Interneuron Production in Mammalian Brain. *Neuron* 81, 574–587 (2014).
3. Sessa, A., Mao, C., Hadjantonakis, A.-K., Klein, W. H. & Broccoli, V. Tbr2 Directs Conversion of Radial Glia into Basal Precursors and Guides Neuronal Amplification by Indirect Neurogenesis in the Developing Neocortex. *Neuron* 60, 56–69 (2008).